



# Relocation of earthquakes in the Southern and Eastern Alps (Austria, Italy) recorded by the dense, temporary SWATH–D network using a Markov chain Monte Carlo inversion

Azam Jozi Najafabadi[1,2], Christian Haberland[1], Trond Ryberg[1], Vincent Verwater[3], Eline Le Breton[3], Mark R. Handy[3], Michael Weber[1], and the AlpArray working group[*]

[1]GFZ German Research Centre for Geosciences, Potsdam, Germany
[2]Institute of Geosciences, Potsdam University, Potsdam, Germany
[3]Institute of Geological Sciences, Freie Universität Berlin, Berlin, Germany
[*]For further information regarding the team, please visit the link which appears at the end of the paper

**Correspondence:** Azam Jozi Najafabadi (azam@gfz-potsdam.de)

**Abstract.** Local earthquakes with magnitudes in the range of 1 - 4.2 ($M_L$) in the Southern and Eastern Alps (2017 - 2018) registered by the dense, temporary SWATH-D network and the AlpArray network reveal seismicity in the upper crust (0-20 km). The seismicity is characterized by pronounced clusters along the Alpine frontal thrust, e.g., Friuli-Venetia (FV) region, in the Giudicarie-Lessini (GL) and Schio-Vicenza domains, as well as in the Austroalpine Nappes and the Inntal area. Some

seismicity also occurs along the Periadriatic Fault. The general pattern of seismicity reflects head-on convergence of the Adriatic Indenter with the Alpine orogenic crust. The deeper seismicity in the FV and GL regions indicate southward propagation of the Southern Alpine deformation front (blind thrusts). The first arrival-times of P- and S-waves of earthquakes are determined by an automatic workflow and then visually/manually checked and corrected. We applied a Markov chain Monte Carlo inversion method to achieve precise hypocenter locations of the 344 local earthquakes. This approach simultaneously calcu-

lates hypocenters, 1-D velocity model, and station-corrections without prior assumptions such as initial velocity models and earthquake locations. A further advantage of the method is the derivation of the model parameter uncertainties and noise levels of the data. The accuracy of the localization procedure is checked by inverting a synthetic travel-time dataset from a complex 3-D velocity model and using the real stations and earthquakes geometry. The location accuracy is further investigated by the relocation of quarry blasts. The average uncertainties of the locations of the earthquakes are below 500 m in the epicenter and

∼1.7 km in depth when using the average $V_P$ and $V_P/V_S$ models and the station-corrections from the simultaneous inversion.

## 1 Introduction

The Alpine Orogen resulted from the collision of the European and Adriatic plates (Dewey et al., 1973; Schmid et al., 2004; Handy et al., 2010, 2015). This collision led to localized deformation patterns within the plates as well as along the suture, often accompanied by seismic activity. The seismicity of the Alps has been investigated both on the scale of the Alpine orogen

(e.g., Diehl, 2008) and on the scale of parts of the chain (e.g., the Southern Alps; Amato et al., 1976; Cagnetti and Pasquale., 1979; Galadini et al., 2005; Anselmi et al., 2011; Scafidi et al., 2015; Bressan et al., 2016; Slejko, 2018; Viganò et al., 2008,





2013, 2015). The Southern and Eastern Alps have heterogeneous earthquake distribution, with regions of high activity, e.g., eastern Southern Alps (ESA), rare activity, e.g., Eastern Alps, or inactivity, e.g., parts of western Southern Alps (WSA) (Fig. 1).

The dense temporary seismic SWATH–D network was deployed from 2017 to 2019 in the Southern and Central Alps (Heit et al., 2017 and Heit et al., submitted). With an average station spacing of only 15 km, this network was designed to capture the shallow crustal seismicity (especially the depths of the earthquakes) with higher accuracy than the coarser permanent networks, including the AlpArray network (AASN with 52 km station spacing; Hetényi et al., 2018), and to obtain an increased spatial resolution of images (e.g., local earthquake tomography, receiver functions) even in a region with low or moderate local

seismicity. The SWATH–D network is located in a key part of the Adriatic indenter for which a switch in subduction polarity was proposed at the transition from Central to Eastern Alps (e.g., Lippitsch et al., 2003).

In this study, we use a subset of 344 local earthquakes recorded by the SWATH–D network and complemented by a selection of 112 AlpArray network stations nearby to precisely relocate the hypocenters of the local earthquakes. This will enable us to identify the status of the seismically active volume and its extension at depth. A further aim of the study is to derive a

high-quality dataset suitable to be used in local earthquake tomography.

Although locating the earthquakes has been a routine task in seismology for decades, there are several challenges related to obtaining a precise location. One of them is the trade-off between the hypocenters and the velocity structure (so-called hypocentre–velocity coupling; Kissling, 1988; Thurber, 1992; Kissling et al., 1994). To yield accurate locations (especially depths), either the velocity model should be well known in advance, or, particularly in the case of earthquakes occurring

within a network (local earthquakes), it has to be simultaneously inverted for (Kissling, 1988; Kissling et al., 1994; Thurber, 1983; Husen et al., 1999). This is conventionally being done by employing iterative inversion strategies based on damped least squares. These methods are quite robust and have been successfully applied for years. However, because they use a linearization and the damped least squares approach, they depend not only on proper choices of initial values for hypocenter coordinates and the velocity model(s) but also on technical parameters such as (fixed) model parametrization (i.e., layers) or damping factors

that all have to be carefully checked and selected.

Recently, a transdimensional, hierarchical Bayesian approach utilizing a Markov chain Monte Carlo (McMC) algorithm was implemented for simultaneous inversion of hypocenters, 1–D velocity structure, and station–corrections particularly for the local earthquake case (Ryberg and Haberland, 2019). This approach has the advantage of being independent of prior knowledge of model parameters (starting velocity model, starting locations of earthquakes) and the inversion results are data-

driven. Moreover, the results can be statistically analyzed, and thus errors are estimated. The method extends the probabilistic relocation approaches (Lomax et al., 2000) by also inverting for the best velocity model(s).

## 2 Tectonic setting and local seismicity

The study region (Fig. 1) is part of the collisional zone between the Adriatic and European tectonic plates. The convergence of these plates began no later than 84 Ma, with the onset of collision at ca. 35 Ma (Dewey et al., 1973; Schmid et al., 2004; Handy



et al., 2010) and culminating since ca. 24 Ma with north-northwestward indentation of the Adriatic continental lithosphere (Handy et al., 2015). Indentation is still ongoing today (Cheloni et al., 2014; Aldersons, 2004; Serpelloni et al., 2016; Sánchez et al., 2018).

Deformation in the target area (red box in Fig. 1) occurred within a corner of the Adriatic indenter, which is delimited to the north by the Periadriatic Fault (PAF) and to the west by the Guidicarie Fault (GF). The PAF is a late orogenic fault active

in Oligo–Miocene time, which was sinistrally offset by the GF in Miocene time. Just north of the northwestern corner of the indenter, where the GF and PAF meet, the Tauern Window (TW) exposes remnants of the European lower plate at the surface (Scharf et al., 2013; Schmid et al., 2013; Favaro et al., 2017; Rosenberg et al., 2018). Therefore, shortening and exhumation within the TW are attributed to faulting along the Adriatic indenter (Scharf et al., 2013; Favaro et al., 2017; Reiter et al., 2018). Counterclockwise rotation of the Adriatic microplate with respect to the Eurasian plate (Le Breton et al., 2017) is interpreted

to have induced seismic activity in the Southern Alps (Anderson and Jackson, 1987; Mantovani et al., 1996; Bressan et al., 2016).

The thrust front in the eastern Southern Alps (ESA) accommodates most of the Adria–Eurasia convergence with pronounced N–oriented present-day motion (Serpelloni et al., 2016; Sánchez et al., 2018). A N–oriented horizontal deformation of ∼2 mm/a is observed in the ESA front (northern part of the Venetian–Friuli Basin; Sánchez et al., 2018; Métois et al., 2015;

Cheloni et al., 2014) with highest recorded geodetic strain rate along the Venetian Front (within the Montello region, around 12.25°E/45.8°N) (Serpelloni et al., 2016). From the Venetian part of ESA toward the Po Basin (PoB), the motion vectors have a slight westward rotation with decreasing magnitudes. In contrast, a progressive eastward rotation of motion vectors is observed from the Venetian area toward the Pannonian Basin. The entire Alpine chain is still undergoing uplift, with a maximum in the inner Alps, e.g., along the border between Switzerland, Austria, and Italy (near 11°E/46.6°N) with values of ≥ 2 mm/a,

whereas uplift rates decrease toward the foreland (Sánchez et al., 2018).

Two sedimentary basins, the Molasse and the Po Basins (MoB and PoB in Fig. 1) dominate the shallow structure at the northern and southern borders of the target area. The MoB, running along the northern front of the Alps, thickens going across strike from the exposed Variscan basement (0 km) to the Alpine Front (5 km, e.g., see depth–to–basement contours in the map of Bigi et al., 1989). The Quaternary and Tertiary sediments of the PoB thicken from 0 km at its northern margin to about 6 km

in the south below the Apenninic Front (Bigi et al., 1989; Waldhauser et al., 2002).



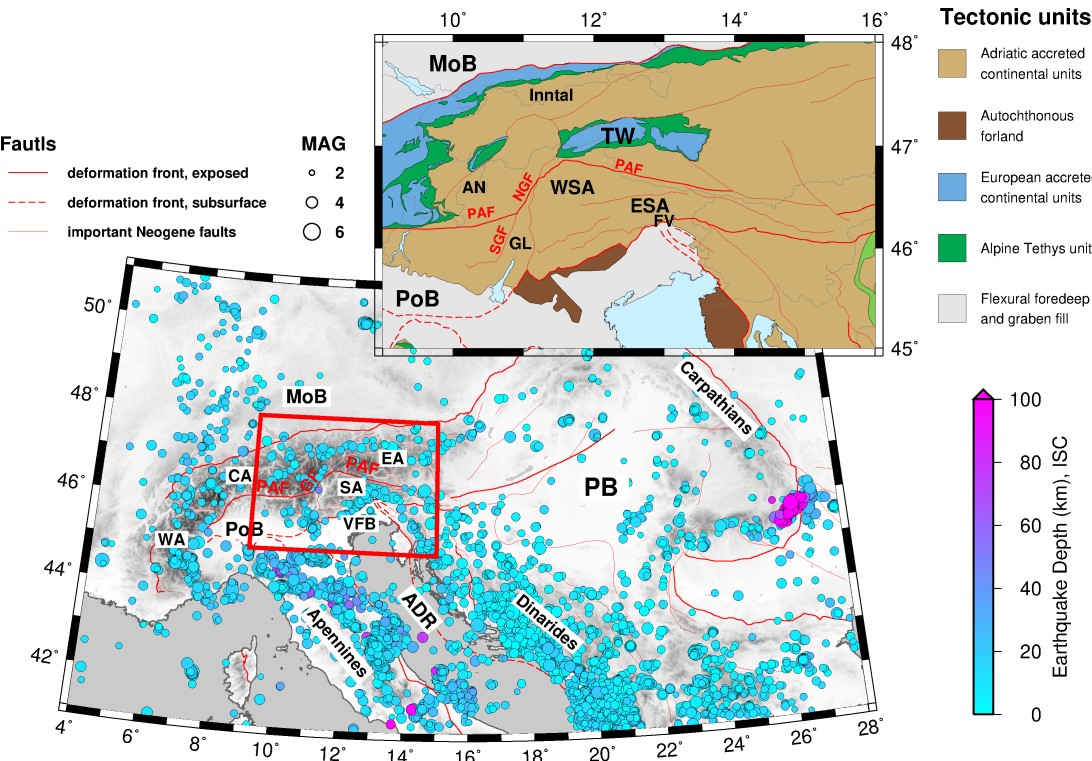

**Figure 1.** Geological map of the Alps showing major deformation fronts, faults, geographical subdivisions, tectonic units, and seismicity (seismic events between Jan 2010 and Dec 2016 with a magnitude larger than 2 from the International Seismological Centre, 2020). The tectonic units of the study area (red box) are shown on the upper right inset map. The maps are modified from Handy et al. (2010) and Schmid et al. (2004, 2008). Legend: **WA** - Western Alps, **CA** – Central Alps, **SA** – Southern Alps, **EA** – Eastern Alps, **ADR** - Adriatic plate, **PB** – Pannonian Basin, **VFB** - Venetian–Friuli Basin, **PoB** - Po Basin, **MoB** - Molasse Basin, **TW** - Tauern Window, **PAF** - Periadriatic Fault, **NGF** - Northern Giudicarie Fault, **SGF** - Southern Giudicarie Fault, **ESA** - Eastern Southern Alps, **WSA** - Western Southern Alps, **GL** - Giudicarie–Lessini region, **FV** - Friuli–Venetia region, **AN** - Austroalpine Nappes.

The seismicity in the study area and surrounding regions has been monitored routinely for many years with stations operated by different national and regional networks(**INGV**–National Institute of Geophysics and Volcanology and **OGS**–National Institute of Oceanography and Experimental Geophysics in Italy, **ZAMG**–Central Institution for Meteorology and Geodynamics in Austria, and **SED** Swiss Seismological Service in Switzerland. Besides routine catalogs, the seismicity has also been investigated in a few seismological studies through various time periods and regions (Galadini et al., 2005; Diehl, 2008; Viganò et al., 2013; Reiter et al., 2018, among others). Diehl (2008) compiled the local earthquake data from 14 seismic networks in the greater Alpine region from 1996 to 2007 and created a uniform and consistent relocated event catalog. The SHARE European Earthquake Catalog (Grünthal et al., 2013) comprised homogeneous earthquakes from 1000 to 2006.

These studies indicate that seismic activity is clustered in the Friuli–Venetia (FV), Giudicarie–Lessini (GL), and the Inntal regions (Fig. 1). The FV region is located along the thrust–and–fold belt marking the active Adria–Europe plate boundary and





was stuck by several earthquakes of $M_L \geq 6$. The energy was released on a system of E–W south–verging thrusts (some of which are blind or unknown) as well as on backthrusts and oblique–slip faults (Nussbaum, 2000; Galadini et al., 2005; Slejko, 2018; Romano et al., 2019). The GL region is located in the deformation zone along the western margin of Adriatic indentation with two major fault and fold systems characterized by high seismic activity: the Giudicarie Belt and the Schio–Vicenza strike–slip fault (Viganò et al., 2015). In contrast, the area east of the Northern Guidicarie Fault (NGF) is obviously much less active, at least since 1994 (Viganò et al., 2015). However, this almost inactive region coincides with a paucity of permanent seismic stations and therefore, precise seismicity investigation using a denser network is required. One of the goals of this study is to correlate seismicity with the pattern of active and inactive faulting mapped at the surface.

## 3 Data acquisition and processing

### 3.1 Network data

For this study, we used mainly data of the SWATH–D network, which was temporarily deployed in a roughly rectangular region in northern Italy and southeastern Austria (black box in Fig. 2). The installation started in Summer 2017 and the network ran for almost two years (Heit et al., 2017 and Heit et al., submitted). The network consisted of 151 broadband seismometers with an average inter–station spacing of 15 km. A total of 78 stations equipped with Güralp (3ESPC) seismometer and Earth Data (EDR-210) recorder transmitted data in near real–time via the cellular network (on–line). However, 73 stations, with data accessed during station services (off–line), were equipped with two different kinds of seismometers and digitizers: 58 of them with Nanometrics (Trillium Compact) seismometers and DATA-CUBE3 (CUBE) recorders, and the rest with Güralp (3ESPC) seismometers and Earth Data (PR6-24) recorders. In Autumn 2018 the network was complemented with an additional 10 on–line stations in the northeastern edge by LMU–Munich. All SWATH–D seismic stations recorded the data continuously with 100 samples per second (sps).

In order to enlarge the stations azimuthal coverage for earthquakes occurring in the periphery of the SWATH–D framework, we additionally selected 112 stations of the larger scale AlpArray Seismic Network (AASN; Hetényi et al., 2018) to include their waveform data in the analysis.




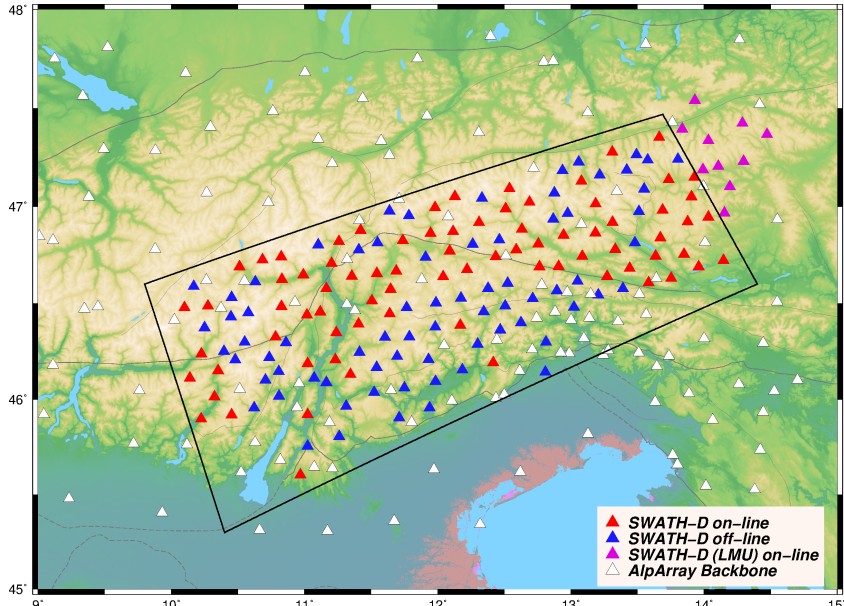

**Figure 2.** Distribution of seismic stations of the SWATH–D Network (red, blue, purple) and selected stations of the AASN (white). The black box indicates the periphery of the SWATH–D network. The faults are from Schmid et al. (2004, 2008) and Handy et al. (2010).

## 3.2 Seismic events and arrival–time picks

The national seismological agencies of Italy (INGV and OGS), Austria (ZAMG), and Switzerland (SED) provide comprehensive earthquake catalogs (minimum magnitude -0.8 by ZAMG) in our study region. Therefore, the process of event detection was skipped and an integrated earthquake list from the national agencies, after removal of common events, formed a proper event list to start with. For this study, we selected events with $M_L \geq 1$ in the time frame between September 2017 and the end of 2018 (16 months) summing up in 2,639 local events, which were observed by 290 selected permanent and temporary stations.

Considering this large number of events and stations, we applied a modified version of the automated multi–stage workflow from Sippl et al. (2013) to the waveform data for picking the arrival–times of P and S waves. This workflow was originally implemented for producing a complete catalog of well–located earthquakes and reliable arrival–times using continuous passive seismic data in the Pamir–Hindu Kush region (Sippl et al., 2013). We, subsequently, assessed the performance of the picking algorithms by quality checking of time and phase–type uncertainties of the picks (appendix A).

We noticed that the number of mispicks and also ignored picks was quite large (Sect. A2). Therefore, a visual/manual inspection on the selected events (384 events with azimuthal gap less than 200° and RMS less than 1 s) was subsequently done to remove or modify obvious mispicks (mainly at large epicentral distances) and also repick the missed arrivals (mostly S–picks of very close stations). This manual/visual inspection was performed on individual station data as well as on seismogram sections of all observations of larger events to check for the correct phase identification (see also Sect. 3.3). Furthermore,

an inspection for identifying potential quarry blasts (and other anthropogenic sources) was simultaneously done. Criteria for potential quarry blasts were: 1) relatively small number of S–picks, 2) relatively small S/P amplitude ratios (see, e.g., Walter



et al., 2018), and 3) large surface waves (observed dispersive waveform characteristics). Based on our assessment, we classified the events into 344 earthquakes, 15 potential blasts, and 25 unclear events.

A Wadati diagram (Wadati, 1933; Kisslinger and Engdahl, 1973; Diehl, 2008) was used to identify and correct obvious
outliers. As seen in Fig. 3, the number of outliers after correction is quite low. The $V_P/V_S$ ratio is 1.72 (calculated by linear least squares) in our dataset.

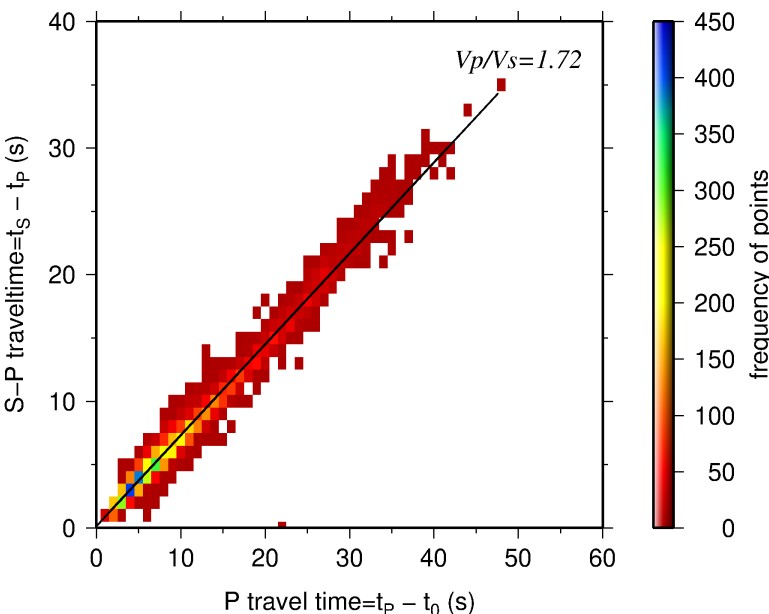

**Figure 3.** Modified Wadati diagram based on P– and S–picks after visual/manual analysis (the color indicates number of picks; please see the color palette table on the right). The origin–times ($t_0$) of the earthquakes are provided by the single–event localization using HYPO71 hypocenter determination program (Lee and Lahr, 1975). The black line shows the fit to the data by linear least squares and its slope indicates the $V_P/V_S$ ratio of 1.72.

### 3.3 Seismic phase identification

The direct Pg, Moho–reflected PmP, and Moho–refracted Pn phases arrive closely spaced in time, especially in the triplication zone, potentially leading to phase–type misidentification. The PmP is always a secondary arrival, but its amplitude can dominate
the first arrival and thus easily being mispicked. On the other hand, for epicentral distances larger than the triplication zone, the Pn amplitude can be so weak, that the Pg or PmP can be wrongly identified as the first arrival. In the local earthquake case, and particularly for regions with varying Moho topography and significant lateral variations in the crustal structure, something to be expected for the Alps, the phase–type identification is even more challenging (Diehl et al., 2009b). The inversion is based on the first arrival–times and therefore checking the phase–type is extremely important.
Figure 4 shows the velocity reduced travel–time of all the first arrivals vs. epicentral distance in order to have a comprehensive assessment of the phase–type identification. For comparison, the synthetic Pg, PmP, and Pn travel–times are calculated independently for each earthquake (based on a velocity model consisting of a homogeneous crustal layer with 35 km thickness





and V$_P$ of 5.8 km s$^{-1}$ over a half-space with V$_P$ 7.8 km s$^{-1}$ according to the ak135 model of Kennett et al., 1995). The whole

range of Pg, PmP, and Pn synthetic travel–time curves (caused by different earthquake depths), hereafter called travel–time

interval, are shown with their zone confined between two green, red, and blue dashed lines, respectively. The picks have a wide

scatter in time, however, depending on the epicentral distance they are still close to the corresponding travel–time interval. Most

importantly, high–energetic PmP waves are not picked as first arrivals. The number of outliers, i.e., picks that are located far

off the travel–time intervals, is not significant compared to the total number of picks. The remaining scatter could be associated

with the 3–D structure not represented by the 1–D model.

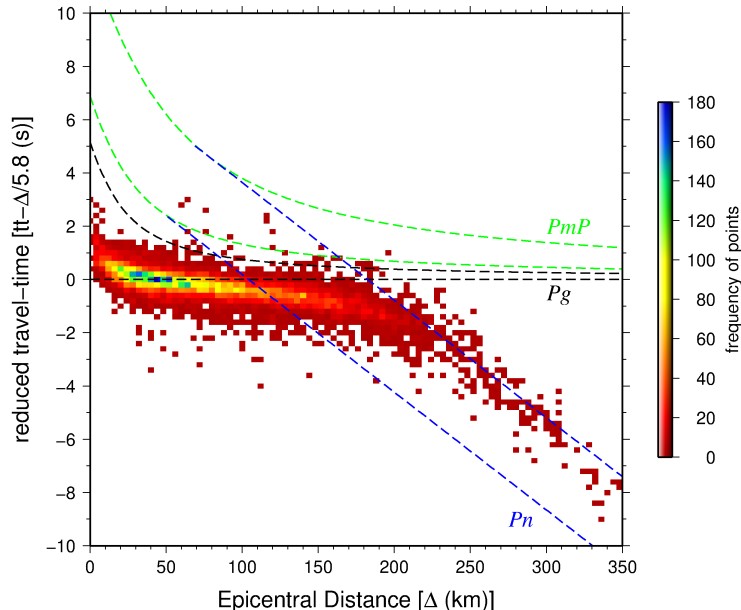

**Figure 4.** The velocity reduced travel–time values vs. epicentral distance for all the P–picks of all local earthquakes (the color indicates
number of picks; please see the color palette table). The zones between two green, red and blue dashed lines indicate, respectively, the
synthetic Pg, PmP, and Pn travel–time interval (see text) for earthquakes with various depths. Synthetic travel–time curves are calculated
using a velocity model with one crustal layer over half space. Note that the vertical axis is based on velocity reduced time. The hypocentral
solutions are provided by the single–event localization using HYPO71.

# 4    Method

## 4.1    Probabilistic Bayesian inversion

Considering that hypocenter locations and velocity model are inherently linked to each other (coupled hypocenter–velocity

problem, e.g., Kissling, 1988; Thurber, 1992; Kissling et al., 1994) especially in the local earthquake case, it is indicated

to simultaneously invert for hypocenters and velocity structure (and/or station corrections). In this study, we use a Bayesian

approach (Bayes, 1763), which has been applied in a number of geophysical studies (Tarantola et al., 1982; Duijndam, 1988a,b;

Mosegaard and Tarantola, 1995; Gallagher et al., 2009; Bodin et al., 2012a,b; Ryberg and Haberland, 2018). This approach





has been implemented recently by Ryberg and Haberland (2019) for the joint inversion of hypocenters, 1–D velocity structure and station–corrections in particular for the local earthquake case. While with classical inversion techniques a discrete best-fitting model **m** (i.e., locations, velocity model, etc.) is derived, in the Bayesian inference all the parameters of the model **m**

are represented probabilistically. For the theoretical background the reader is referred to Bodin and Sambridge (2009) and Bodin et al. (2012a,b). We define a uniform and wide range of values for each individual model parameter (before knowing the observed data; prior) and after performing the inversion for random combinations of the parameters, we derive a probability distribution individually for each model parameter (after combining the prior information with the observed data; posterior). Thus, a large suite of models is generated, all of them fitting the travel–time observations. The choice of a uniform and

extensively wide range of model parameters (Table 1) guarantees that the final model is dominated by the data rather than by the prior information (i.e., starting model, choice of parameters).

**Table 1.** Prior distribution (model space), starting model, and width of Gaussian distribution for each model parameter.

| Model parameter | Lower boundary | Upper boundary | Starting model | Gaussian Width |
|---|---|---|---|---|
| Earthquake epicenter (x,y) | -300 km | +300 km | random and uniform | 2 km |
| Earthquake depth (z) | 0 km | 200 km | random and uniform | 2 km |
| $V_P$ | 2 km s$^{-1}$ | 12 km s$^{-1}$ | normal with mean=6 and sigma=0.5 km s$^{-1}$ | 0.05 km s$^{-1}$ |
| $V_P/V_S$ | 1 | 2.5 | normal with mean=$\sqrt{3}$ and sigma=0.2 | 0.05 |
| Layer depth (h) | 1 | 200 | random and uniform | 10 km |
| Number of layers | 1 | 200 | Normal with mean=5 and sigma=3 | - |
| Noise $\sigma_P$ and $\sigma_S$ | 0.001 s | 10 s | 1 s | 0.01 s |
| Station correction $\tau^P$ and $\tau^S$ | -5 s | 5 s | 0 s | 0.05 s |

## 4.2 Model parametrization and forward problem

Hypocenters are described by spatial $(x_0, y_0, z_0)$ and temporal $(t_0)$ coordinates and the 1–D velocity structure is described by a variable number of horizontal layers with individual values of $V_P$ and $V_P/V_S$. Moreover, the model **m** comprises sta-

tion–corrections for P and S waves ($\tau^P$ and $\tau^S$), which account for deviations of the 1–D model from the real 3–D velocity distribution and local shallow velocity structures beneath the stations. The inversion also derives the quality of the data expressed as noise levels for P– and S–picks ($\sigma_P$ and $\sigma_S$), which are also unknowns in the inversion. The noise includes actual picking errors, systematic errors of observations/measurements, and any approximation or errors of forward travel–time calculation (see below) and is assumed to be normally distributed and uncorrelated. Therefore, the complete model to be inverted

for is defined as:

$$\mathbf{m} = [K, V_{P_i}, V_P/V_{S_i}, h_i, x_j, y_j, z_j, t_j, \tau^P{}_k, \tau^S{}_k, \sigma_P, \sigma_S] \tag{1}$$





where K is the number of layers, h is layer depth, and i, j, and k refer to layer, earthquake, and station index respectively. It should be noticed that the number of layers is also unknown. In other words, the model space dimension is not fixed in advance and hence the posterior is a transdimensional function.

In order to sample models from the prior probability distribution, we apply a transdimensional, hierarchical Markov chain Monte Carlo (McMC) method (Green, 1995). The McMC creates a huge number of different models **m** by random selection of model parameters (see Equation 1 and Sect. 4.3). Accordingly, to compute travel–times the forward problem has to be solved for a very large number of times. We use the 2–D finite differences (FD) Eikonal solver (Podvin and Lecomte, 1991) for the forward calculation. This algorithm requires a regular mesh. Therefore, the irregular velocity model (i.e., layers) is converted to a fine and uniform mesh based on Voronoi cells (constant velocity layers)(Aurenhammer and Klein, 2000), which sets the velocity at each mesh point to the value of the nearest point from the irregular model. The Voronoi mesh, in our case, has a cell spacing of 1 km vertically and horizontally.

Once the travel–times are calculated, a misfit function particularly for each model is defined as the summed differences between the observed (**d**) and calculated travel–times. The misfit function is then used to build the Gaussian likelihood function and the posterior values of the proposed model (for detailed information one could refer to Ryberg and Haberland, 2019).

### 4.3 Markov chain Monte Carlo (McMC) algorithm

Given the Bayesian approach described above, the McMC algorithm generates an ensemble of models with parameters within the prior distribution. We mainly follow the hierarchical, transdimensional procedure proposed by Ryberg and Haberland (2018, 2019) that supports the calculation of both model parameters and model dimensionality. The evolution of a model along the Markov chain consists of four main steps: (1) Choose a random initial model **m** (Table 1). (2) Generate a new model from the prior distribution by perturbing the current model parameters (changing one of the velocity parameters of a random node, shifting the position of a random earthquake, adding a new layer, removing a random layer, altering one of the noise parameters, changing a randomly selected station-correction). The changes in the $V_P$, $V_P/V_S$, cell position (i.e., layer), and noise levels must be according to Gaussian probability distribution centered at the current value. The values of the new model are randomly selected within very wide bounds (Table 1) and thus do not truncate the posterior probability distribution. (3) The forward calculation (Sect. 4.2) is then performed on the new model. The newly estimated travel-time data are compared with the observed data **d** and then the misfit function, data likelihood, and the posterior probability are determined. (4) The newly proposed model is accepted or rejected based on the criteria of Bodin and Sambridge (2009). If the new model is rejected, then the current/old model is retained by reiterating step 2. However, there is still a probability of acceptance even if the fit of the new model is worse than that of the old model. If the model is accepted (i.e., when it is better than the previous model) then it acts as a starting model for another iteration (step 1).

By reiteration of steps 1 to 4, a chain of models is produced, which is in fact the Markov chain. This chain is continued until the misfit is not significantly decreasing any more (burn–in phase). Thereafter, a stationary model space sampling is achieved. If these sequences are repeated long enough, a chain provides an approximation of the posterior distribution for the model parameters. To accelerate the model space sampling, up to 1000 separate and independent chains are investigated in parallel.





The main idea of the McMC method is to keep prior assumptions of the model parameters (conventionally used as initial values) at a minimum and thus minimizing unwanted artificial effects on the final model. Therefore, the McMC method only uses the travel times and does not depend on initial hypocenters, origin times, velocity models, or even the model parametrization (e.g., grid node spacing).

## 5  Uncertainty estimates using synthetic tests

As a variable Moho topography and complex lithosphere structure are expected for our study region, the 1–D model derived by and used in our inversion can only be a rough abstraction of the real conditions. Nevertheless, this simplification could potentially have influence on earthquake location correctness and accuracy. To assess the performance of our inversion strategy in this respect, using a 3–D velocity model and earthquakes and station distribution of the real data, we calculated the synthetic
travel–times, added synthetic noise and finally inverted them in the same way as the real data (see below). Comparing input event locations (synthetic) and the inverted (output) ones allows us to study the location accuracy and potential systematic errors related to the use of a 1–D model, which we can also expect for the derived real hypocenters.

The background 3–D velocity model is designed based on the Moho topography of the European and Adriatic plates from Spada et al. (2013). The $V_P$ starts with 6 km s$^{-1}$ at -5 km, gradually increases down to the Moho depth, and reaches 7 km s$^{-1}$.
At Moho depth, the velocity increases from 7 km s$^{-1}$ to 8 km s$^{-1}$. The velocity below the Moho increases again gradually until it reaches a velocity of 8.05 km s$^{-1}$ at 90 km depth (Kennett et al., 1995; Christensen and Mooney, 1995). The Moho depth in the target area changes from 19 km in the PoB to more than 55 km along the Adria–Europe plate boundary (Spada et al., 2013). We considered the model's surface at -5 km to avoid wave propagation through the air.

In order to simulate realistic 3–D effects in the context of expected crustal structure, we superimposed shallow high– and
low–velocity anomalies onto the background 3–D model presented above. We assumed that the hard rocks of the European basement exhumed in the TW have higher velocities than the surrounding Austroalpine nappes and, thus, form a high-velocity anomaly (+5 %) within a rough outline of the TW and from the surface to the depth of 10 km (for more information on the TW see, e.g., Bertrand et al., 2015). Furthermore, we designed lower velocity anomalies for the PoB and MoB sedimentary basins (the coarse outline of the basins are taken from Waldhauser et al., 2002). An average $V_P$ of 4.35 km s$^{-1}$ is considered for the
entire MoB from the model's surface down to 2 km depth. For the PoB, a layer of 6 km thickness starting from the model's surface and with $V_P$ of 2 km s$^{-1}$ less than the background model is constructed. The shallow anomalies are then inserted in the background model. Taking the shallow anomalies into account not only makes the model more realistic but also allows us to check whether the station–corrections reflect travel–time deviations due to velocity variations in the shallow subsurface. Figure 5 displays the 3–D synthetic $V_P$ model from different views and also the 1–D model at two different viewpoints. The
$V_P$/$V_S$ ratio in the entire model is set to $\sqrt{3}$.

The P and S arrival–times are calculated using the 3–D FD Eikonal solver (Podvin and Lecomte, 1991; Tryggvason and Bergman, 2006). We generated an FD grid with an increment of 1 km horizontally and vertically, resulting in a grid dimension of 601×321×96 with a total number of 18,520,416 nodes. For simulating a realistic travel–time dataset, we adopted the





geometry of stations and hypocenters of the real data. Using 273 stations and 344 earthquakes, altogether 12,534 P and 7,258

S synthetic picks result from the forward calculation. Thereafter, according to the manual arrival–time uncertainties in Table A1, we added random noise to the arrival–times, and this dataset was then used for the simultaneous inversion by McMC.

**(a)** **(b)**

**(c)** **(d)**

**Figure 5.** 3–D synthetic V$_P$ velocity model. The V$_P$/V$_S$ ratio was fixed to $\sqrt{3}$ in the entire model. **(a)** and **(b)** Contours of the Moho and shallow anomalies of the PoB, MoB, and TW from different view directions, **(c)** modified Moho depth map based on Spada et al. (2013), **(d)** 1–D velocity model of 2 different locations with shallower and deeper Moho, retrieved from the 3–D synthetic velocity model (the locations are shown with red and black crosses in **(a)** and **(c)**, respectively). The faults are same as Fig. 2.

To explore the model parameters, 1000 Markov chains each with 1000 random initial models are used for simultaneous inversion of McMC. Following the strategy in Ryberg and Haberland (2019), we derived ∼30,000 final models from the Markov





chains to define model parameters based on the average $\mu$ and standard deviation $\sigma$. For the earthquake epicenters (x and y)

and station–corrections ($\tau^P$ and $\tau^S$) the classical average of the Gaussian distribution is used. However, the depth distribution of shallow earthquakes is truncated by the upper model boundary. In order to accommodate this, we used an algorithm based on truncated Gaussian distributions (Ryberg and Haberland, 2019) to derive true depth averages and uncertainties. The $V_P$ and $V_P/V_S$ values are defined based on the modified average (Ryberg and Haberland, 2018) and standard deviation.

Figures 6 and 7 represent the recovery of the earthquakes. The earthquakes are determined with an average uncertainty

of 240 m in longitude, 270 m in latitude, 1.24 km in depth, and 0.38 s in origin–time (Fig. 6b and error bars in Fig. 6a). Figure 6a shows that the epicenters are recovered very well (the blue dots are located on top of red circles). Further assessment of the earthquakes recovery (Fig. 7a) demonstrates that the misfit between synthetic and recovered hypocenters is close to zero with a standard deviation of ∼400 m in epicenter and 1.31 km in depth and 0.12 s in origin–time. The recovered noise levels, which are representative of the unresolved part of the travel–times, are calculated for individual pick types and quality

classes separately (Fig. 7b). They are also close to the random noise, which was added to the synthetic arrival–times (manual arrival–time uncertainties in Table A1), however, slightly higher. This deviation could be explained by the forward modeling errors and the fact that the inversion derives a 1–D velocity model from data of a 3–D input model.



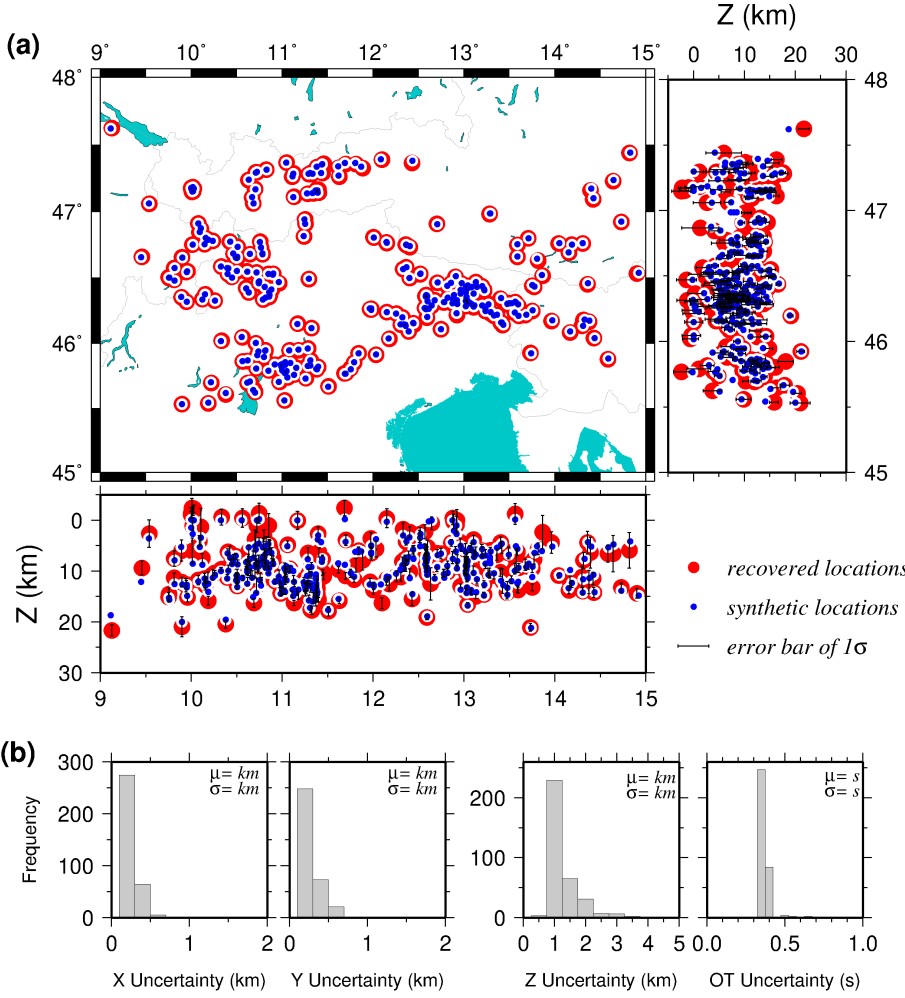

**Figure 6. (a)** Recovery of the hypocenters after synthetic test. Red circles are recovered earthquakes with $1\sigma$ error bar (error bars of latitude and longitude are invisible at this scale) and blue dots are the synthetic locations. **(b)** Histograms of hypocenter uncertainties after McMC simultaneous inversion for synthetic dataset.

Figures 8a shows the derived $V_P$ and $V_P/V_S$ models as heat–maps showing the posterior distribution of all the models. In addition, it shows the average value and standard deviation of all the inferred models and also a reference model with maximum likelihood, i.e., maximum posterior probability (similar to Ryberg and Haberland, 2019). As seen, no clear, single Moho velocity jump is recovered, but there are rather slight velocity jumps at different depths from 30 to 50. This reflects the variable Moho topography, which cannot be modeled in the 1–D velocity model. Because of a dense ray–coverage from the surface down to a depth of $\sim$20 km, the $V_P$ uncertainty is almost zero in this depth range. Below 20 km, the uncertainty varies between 0.1 and 0.66 km s$^{-1}$. The $V_P$ model is resolved until 63 km depth corresponding to the maximum ray penetration. However, as a 1–D model cannot be representative of the 3–D structure, especially in a region with expected fluctuating Moho


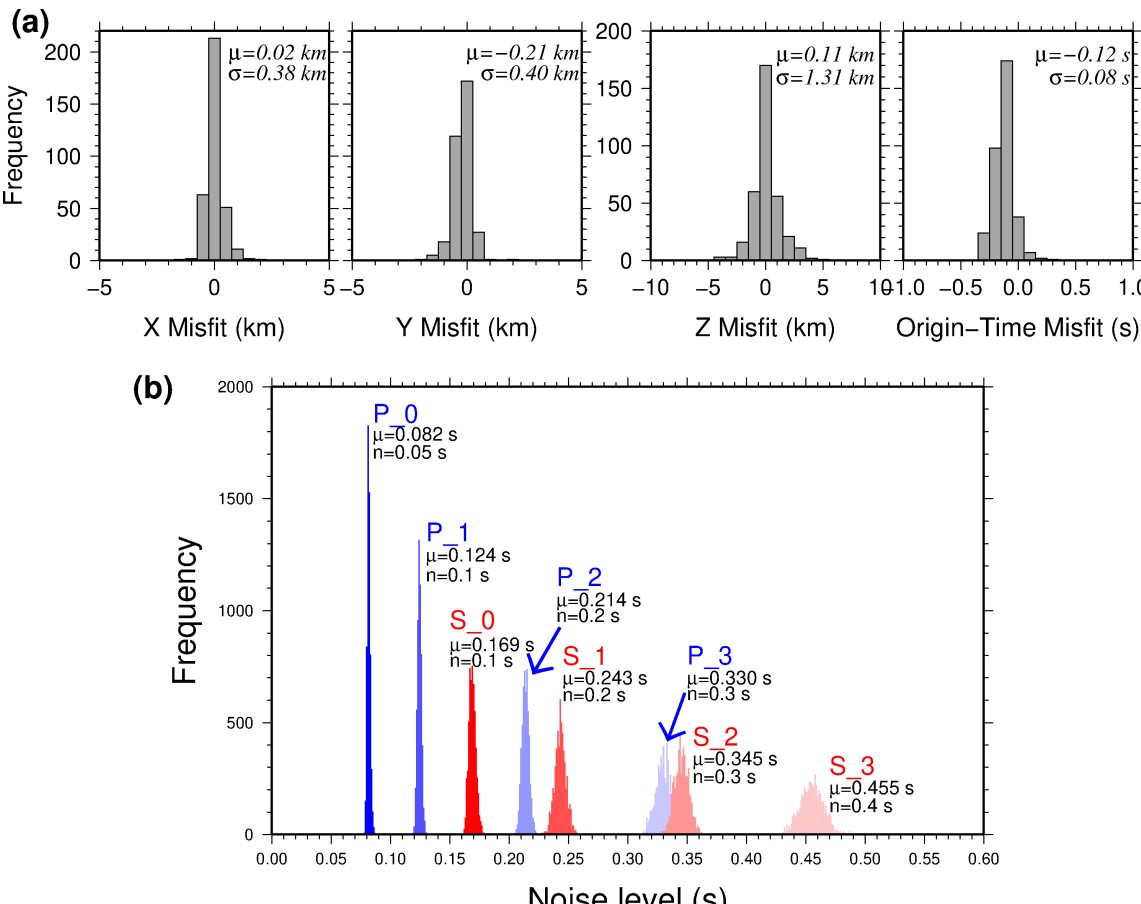

**Figure 7. (a)** Histograms of misfit between recovered hypocentres and original ones. The hypocenters are recovered very well with average deviations of 20 m (±380 m) in longitude, 210 m (±400 m) in latitude, 110 km (±1.131 km) in depth and 0.12 s (±0.08 s) in origin–time **(b)** The recovered noise levels after McMC for individual pick types and quality classes are slightly higher than the random noise, which was added to the synthetic travel–times data (n).

and complex crustal structure, a geologically meaningful interpretation of the derived $V_P$ model is hardly possible. The $V_P/V_S$ ratio was fixed to the square root of 3 for the whole region in the forward modeling and the same value is recovered down to 35 km depth with small uncertainty (less than 0.022). The uncertainty of $V_P/V_S$ increases below 35 km depth to a maximum value of 0.046.

The stations–corrections (Fig. 8b) correspond, to large extent, to the shallow anomalies of the 3–D synthetic model (Fig. 5). The shallow high–velocity anomaly in the TW is expressed by negative corrections (large circles in Fig. 8b) corresponding to earlier arrivals, while the shallow low–velocity anomalies of the PoB and MoB correspond to positive corrections (large crosses in Fig. 8b) reflecting later arrivals.
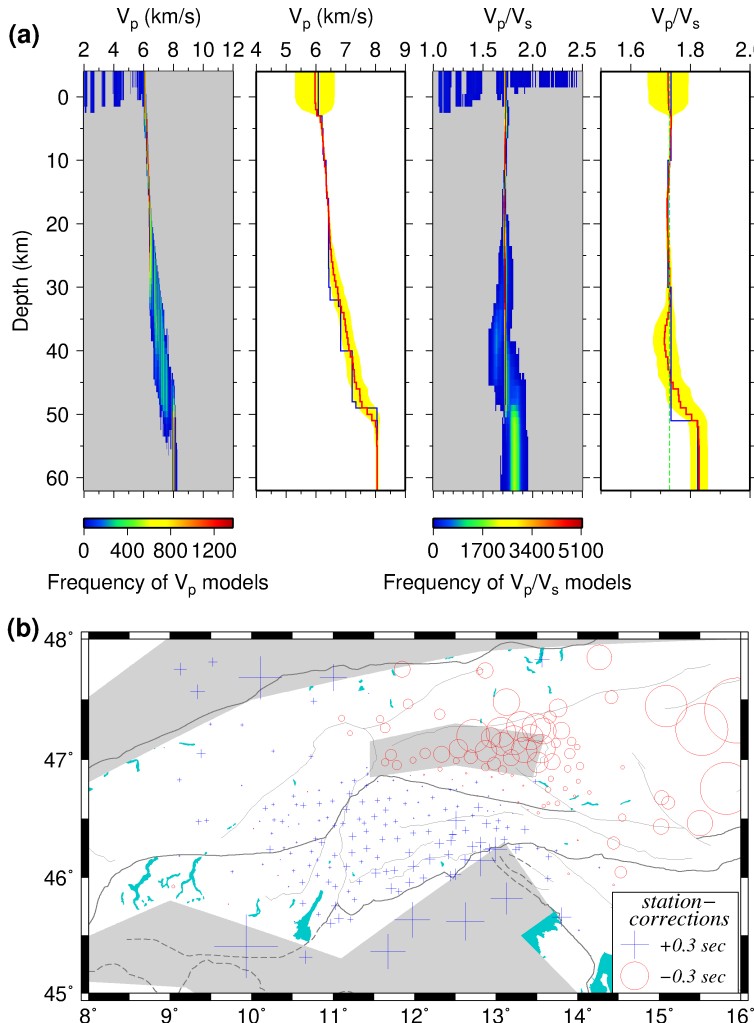

**Figure 8. (a)** Recovery of the $V_P$ and $V_P/V_S$ models after synthetic test. The results are shown with heat–maps (probability histogram versus depth; warm and cold colors correspond to higher and lower probabilities, respectively) with gray color showing zones with no model achievement. The modified average model (red solid line), most probable model (blue solid line), and corresponding uncertainty of $1\sigma$ (gray zone) for $V_P$ and $V_P/V_S$ models are shown as well. The $V_P/V_S$ ratio of $\sqrt{3}$ is displayed with the green dashed line. **(b)** Recovered station–corrections after McMC inversion. Blue crosses show positive values reflecting lower velocities and red circles display negative values indicating higher velocities than expected, respectively. Regions indicated by gray color have shallow high/low–velocity anomalies (see Fig. 5 for more information). Symbol size corresponds to correction amplitude. The faults are same as Fig. 2.

# 6 Results in discussion

In this section, the final results of the McMC inversion are presented. For the simultaneous inversion of the hypocenters, the 1–D velocity model, and the station–corrections, we used the travel–time dataset of the earthquakes, which have at least 10



P–picks and 5 S–picks and an azimuthal gap less than 180° (301 earthquakes with 11,186 P–picks and 6,560 S–picks). In the simultaneous inversion, for the first 300,000 models of the Markov chains, only the hypocenters are allowed to be changed. Thereafter, additionally, $V_P$, $V_P/V_S$ ratio, station–corrections, and noise levels are permitted to change as well. After inversion

of ∼400,000 models by the Markov chains, the model space reached stationarity (burn–in phase) and the remaining models had stationary RMS misfits around 0.36 s (Fig. 9).

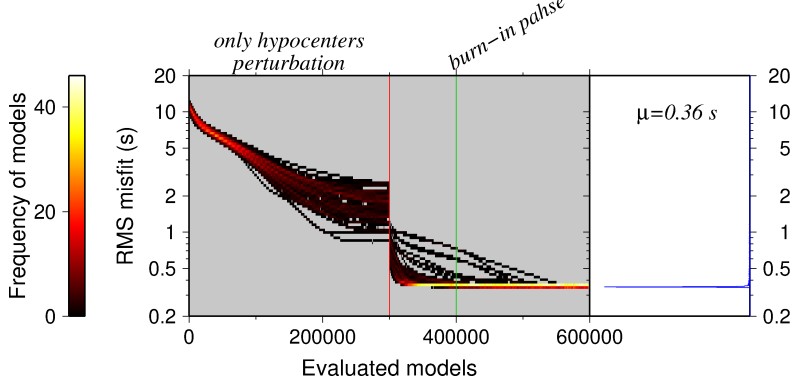

**Figure 9.** Left panel: RMS misfit along 1000 Markov chains during the inversion evolution (color indicates number of models; please see color palette table on the left). Before the red line (300,000 models) only the hypocenters are perturbed and beyond that the $V_P$, $V_P/V_S$ ratio, station–corrections, and noise levels are allowed to change as well. The green line indicates the burn–in phase after ∼400,000 models. Right panel: the histograms of RMS misfit for all the models after burn–in phase. The best–fitting models are characterized by an average 0.36 s RMS misfit.

### 6.1   1–D velocity model and station–corrections

It turns out that the algorithm needs roughly 5 layers to best fit the data (Fig. 10a). Similar to the synthetic test (Fig. 8), the final velocity models are shown with the average of all remaining models and the model with maximum posterior probability.

In general, for both $V_P$ and $V_P/V_S$ models (Fig. 10b) the average value (red line) varies gently, whereas the model with maximum posterior probability (blue line) is rather coarse and contains discontinuities.

The derived $V_P$ model is well–resolved down to a depth of 63 km with an uncertainty (1$\sigma$) of 0.01 to 0.45 km s$^{-1}$ (gray zone). If we consider the modified average model as the reference model, $V_P$ starts with a rather high value of 5.94 km s$^{-1}$ at the surface down to a depth of around 20 km. Thereupon, it gradually increases until it reaches ∼6.8 km s$^{-1}$ at around 35

km depth. Moreover, at around 56 km depth the velocity jumps again to 8.2 km s$^{-1}$. These two step–like velocity jumps most likely reflect the transition from crustal to upper mantle velocities, which show a large depth–variability throughout the region (see, e.g., Fig. 5). The $V_P$ values are in good general correspondence (given the discrete parameterization) to the model by Diehl et al. (2009b) being representative for the large Alpine region, however, mantle velocities are reached deeper than in the model by Diehl et al. (2009b).

The $V_P/V_S$ model starts with high values at the surface (to ∼5 km depth), shows reduced values of around 1.70 down to 30 km depth before reaching 1.77 at greater depths (with uncertainty between 0 and 0.05). This was basically expected from the





Wadati diagram (Fig. 3) and is in agreement with values previously derived, e.g., by Viganò et al. (2013). Both $V_P$ and $V_P$/$V_S$ models are available in the supplementary material S1.

The station–corrections derived from the McMC simultaneous inversion potentially indicate local, shallow 3–D velocity
anomalies in the subsurface, which cannot be accounted for by the 1–D model. The McMC inversion assumes that P and S station–corrections have an average of zero. Negative corrections indicate earlier wave arrival and thus higher velocities in the (shallow) subsurface, whereas, positive correction implies delayed arrival indicative of lower velocities.

The pattern of corrections (Fig. 10-c) shows coherent negative correction associated with the EA, ESA, and CA. The large negative values in the eastern part (east of 15° E) might be related to proximity to the edge of the network and thus be dominated
by mantle phases (faster arrivals). Surprisingly, in between the negative values in the Alpine Chain, a pattern of slight positive corrections in the WSA is observed. Besides, extreme positive corrections are seen in the PoB and MoB as it is expected for sedimentary basins. The pattern of stations in the ESA and a few stations in the WSA and CA agree very well with results by Diehl et al. (2009b).

A detailed interpretation of corrections pattern is highly ambiguous because it contains an overlay site effect and/or other
3–D variations such as Moho topography. It proved to be useful for accurately localizing earthquakes, for subsequent 3–D inversion (i.e., local earthquake tomography), and to show the consistency of phase data through inspection of the general pattern.



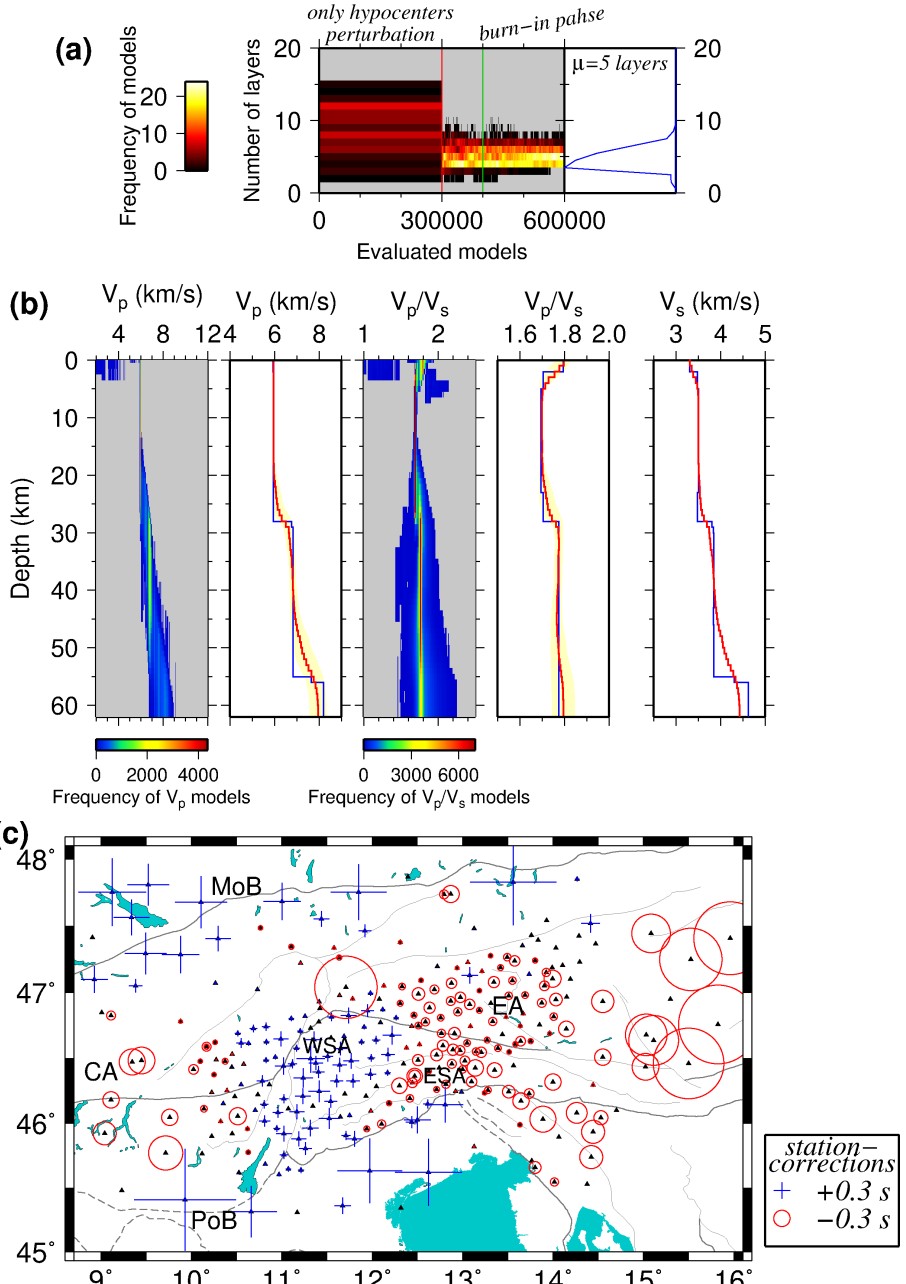

**Figure 10. (a)** Left panel: probability of number of layers of the random models introduced by Markov chains during the evolution (color indicates number of models; please see color palette table on the left). Right panel: histogram of number of layers of the models after burn–in phase. **(b)** $V_P$, $V_P/V_S$, and $V_S$ models after McMC simultaneous inversion. Figure characteristics are similar to Fig. 8. The velocity models are well recovered down to ∼63 km depth. **(c)** P–wave station–corrections corresponding to the 1–D velocity model in **b**. Negative corrections (red circles) indicate earlier arrivals (indicative for higher velocities in the shallow subsurface) and positive corrections (blue crosses) indicate delayed arrivals (representative for lower velocities underneath the station). The faults are same as in Fig. 2. Legend: **CA** - Central Alpine; **EA** - Eastern Alpine; **ESA** - Eastern Southern Alps; **WSA** - Western Southern Alps; **MoB** - Molasse Basin; **PoB** - Po Basin.

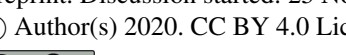


## 6.2 Estimation of hypocenters accuracy based on relocation of quarry blasts

To validate the localization procedure, the detected 15 quarry blasts (based on manual/visual inspections; see Sect. 3.2), which
were relocated independently by McMC using the previously derived 1–D velocity model and station–corrections from the
simultaneous inversion (Sect. 6.1). Fig. 11 focuses on the blasts distribution associated with two quarries close to the villages
of Albiano (Italy) and Gummern (Villach, Austria). After the relocation of the blasts, we see that the epicenters are within the
quarry area and the depths are in the range of the quarry topography (considering the average and the uncertainty of $1\sigma$), some
of them are offset by a maximum of hundreds of meters. This indicates that, although the number of picks (especially S–picks)
is generally lower for blasts, the McMC routine provides high precision hypocentral solutions. We expect that the accuracy of
earthquake hypocenters is even better because they have usually a higher number of S–observations.

The hypocentral solution of the blasts, in comparison with those obtained by INGV/ZAMG, have an average difference of
$\sim$160 m in longitude, $\sim$1 km in latitude, and $\sim$4.6 km in depth, so the depths are considerably better resolved and recovered
in our study, due to the availability of the denser SWATH–D network.

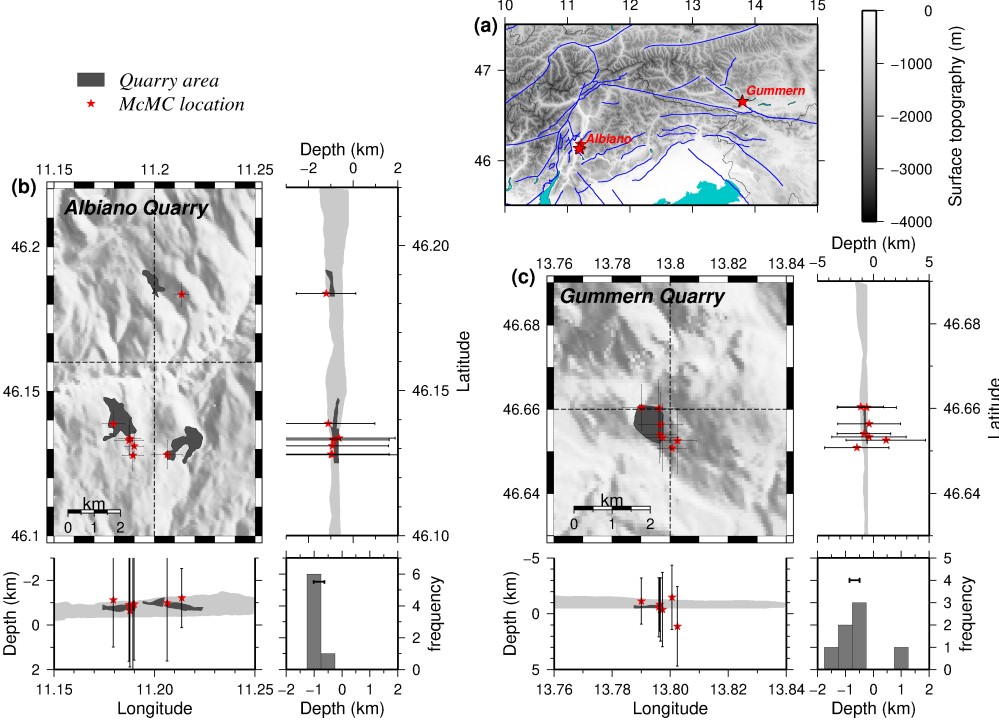

**Figure 11. (a)** Map view showing the location of quarry blasts relocated using single–event mode of McMC, with solutions of the 1–D velocity model and station–corrections from the simultaneous inversion of earthquakes. **(b)** and **(c)** Epicenter and depth distribution of the blasts associated to two quarries close to Albiano in Italy and Gummern in Austria, respectively. The light gray band in the cross sections shows the surface elevation variation within the map–view boundary. The dark gray zones show the location and and surface elevations of the quarries. The depth histogram is also shown for each quarry (the bar in the depth histograms displays the surface elevation within the quarry area). Please note that elevations above the see level are shown with negative values.





## 6.3 Seismicity distribution

The distribution of seismicity in the Southern and Eastern Alps is shown in map view and cross sections in Figs 12a and 13. This includes the same 301 events used for the simultaneous inversion (Sect. 6.1) as well as 43 additional earthquakes with slightly fewer picks for a total of 344 earthquakes, 12,534 P–picks, and 7,265 S–picks. For this relocation, we used the modified average $V_P$ and $V_P/V_S$ models as well as the corresponding station–corrections and noise levels resulting from the simultaneous McMC inversion. The hypocentral uncertainties ($1\sigma$) based on statistical analyses of the results of the McMC inversion are 1.73 km in depth and ~500 m for the epicenters (Fig. 12b). This is in agreement with the synthetic tests (Sect. 5) and the relocation of the quarry blasts (Sect. 6.2). The list of earthquakes is available in the supplementary material S2. The average differences between the epicentral positions derived in this study and those provided by the agencies are 2.3 km, with maximum differences of 11.5 km. The differences are even larger for depth (average 2.9 km and maximum 12.5 km).

The seismicity is clustered in the same areas as in previous seismic studies of the region (e.g., Reiter et al., 2018). Most activity is seen within the orogenic retro–wedge in the FV region, with somewhat lesser activity in the SW part of the Giudicarie Belt (GL region). Further activity is located in the Austroalpine Nappes north of the PAF and the region around Innsbruck (upper and lower Inntal and the Stubai Alps). The earthquakes reach a maximum depth of ~20 km (Fig. 12a), most of them occurring in the depth range of 5 to 15 km. Regions of little or no seismicity are observed at the northeastern corner of the eastern Adriatic or Dolomites Indenter, southeast of the NGF (e.g., Reiter et al., 2018), and north of the PAF (Fig. 12a).

The overall pattern of seismicity reflects the head–on convergence of the Adriatic Indenter with the Alpine orogenic crust, accommodated along thrust faults and folds in the FV region and segmented in the western part of the ESA by strike–slip faults of the Giudicarie Belt.



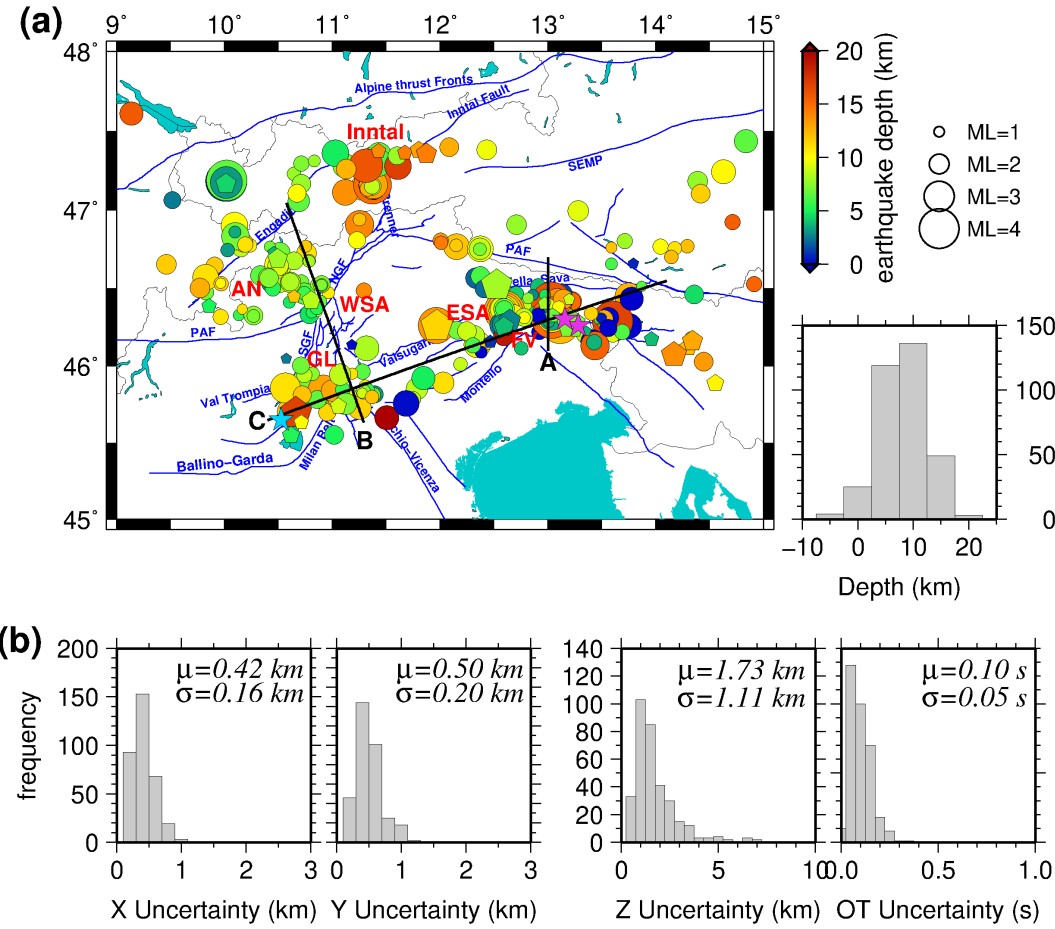

**Figure 12. (a)** Distribution of the 344 well–located earthquakes in the target area after McMC inversion (single–event mode). Circles are earthquakes that were incorporated in the simultaneous inversion and pentagonals are 43 additional earthquakes with fewer picks. The color and size of the circles indicate the depth and magnitude of the earthquakes, respectively (magnitude is taken from national catalogs). Purple stars show the location of Friuli earthquake of 1976 May 6 and its major aftershock on 1976 September 15 (Slejko, 2018) and the blue star is the Salò earthquake on 2004 November 24 (Viganò et al., 2015). Black lines indicate cross sections shown in Fig. 13. Faults are compiled from geological maps of the Italian geological survey, including Avanzini et al. (2010); Bartolomei et al. (1967); Bosellini et al. (1967); Braga et al. (1971); Castellarin et al. (1968, 2005); Dal Piaz et al. (2007) and complemented by fault traces published in Handy et al. (2010); Schmid et al. (2004, 2008). Legend: **ESA** - Eastern Southern Alps; **WSA** - Western Southern Alps; **FV** - Friuli–Venetia region; **GL** - Giudicarie–Lessini region; **AN** - Austroalpine Nappes,**PAF** - Periadriatic Fault, **NGF** - Northern Giudicarie Fault, **SGF** - Southern Giudicarie Fault,. The histogram in the right side indicates the depth distribution of the earthquakes in the crust. **(b)** Histograms of the hypocenter and origin–time uncertainties of the 344 well–located earthquakes after McMC inversion.





**Figure 13.** Seismicity distribution in depth along 3 cross sections of A, B, and C of Fig. 12a (for a better clarity the depth and length scales of cross section A are magnified by a factor of 1.5). Purple stars show the location of Friuli earthquake of 1976 May 6 and its major aftershock on 1976 September 15 (Slejko, 2018) and the blue star is the Salò earthquake on 2004 November 24 (Viganò et al., 2015). Geological structures of the FV region along cross section A are modified after Nussbaum (2000) with the blind Trasaghis Thrust taken from Galadini et al. (2005). Geological structures NW and SE of the NGF in cross section B are modified after Rosenberg and Kissling (2013) and Verwater et al. (in preparation), respectively. Legend: **NGF** - North Giudicarie Fault, **CA** - Campofontana Fault, **PT** - Priabona–Trambileno Fault, **CAL** - Calisio Fault.

**Friuli–Venetia (FV) region**

In accordance with the previous studies and national catalogs (and references therein Bressan et al., 2016; Reiter et al., 2018), most of the seismicity in our dataset occurs in the ESA, i.e., within the FV region, coinciding with the eastern part of the deformation front between the Adriatic microplate and the PAF. This was also the location of the destructive Friuli earthquake of 1976 May 6 and its major aftershock on 1976 September 15 (purple stars in Figs 12a and 13a&c), which are associated with the Susans–Tricesimo Fault and buried or "blind" Trasaghis Thrust, respectively (Poli et al., 2002; Galadini et al., 2005;

Slejko, 2018). We find most earthquakes around 13° E (close to the villages of Tolmezzo and Gemona) down to a depth of





around 17 km (Figs 13a and 13c – around km 220 corresponding to 13° E). However, the seismicity is distributed over a wider area (see also Fig. 13c eastern part; between km 140 and 280). As stated above, a direct connection of individual earthquakes in our dataset to known faults near the surface is difficult. Nevertheless, the clustering of seismicity in the FV region between the Alpine frontal thrusts and the Fella–Sava fault in the north suggest that several frontal thrusts and backthrusts are active.

According to the fault distribution (and naming) of Galadini et al. (2005), the seismicity along the cross section A (Fig. 13a) indicates that the Susans–Tricesimo and Trasaghis faults, and potentially the Maniago Thrust are the most active faults in this region. Most earthquakes are located below the Maniago Thrust, one of the main Alpine frontal thrusts in the tectonic model of Nussbaum (2000). This suggests that there is an active fault at greater depth (Fig. 13a). We interpret this to indicate a blind (i.e., that has not reached the surface), southward propagating thrust front. In the profiles of Galadini et al. (2005) and Merlini

et al. (2002) further to east, this deeper blind thrust (named Trasaghis Thrust in Galadini et al., 2005) reaches the surface, where it offsets Plio–Pleistocene sediments. More near–surface activity down to a depth of 6 km can generally be related to the Friuli thrust system (see Nussbaum, 2000) and Fig. 13a).

**Giudicarie–Lessini (GL) region**

The seismicity in the GL region correlates with the NNE–SSW transpressive fault system of Giudicarie Belt and the NW–SE tending Schio–Vicenza strike–slip fault system. Seismicity of the Giudicarie fault system is concentrated mainly in the south (southwest of Lake Garda at depths ranging from a few km down to 15 km, in agreement with Viganò et al. 2015; Fig. 12a). The big Salò earthquake on 2004 November 24 with $M_L$=5.2 (blue star in Figs 12a and 13c; Viganò et al., 2015) was suggested to have occurred on a low angle thrust connected to the steep Ballino–Garda Fault, which detaches the sedimentary cover

(hanging wall) from the underlying crystalline basement in its footwall (Viganò et al., 2008, 2013, 2015).

The NNW–SSE trending system of vertical strike–slip faults making up the Schio–Vicenza Fault is located between the active NW–dipping thrusts of the Giudicarie Belt to the west and the NNW–dipping thrusts of the Valsugana Thrust System to the east. The short observational period of our network and the low seismicity rate limits the unambiguous association of the earthquakes to specific faults. However, some of the captured seismicity seems to take place along the Campofontana (CA),

Priabona–Trambileno (PT), and Calisio (CAL) Faults (Fig. 13c – western section). The Schio–Vicenza Fault system is the most active structure in the GL region.

Figure 13b shows the distribution of seismicity superposed on the main geological structures, which indicates that seismicity is focused in its southern part of the GL region. This suggests that the most southern and deepest faults (Fig. 13b) remain active, while the more internal fault systems have become seismically inactive (Verwater et al., in preparation). Moreover, we observe

that the seismicity is deeper than the frontal thrust modeled from balanced geological cross sections (Fig. 13b; Verwater et al., in preparation), similar to observations within the FV region (as mentioned above). We interpret this to reflect southward propagation of the Southern Alpine deformation front (blind thrusts) towards the Po Plain.

**Lateral variations in clustering of seismicity from the WSA to ESA**





A cross section running orogen–parallel from the GL to the FV region (Fig. 13c) indicates a similar depth of earthquakes for both regions. The FV region shows relatively high seismic activity, located at the junction with the Dinaric Front (Doglioni and Bosellini, 1987). However, in the central part of cross section C (around km 150 of Fig. 13c), sparse seismicity may indicate a seismic gap (Anselmi et al., 2011; Burrato et al., 2008). Alternatively, the sparse seismicity (especially at shallow depths) could indicate that deformation within this area is occurring aseismically, as proposed by Barba et al. (2013) and Romano

et al. (2019) for the Montello Thrust (Fig. 12a). Strain rates within the Montello region are among the highest within the ESA (Serpelloni et al., 2016), which combined with its sparse seismicity indicates that the majority of deformation within this area is most likely occurring aseismically (Barba et al., 2013).

### Engadin and Austroalpine Nappes

The seismicity in the Austroalpine Nappes (close to Italy–Switzerland border, Fig. 12a) reaches a rather homogeneous depth around 11 km and does not seem to coincide with known geological structures in the area (e.g., Engadin Fault, Fig. 13b, left part). Although earthquakes are situated at a uniform depth, the distribution of these events in map view is quite diffuse (Fig. 12a). Note that no seismicity clusters around the Engadine Fault in cross section B (Fig. 13b, left part), although the map view shows that seismicity coincides with the fault trace of the Engadine Fault NE and SW of cross section B. This could be related

to block rotation along the Engadine Line (Schmid and Froitzheim, 1993).

### Inntal region

A cluster of seismicity is found in the Inntal region at a depth range of 5 to 18 km (12a). This region is known for earthquakes with strike–slip and oblique–slip mechanisms that have been attributed to the activity of the Inntal Fault (Reiter et al., 2018)

and the Alpine basal thrust (Reiter et al., 2003). Historical earthquakes (e.g., the earthquake of 1670 July 17 in Hall in Tyrol; Hammerl, 2017) were also reported. Current seismicity (12a) extends further south into the Brenner region, where it has been associated with activity of the Brenner normal fault (Reiter et al., 2005).

### Other structures

Several earthquakes in a depth range of 5 to 14 km are clearly aligned along the PAFbetween 12° E and 12.5° E, possibly indicating ongoing activity of this fault (see also Reiter et al., 2018). The ENE–WSW Valsugana Fault system does not seem to be seismically active, similar to the conclusion that Viganò et al. (2015) reached based on his analysis of the time frame of 1994–2007.

## 7   Conclusions

Carefully analyzed earthquake hypocenters and 1–D $V_P$ and $V_P/V_S$ models for the Southern and Eastern Alps (the time period of September 2017 to December 2018) have been presented in this study. The results can be summarized as follows:





- The automatized procedure of P– and S–wave first arrival–time picking provided the initial dataset for further visual/manual analysis of the picks in order to remove/modify mispicks and pick the ignored arrivals.

- By visual/manual inspection of the waveform data, we identified the potential quarry blasts which were used to
investigate the high accuracy of the derived hypocenters.

- The arrival–time dataset of the earthquakes was utilized for simultaneous inversion of 1–D velocity model, hypocenters, and station–corrections using a McMC approach without any prior assumption on the velocity or earthquake locations. The McMC approach was also validated with a synthetic dataset that was created using a close–to–realistic 3–D velocity model. The test showed that the McMC is a viable approach to yield highly accurate hypocenters of local earthquakes even in complex
3–D velocity conditions. Furthermore, we should emphasize that the quality classes of the picks have been incorporated in the McMC inversion and noise levels are recovered individually for each class of picks.

- The seismicity is found in diffuse clusters in the upper crust (0-20 km) with pronounced activity in broader zones of the Alpine frontal thrust, e.g., FV region, along the GL and Schio-Vicenza domains, and in the Internal Alpine and Inntal areas. The seismicity analyzed here (16 months of data) confirms previous seismotectonic characterizations based on long-
term records. We interpret the deeper seismicity in the FV and GL regions as an indication for southward propagation of the Southern Alpine deformation front (blind thrusts).

- The 1–D $V_P$ and $V_P/V_S$ models and the reassessed precise hypocenters (with average uncertainties of around 500 m for the epicenters and 1.75 km for the depth) form necessary initial data for further studies, e.g., Local Earthquake Tomography (LET).

**Appendix A:  Automatized Arrival-Time Picking and Earthquake Relocation**

**A1   Workflow**

Our modified workflow consists of three main stages: 1) the NonLinLoc ray-tracer (Lomax et al., 2000) for predicting initial P arrival-times based on preliminary origin-time and hypocenters (integrated earthquake catalog from national agencies) and the minimum 1-D velocity model of Diehl (2008); 2) the MPX algorithm (Aldersons, 2004; DiStefano et al., 2006) for improving
the first alerts of P-picks and classifying them into four quality classes (0 to 3, Table A1) based on Fisher statistics (Diehl, 2008, ;Appendix D: Users guide for MPX picking system); 3) the Spicker algorithm (Diehl et al., 2009a) for picking S-onsets based on P-picks, event location, and a preliminary S-pick (determined by NonLinLoc ray-tracing; Lomax et al., 2000). Spicker also rates picks into four quality classes (Table A1).

After each stage, in order to remove the effect of low-quality picks on hypocenter locations, a multiple relocation procedure
using HYPO71 (Lee and Lahr, 1975) and various subsets of picks is performed. The procedure starts with localization of the events using the most reliable picks (higher quality classes and closer distance to the event) and this first location is considered to be the reference for testing further picks. Thereafter, the picks with lower quality and those from distant stations are incorporated one-by-one. The newly added pick is kept if its residual is less than a distance-dependent, user-defined value and also if the root mean square (RMS) residual of the event location is still acceptable (less than 1 s). Picks leading to not





acceptable RMS residuals are removed from the dataset. Moreover, in this relocation procedure, not only low-quality picks but also events with a small number of high-quality picks are removed. This procedure results in not dominating the location by low-quality picks or not sorting out better picks. Since MPX and Spicker depend on the correctness of the predicted arrival-times (the predicted arrival-time itself depends on the event location), therefore picks and location are highly inter-dependent. This can be controlled by iteratively improving the event location, repicking arrival-times based on the updated location, and

adding or removing dubious picks in each step.

## A2    Uncertainty assessment of arrival-time picks

To assess the performance of the picking algorithms, a dataset consisting of 14 earthquakes with various magnitude ranges and randomly scattered in the region, are manually analyzed. Based on the arrival-time uncertainty, the P- and S-picks are qualified

into 5 classes (0, 1 ,2, 3, and 4 meaning respectively 100%, 75%, 50%, 25%, and 0% of contribution into location) and they are considered as reference for calibration (Table A1).

**Table A1.** Statistics of the automatic and reference picks associated with the quality classes for a dataset of 14 earthquakes.

| pick quality class | #auto picks | #reference picks | manual arrival-time uncertainty (s) | auto-picking $\mu^*$ (s) | auto-picking SD** (s) |
|---|---|---|---|---|---|
| P-all | 757 | 871 | - | 0.071 | 0.189 |
| P-0 | 195 | 299 | 0.05 | 0.001 | 0.039 |
| P-1 | 245 | 156 | 0.1 | 0.063 | 0.123 |
| P-2 | 160 | 171 | 0.2 | 0.158 | 0.283 |
| P-3 | 157 | 120 | 0.3 | 0.061 | 0.384 |
| P-4 | - | 125 | >0.3 | - | - |
| S-all | 343 | 445 | - | 0.103 | 0.642 |
| S-0 | 21 | 69 | 0.1 | 0.056 | 0.053 |
| S-1 | 56 | 61 | 0.2 | 0.07 | 0.132 |
| S-2 | 77 | 108 | 0.3 | 0.115 | 0.235 |
| S-3 | 189 | 121 | 0.4 | 0.223 | 0.817 |
| S-4 | - | 86 | >0.4 | - | - |

\* The mean of misfits between automatic and reference picks

\*\* The standard deviation of misfits between automatic and reference picks





The misfits between reference and automatic arrival-times are calculated (individually for each quality class of the P- and S-picks) and the mean and standard deviation values are shown in Table A1. The picks with higher qualities have smaller mean values which means their misfits are lower (*i.e.,* automatic picks are closer to reference picks). Moreover, except for S-3, the standard deviation of each class is very well comparable with manual arrival-time uncertainty.

Another way of assessing the performance of the automatic picking algorithms is the so-called confusion matrix. The confusion matrix, in general, shows the performance of predicted vs. the actual data of different classifications in machine learning (Provost and Kohavi, 1998). For our dataset, it shows automatic picks vs. reference picks (similar to DiStefano et al., 2006; Diehl et al., 2009a; Sippl et al., 2013). In the confusion matrices displayed in Fig. A1, three values represent the performance of the automatic pickers in each quality class: 1) number of picks, 2) hit-rate (fraction of reference picks of a certain class that are assigned to a specific class by the automatic picking) that is summed up to 100% in each row, and 3) the standard deviation of the residuals between automatic and reference picks. The red and orange colors refer to picks that are upgraded to higher qualities by the automatic algorithms (unfavorable), whereas the green sectors show the picks that are underestimated by the automatic picker (more eligible). Having low hit-rates in the orange and red cells and high hit-rates in the green fields point toward a conservative assessment of the picking algorithms. The white fields in the confusion matrices illustrate the picks that are ideally qualified based on the reference dataset.

| | | Automatic P-picks from MPX | | | | |
| --- | --- | --- | --- | --- | --- | --- |
| | | 0 | 1 | 2 | 3 | ignored |
| Reference P-picks | 0 | N=158 53.38% σ=0.039 | N=81 27.36% σ=0.124 | N=13 4.39% σ=0.069 | N=19 6.42% σ=0.301 | N=25 8.45% - |
| | 1 | N=26 16.77% σ=0.089 | N=61 39.35% σ=0.123 | N=31 20.01% σ=0.124 | N=19 12.26% σ=0.300 | N=18 11.61% - |
| | 2 | N=6 3.59% σ=0.009 | N=38 22.75% σ=0.173 | N=41 24.55% 0.283 | N=30 17.96% σ=0.273 | N=52 31.15% - |
| | 3 | N=2 1.69% σ=0.070 | N=21 17.8% σ=0.214 | N=23 19.49% σ=0.166 | N=21 17.8% σ=0.384 | N=51 43.22% - |
| | 4 or ignored | N=1 0.43% - | N=40 17.17% - | N=51 21.89% - | N=65 27.89% - | N=76 32.62% - |

| | | Automatic S-picks from Spicker | | | | |
| --- | --- | --- | --- | --- | --- | --- |
| | | 0 | 1 | 2 | 3 | ignored |
| Reference S-picks | 0 | N=9 13.04% σ=0.053 | N=8 11.60% σ=0.088 | N=6 8.69% σ=0.250 | N=8 11.60% σ=0.665 | N=38 55.07% - |
| | 1 | N=3 5% σ=0.063 | N=11 18.33% σ=0.132 | N=10 16.67% σ=0.319 | N=7 11.67% σ=0.586 | N=29 48.33% - |
| | 2 | N=4 3.74% σ=0.081 | N=15 14.02% σ=0.140 | N=17 15.89% σ=0.235 | N=19 17.75% σ=0.882 | N=52 48.60% - |
| | 3 | N=2 1.67% σ=0.074 | N=7 5.83% σ=0.115 | N=15 12.5% σ=0.227 | N=32 26.67% σ=0.817 | N=64 53.33% - |
| | 4 or ignored | N=2 0.91% - | N=15 6.85% - | N=28 12.79% - | N=121 55.25% - | N=53 24.20% - |

**Figure A1.** Confusion Matrices of P- and S-picks, comparing the automatic picks with reference picks. Columns are quality classes of the automatic P-picks from MPX (upper table) and S-picks from Spicker (lower table), and rows represent the pick weighting classes provided by the human analyst for reference data. The parameters in each cell of the matrices are explained in the text.

The confusion matrices of both MPX and Spicker show relatively high values of hit-rate (alternatively N) in some orange and red cells that we interpret as mispicks. For instance, the MPX (Spicker) picked 157 P-picks (166 S-picks) that are either ignored or qualified as class 4 by a human analyst. As the human analyst adopted class 4 for far stations with unclear arrivals, far automatic picks are unreliable. On the other hand, it seems that Spicker ignored a large number of picks (high hit-rates in the most right column of the lower matrix). This is eminent for picks with manual quality class 0 that are potentially associated with closest stations based on human analysis.

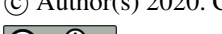



*Data availability.* SWATH–D data are archived at the GEOFON datacenter (FDSN network code ZS, 2017–2019; Heit et al., 2017; GFZ, 2020). Data of the permanent (network codes BW, CH, GR, IV, MN, NI, OE, OX, RF, SI, SL, ST) and temporary (network code Z3) networks

of AASN are available through the GEOFON data center at (GFZ, 2020). The supplementary materials related to this manuscript are available in the attached .zip archive.

*Team list.* The complete member list of the AlpArray Working Group can be found at http://www.alparray.ethz.ch.

*Author contributions.* AJN performed data preparation, analysis, and simultaneous inversion; she also prepared all figures and most of the text. CH supervised the analysis and contributed to the interpretation of the results. TR assisted with the inversion and modified the code.

VV, ELB, and MRH contributed to the interpretation and the discussion of results. MW promoted the work and performed text corrections. All coauthors contributed to the text and the figures.

*Competing interests.* The authors declare that they have no conflict of interest.

*Acknowledgements.* We would like to express our appreciation to Ben Heit for the coordination of the SWATH–D network, and to all people involved in the instrument preparation, field work, and data archiving. We thank numerous landowners for their willingness to

host the stations, and all communities, authorities, and institutes in the region for their great support of the project. Funding for the network came from the German Science Foundation DFG through the Priority program SPP 4D–MB and the GFZ Potsdam. Instruments for the SWATH–D network were provided by the Geophysical Instrument Pool Potsdam GIPP of the GFZ Potsdam. Discussions with many colleagues within the Priority program are greatly acknowledged. We also appreciate Christian Sippl for his guidance in implementing the automatic workflow. We also would like to acknowledge the national seismological agencies of Italy (INGV, OGS), Austria

(ZAMG), Switzerland (SED) for providing their comprehensive earthquake catalog. The authors would like to thank to the AlpArray Seismic Network Team: Gyo¨rgy HETE´NYI, Rafael ABREU, Ivo ALLEGRETTI, Maria-Theresia APOLONER, Coralie AUBERT, Simon BESANÇON, Maxime BE`S DE BERC, Go¨tz BOKELMANN, Didier BRUNEL, Marco CAPELLO, Martina Cˆ ARMAN, Adriano CAVA-LIERE, Je´roˆme CHE`ZE, Claudio CHIARABBA, John CLINTON, Glenn COUGOULAT, Wayne C. CRAWFORD, Luigia CRISTIANO, Tibor CZIFRA, Ezio D'ALEMA, Stefania DANESI, Romuald DANIEL, Anke DANNOWSKI, Iva DASOVIC´, Anne DESCHAMPS,

Jean-Xavier DESSA, Ce´cile DOUBRE, Sven EGDORF, ETHZ-SED Electronics Lab, Tomislav FIKET, Kasper FISCHER, Wolfgang FRIEDERICH, Florian FUCHS, Sigward FUNKE, Domenico GIARDINI, Aladino GOVONI, Zolta´n GRA´CZER, Gidera GRO¨SCHL, Stefan HEIMERS, Ben HEIT, Davorka HERAK, Marijan HERAK, Johann HUBER, Dejan JARIC´, Petr JEDLICˆKA, Yan JIA, He´le`ne JUND, Edi KISSLING, Stefan KLINGEN, Bernhard KLOTZ, Petr KOLI´NSKY´, Heidrun KOPP, Michael KORN, Josef KOTEK, Lothar KU¨HNE, Kresˇo KUK, Dietrich LANGE, Ju¨rgen LOOS, Sara LOVATI, Deny MALENGROS, Lucia MARGHERITI, Christophe MARON,

Xavier MARTIN, Marco MASSA, Francesco MAZZARINI, Thomas MEIER, Laurent ME´TRAL, Irene MOLINARI, Milena MORETTI, Anna NARDI, Jurij PAHOR, Anne PAUL, Catherine PE´QUEGNAT, Daniel PETERSEN, Damiano PESARESI, Davide PICCININI, Clau-





dia PIROMALLO, Thomas PLENEFISCH, Jaroslava PLOMEROVA´, Silvia PONDRELLI, Snjeţan PREVOLNIK, Roman RACINE, Marc RE´GNIER, Miriam REISS, Joachim RITTER, Georg RU¨MPKER, Simone SALIMBENI, Marco SANTULIN, Werner SCHERER, Sven SCHIPPKUS, Detlef SCHULTE-KORTNACK, Vesna Sˇ IPKA, Stefano SOLARINO, Daniele SPALLAROSSA, Kathrin SPIEKER, Josip

STIPCˆEVIC´, Angelo STROLLO, Ba´lint SU¨LE, Gyo¨ngyve´r SZANYI, Eszter SZU˝CS, Christine THOMAS, Martin THORWART, Frederik TILMANN, Stefan UEDING, Massimiliano VALLOCCHIA, Ludeˇk VECSEY, Rene´ VOIGT, Joachim WASSERMANN, Zolta´n WE´BER, Christian WEIDLE, Viktor WESZTERGOM, Gauthier WEYLAND, Stefan WIEMER, Felix WOLF, David WOLYNIEC, Thomas ZIEKE, Mladen TֲIVCˆIC´ and Helena TֲLEBCˆI´KOVA´. We used Generic Mapping Tools (GMT; Wessel and Smith, 1991; Wessel et al., 2019, https://www.generic-mapping-tools.org/) for plotting the figures.



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
