# Peer review of "Relocation of earthquakes in the Southern and Eastern Alps (Austria, Italy) recorded by the dense, temporary SWATH–D network using a Markov chain Monte Carlo inversion"

_Solid Earth, 2020_

## Referee Comment (RC1) · Anonymous Referee #1 · 20 Jan 2021

In this paper, the authors relocate 344 earthquakes in the southern and eastern Alps by exploiting arrival time data from the temporary SWATH-D network, supplemented by Alp-Array stations. Overall, the paper is well written, the methods well explained and tested, and the results carefully obtained and discussed. I suppose one could argue that the work they carried out would normally be integrated into a local earthquake tomography study, in which preliminary location of events is undertaken and a robust 1-D reference model is generated prior to the tomography. However, given that this is a new and large dataset from a region of significant interest, that cutting edge methods

were used for the hypocenter locations and 1-D velocity structure determination, and that the final earthquake distribution does provide insight into active faults in the region, I would be happy to see the paper published following minor revisions.

(1) Line 9: I would be tempted to replace "precise" with "robust".

(2) Line 43: I'm not sure I would say "...they depend not only on proper choices of initial values for hypocenter coordinates..." - what is meant by "proper" in this context – that they are close enough to the "true" location to make the inverse problem locally linear?

(3) Line 44: I don't think I would describe damping as a "technical parameter". It is better described as regularisation.

(4) Lines 115-119: In a way it's a pity that automatic event detection was skipped, since presumably the combined array used in this study provides a much denser data coverage of the region compared to what national seismological agencies have access to. Consequently, there is probably quite a lot of seismicity that has been overlooked, and if the purpose of this paper is to examine earthquake distribution and its relationship to active faulting in the region, then that is somewhat unfortunate. However, on reading the next lines, it appears that an automated picking algorithm was applied, but presumably only to data windowed by the pre-existing catalogue? It would also be interesting to know why the combined national seismological agencies were able to detect and presumably locate 2,639 local events, yet only 384 were deemed good enough for the current study. I understand the issue of seismic gaps and noise, but are the national networks more dense than the network used in this study in some regions?

(5) Line 140: Should be "..and thus is easily mispicked."

(6) Line 175: It is not clear to me how station terms can account for 3-D variations in velocity structure that are not considered in the inversion for 1-D velocity structure.

(7) Line 188: "...for a very large..." - should be "...a very large...".

(8) Lines 188-193: Perhaps I misunderstand something, but the velocity model is defined by a series of horizontal layers, each with constant Vp and Vp/Vs? So why does a 3-D Voronoi mesh with 1 km spacing vertically and horizontally come into it? While it may be technically correct to use this terminology, isn't it less confusing to describe this as a regular mesh in 3-D with 1 km spacing? Also, is there any need to correct for Earth's sphericity, since I believe that the Podvin and Lecomte method is in Cartesian coordinates?

(9) Lines 193-195: This is perhaps slightly confusing, because the first part seems to indicate an L1 measure of misfit, but with a Gaussian likelihood function, the actual misfit would be L2.

(10) Line 215: It would be interesting to have some numbers on how many iterations constitute the burn-in phase, and how many subsequent iterations were used to build the posterior PDF. (11) Line 263: Should be "earthquake", not "earthquakes".

(11) Line 273: I find it interesting that the Vp uncertainty is almost zero in the 0-20 km depth range, which the authors put down to dense ray coverage. What values do these uncertainty estimates take, and are they comparable with, say, the standard deviation of the lateral heterogeneity of the synthetic model input at that depth?

(12) Line 284: Should be "Results and discussion".

(13) Line 288: What happens if all model unknowns are allowed to vary in the initial tranche of iterations?

(14) Line 319: Should be "A detailed interpretation of the pattern of corrections...."

(15) Line 323: Should be "Estimation of hypocenter accuracy...."

(16) Section 6.3: This section is essentially fine, although with a relatively modest database of 344 earthquakes, it is not entirely clear what new insights are brought to the table beyond what might be gleaned from national catalogues that have been accumulated over periods of decades and involve many more earthquakes (albeit not as well located). To some extent this brings us back to the question of trying to use

auto-detection methods that take advantage of this large array to find potentially large numbers of small earthquakes missed by the national agencies.

---

## Referee Comment (RC2) · Edi Kissling (Referee) · 20 Jan 2021

Review of SolidEarth-manuscript Najafabadi et al. Dec2020

The manuscript regards the compilation of a local earthquakes catalog of 16 months period with the application of a few modifications and improvements to standard location procedure using the dense AlpArray and SWATH-D temprary station network. The study comprises different topics -procedural steps and results- each of potential interest to a wide range of readership.

[Figure]

Interesting enough the first such topic addressed in the abstract is the description and attempted correlation of the seismicity with the regional geology and tectonics. While certainly precisely relocated, the 344 local earthquakes of a 16 months period by no means could be taken as representative for the seismicity in the region and it should not come as a surprise -and not be seen as regional "characteristics"- that it appears in clusters. For a seismotectonic interpretation linking such observed clusters with tectonic faults to conclude, f.e., that "the general pattern of seismicity reflects head-on convergence of the Adriatic indenter with the Alpine orogenic crust." one would have hoped the authors to take advantage of the great data set with on average 36 P observation per event to complement the hypocenter locations with focal mechanisms at least for the larger magnitude events. Furthermore, a comparison and thourough discussion of the relation of the presented high-precision short-period seismicity with the long-term seismicity pattern reveiled by the official catalogs over the past 30 years is not only possible but necessary.

The main topic and work of the study regards the successful application of a Markov chain Monte Carlo inversion of the 12,534 P and 7,258 S observations from 344 local earthquakes to obtain a 1D velocity model and station delays for the region that allows high-precision hypocenter locations. The derivation of the model is well explained and complemented with the description of a synthetic test to provide a statistical estimate of the location uncertainties. In addition, a "ground truth" location experiment with quarry blasts is presented and discussed. This part of the manuscript is very clearly presented and contains a lot of technical details that allow the interested specialist to follow most steps. Considering the readership that might be interested in the seismicity and their tectonic interpretation though, I suggest to most of the chapter 5 could be moved to the supplementary material. What is missing, however, is a critical discussion of the results and, in particular, their relevance and meaning for the seismic catalog of 344 events presented. Considering that with the Markov chain MC inversion the authors address the coupled hypocenter-velocity model problem for the complet (very high-quality in terms of number of observations per event) 344 event data set, I do not understand

why there is no mentioning about the internal consistency of the hypocenter solution or about the great potential of these results as initial data (hypocenters and model alike) for 3D seismic tomography. Rather, the list of relocated earthquakes is presented simply as a higher-precision-"than INGV/ZAMG" catalog for the region.

Finally, the study also contains a section interesting for seismologists (observatory tasks and seismic tomography) about the semi-automated picking of such a large data set. Much of the description of this work part is already allocated in the supplementary material and it should remain there. It would be logic though if the reference to this work and the presentation and discussion of the results would be appearing before and not after the Markov chain MC inversion of the data set (see Figure 7). Furthermore, some details important for the specialist are missing (see individual points 10 to 15 below).

The chapters 6 (Results in discussion) and 7 (conclusions) read like they were written by different people with totally different interests and perspectives and the only connection between the two parts are the 344 precisely located hypocenters. There is no geologic-lithologic interpretation or at least comment about correlation presented between the other results (notabene of great importance for the claimed reliability and accuracy of the hypocenters) of the coupled problem, the velocity model and station delays. The tectonic interpretation of the seismicity presented in chapter 6.3 (pages 21 to 25) is missing taking explicitly into account (and explaining to the reader how and why this is used as arguments for the interpretation) the great advantage of this study (having an event data set of high internal consistency and high precision hypocenter location of quantitatively known uncertainties) and the significant limitations (pre-selected events of unspecified magnitude of completeness and only 16 months of observation period).

In consideration of the above general remarks on quality and deficiancies , I suggest moderate revision of the manuscript before publication.

Specific points:

(1) Line 12. Replace "accuracy" with "precision estimate". Note that in line 13 you correctly assess the "accuracy" with the blast location test. (2) Line 15. Delete the rest of the sentence after:" ..1.7 km in depth." (3) Line 27. Replace "accuracy" with "precision" (4) Line 48. "... has the advantage of being" largely "independent ..." (5) Line 51. "best model(s)." a note on ambiguity would be useful (6) Line 64. Please outline Adriatic microplate in one of the Figures. (7) Lines 67-75. Needs a figure to show the strain if you keep the introduction as is and the chapter 6.3. (8) Figure 1. See point 6 above. Red box in bottom figure does not correspond with bounds of upper figure. This figure is not providing all necessary tectonic information mentioned in the manuscript. You should note that the seismic catalog presented by ISC is by far not complete down to magnitude 2. If you want to show the big picture use either EMSC catalog likely complete to M3 or ISC likely complete to M3.5. Otherwise you could use a composite of the various national catalogs that probably are complete to M2.5. (9) Line 85. Actually there are earlier catalogs that were compiled: European Geotraverse Blundell et al. 1992, Solarino et al. 1997 (10) Lines 115 to 119. You need to elaborate in detail (this can be done in supplementary material but it is absolutely necessary to have this information) how you identified the "common" events and how in the end you established the event list of the 2619 events. (11) Lines 120 - 126. The discussion of the results of this semi-automated picking (that is well described in suppl.) needs to be more extensive and detailed. On what basis did you define the selection criteria (gap<200 , why not <180') (RMS <1s), why no mentioning of number of P obs? Did you check all 12534 P obs manually? (12) Line 134-6 and Figure 3. The Wadati diagram shows significant numbers of observations with +/-3s residual relative to constant Vp/Vs ratio. Note the the Vp/Vs ratio varies within the crust and at Moho. You may see this in the Figure as the straight line is systematically shifted onto the side of the highest point density after about 25s P travel time. What residual range do you define as corresponding to the sum of 3D, lithologies and regular Gaussian observation uncertainty effects and what value denotes an outlier? (13) Line 144 and Figure 4. "Checking the phase-type is extremly important." I fully agree and record

section display is a good zero-order approach. However, I do not think your figure 4 is of help for doing this. How realistic is the ak135 global model for P phase identification considering the Moho topography by Spada et al. 2013 (your Fig.5c) or the 3D LET model by Diehl et al. 2009 but most important the literally more than a dozen refraction seismic lines that have been published (for a review see Kissling et al. 2006). (14) Line 152. Please provide clear evidence and explain in detail strategy to identify PmP phase by using a totally inadequat 1D model. (15) Lines 152/3. "the number of outliers, ..., is not significant to the total number of picks." What value do you define as being an outlier? (analog question to point 12) Note there are dozens of observations +/-2s from the main intensity of data points (that by the way is totally off your Pg line) and that individual hypocenter location precision (and even more important for accuracy) is in truth measured by the fit of just those observations that refer to the specific event. (16) Line 218. "... does not depend on initial hypocenters, ..." I seriously doubt this (does not depend) and suggest to phrase it differently. Consider how you would identify an outlier with Figure 4 if you do not have a rather good idea the initial hypocenter! Furthermore, consider that you were using a priori information from existing catalogs for your semi-automated picking and that even with all this information you apparently found mispicks and had to select the 384 best events. (17) Line 226. This does not provide an "accuracy" estimate! May be internal consistency, precision. And you need to provide reasons why this should be expected to be of relevance for the real individual event locations. By the way, your blast test shows otherwise! (18) Line 230. This statement about Moho velocities is simply wrong. No 7km/s velocity has been reported in the Alps. Please check the literature. (19) Line 232. The 19km are just along the flank of the Ivrea body and not relevant for the Moho topography beneath the Po plain. (20) Line 233. -5km is much to high and you are doing something wrong if you need to avoid rays through air by such model top elevation. Note that the increase in pressure within the earth causes a velocity increase with depth and the seismic waves to show a downward curvature You should use the average station elevation for ray tracing. (21) Line 245. Choosing a constant Vp/Vs ratio of SQR(3) is very

problematic as we know it is wrong because it varies and likely the average is different. (22) Line 250/1. Not only refer to table but provide correct value here. (23) Figure 5. Your model extent in Figure 5a does not correspond with your map extent in Figure 5c AND either of these extents differ from your study region shown in Figure 1 AND all of these are different than your Figure 2. Make certain you everywhere show the same study region extent, if you want to show more area around, then mark the study region. (24) Figure 5b. The Moho topography is wrong. There is a Moho offset across the plate boundary but you show a vertical Moho interface! (25) Lines 259/60. "average uncertainty of 240m in longitude, 270m in latitude ...." How did you determine that? In such way this information is not usefull. With what probability do we have what location uncertainty for any single hypocenter? (26) Figure 7. Move to supplementary material. (27) Figure 8. Again a DIFFERENT STUDY REGION SHOWN!. What about the stations to the West of the Tauern window (as example, there are other regions with no visible symbols)? Do they have all zero station delay values or did you not obtain any values for them? Please explain in more detail what the velocity-depth function shows. In may view, it documents the data set is not capable to resolve the velocity structure below 30km depth and certainly not the Moho. This does not come as a surprise as it is well known that you loose vertical resolution below your deepest hypocenters. (28) Lines 297 to 304. This model discussion is inadequate with regards to the previously published information about the crustal structure. If your model does not allow to resolve it, then say so and it is OK. But do not claim it is in agreement with prior independent knowledge if it is obviously not. (29) Line 320. "it proved to be useful for accurately localizing earthquakes ..". I am missing the prove. Please explain how this was proved. (30) Figure 10. Figure 10b I would again derive the conclusion from this figure that you are lacking resolution power below 30km for Vp and below 20km for Vp/Vs ratio. Figure 10c. Now this looks like the study region. Why not always marking this extent where you do have data from? How do you interpret the distribution of the station delays? Note that there is a single station delay strongly different from all others within its vicinity located near 11.7E/47N. If I obtained such

result I would check if it is real or caused by bad data of some sort. Note that otherwise your largest station delays are all within the periphery of the study region as this is well known from minimum 1Dmodel applications. (31) Lines 325 to 335 and Figure 11. What hypocenter depth did you test these blast locations with? How do you define the mis-location vector? Relative to the center of the quarries or do you know the precise location within the quarry for each blast? This accuracy test shows that your previous precision estimates were a bit to optimistic but the accuracy is still very good. (32) Line 331. Please explain in theory why you suggest that including S observations improve the hypocenter location solution? (33) Lines 340/1. Your absolute depth uncertainty has been documented by your ground truth accuracy test (Fig.11) to be a few km and the epicenter location uncertainty is about 1km. Please correct your numbers. (34) Lines 343/4. These differences are indeed significant. However, as it just regards a selected best event data set with on average 36 P observations this is comparing apples and grapes. Your data set is excellent for seismic tomography but absolutely not representative for a seismicity catalog (magnitude of completeness? Just 16 months). On the other hand, the national seismic catalogs contain many poorly locatable –or you could also say difficult to locate- events that need extra processing time to obtain a complete catalog and there is the significant difference in number of stations. I believe it would be useful to discuss thourougly these difference in addition to presenting just the numbers. (35) Figure 12. Figure 12b is not needed, just provide the uncertainty estimates. Note that for cluster interpretation relative hypocenter location uncertainty estimates (and that is what you obtain with your Markov chain MC inversion of the coupled problem that includes a joint hypocenter determination approach) are most important while for absolute location obviously the accuracy is key. (36) Figure 13. For seismotectonic interpretation of clusters along a fault system, you should definitely employ focal mechanisms.

---

## Referee Comment (RC3) · Anonymous Referee #3 · 15 Feb 2021

The present manuscript by Jozi Najafabadi et al. presents the compilation of a seismicity dataset that will, I presume, eventually be used for a local earthquake tomography study of the Eastern Alps. Using recordings from the dense SWATH-D deployment, they present a careful procedure of obtaining and verifying arrival time picks, derivation of optimal hypocenter locations and a best-fit 1D velocity model. The Bayesian approach for the inversion of hypocenters, velocity model and station corrections is something new, and the present manuscript provides a nice case study for its application. Lastly, the obtained hypocentral locations are compared to mapped faults, from

which the apparent activity or non-activity of a number of structures in the Eastern Alps is inferred. This last part is where I see some potential problems that will make some changes to the manuscript necessary. Overall, the paper is well written and definitely of interest to the readership of Solid Earth and the special volume "New insights on the tectonic evolution of the Alps and the adjacent orogens". I recommend moderate revisions before publication.

General comment:

In section 3.2, it is briefly mentioned that the events for which arrival times on SWATH-D stations were obtained and that were then relocated, used for deriving the 1D velocity model etc. were selected from a synthesized catalog that was based on the bulletins of national agencies. For the tectonic interpretation to be viable, this part needs to be made much more transparent. While it is fine to choose a subset of events based on network criteria when working towards a tomography study, it is a completely different thing when the activity or (more crucial) non-activity of faults is inferred from such a subset. In clearer words: the authors need to convincingly show that their chosen sub-set of events is representative, and does not systematically miss events from certain regions. I suggest to provide a map with all 2639 events from the different national catalogs, in which the chosen 384 are marked. It would likely be even better if the station distribution from the different national networks could be shown as well. I also suggest to better describe the reasoning behind this approach of choosing a subset of events from national bulletins. Are the national bulletins complete enough that one can exclude that the dense SWATH-D network contains signals from small, previously undetected events? Or was the focus on the larger events that would generate arrival times at a larger number of stations?

I agree with a previous reviewer that fault plane solutions would be nice to have for a more detailed tectonic interpretation. However, I can see that the main aim of the manuscript is the description of the dataset that will be used for tomography, and spec-ulate that the tectonics part was added mainly for the sake of the Special Volume topic.

I believe the careful derivation of the hypocenters and velocity model, using a rather novel approach and performing many quality checks, is in itself enough material, so that a deeper-going tectonic interpretation employing focal mechanisms is not strictly necessary here,

Specific comments:

l.25: Why were only data from 2017/2018 used when the stations ran into 2019? Should be mentioned with a word or two. Also, mentioning the total number of SWATH-D stations here could be useful, especially since the number of AlpArray stations that were also used is brought up in the next paragraph.

l.33/34: "to identify the status of the seismically active volume...". This is a strange formulation, and should be changed.

l.36: remove the

l.55ff: I would recommend to use fewer abbreviations, this is making the manuscript unnecessarily hard to read. Best limit abbreviation to a handful of terms that really show up a lot throughout the manuscript, and write out the rest (this is maybe also my personal taste...).

l.82: this bracket is not closing again

l.91: stuck should be struck; about what time interval are we talking for the ML>6 earthquakes? Last decades, centuries, millennia?

l.96/97: This statement is problematic, because while the present study is using a denser seismic network, the chosen approach of using a subset of events from agency bulletins (see General Comments) makes it impossible that previously missed events (if they exist) will be detected. Thus, the present study can do nothing to address the problem that is hinted at here (inactive region maybe because not well instrumented).

l.105: The "however" doesn't fit here

l.111: remove "stations"

ll.115-119: For me, this paragraph is the main problem of the manuscript as is. At least for the tectonic interpretation part, the authors need to convince the reader that no selection bias of earthquakes exists, i.e. that regions interpreted as aseismic based on the chosen 344 events are also aseismic if one looks at the entire >2600 events in the original database (see recommendatiosn in General comments). Also, the statement here seems to indicate that the national bulletins were deemed complete, which stands in contrast to l.96/97.

l.135/136: how were outliers defined, and where can I see outliers in Figure 3 (can they be marked?)

l.138: If a part of the goal audience are people mainly interested in the activity of structures in the (south)eastern Alps, the three phases and the triplication distance should be briefly explained, e.g. in a brief sketch that could be added to Figure 4. Also, giving an estimate of the overtaking distance, e.g. with the crustal thickness and velocity given in ll.147/148, would be beneficial.

l.140: be (remove ing)

l.158: is indicated the right word here?

l.160: Well, a Bayesian-type approach has been used for all these geophysial studies. As it is written, it sounds like this was always the exact same approach (which it wasn't)

l.176: structure (-s)

l.212: reformulate that first sentence

Section 5: I am not completely sure I understand the reasoning behind this test. The authors construct a first-order 3D velocity model of the Alps based on published data and perform a retrieval check using the real data (hypocenters, stations) as input. Thus any misfit in the output should stem from 3D structure and/or general uncertainty, but

only with the assumption of this specific 3D model...since the true 3D structure of the Alps will almost certainly differ from the utilized model (presumably only to second-order differences?), do we have any idea if the retrieved 3D effects are similar for a (subtly, substantially) different reality? I think the paragraph needs a more detailed description about the purpose of the test, what it is supposed to show and what it can not show. Nevertheless, I appreciate the effort that went into performing it!

l.268: Figure (-s)

l.271: reformulate "rather slight", add km after 50

l.275: reformulate "fluctuating"

l.319: "contains an overlay site effect" I'm not sure I understand what exactly is meant here.

l.324: This is not really a sentence

l.332/333: Can hypocenters from the INGV/ZAMG catalogs also be shown in Figure 11, to better illustrate the improvement in hypocenter location?

l.344: it would be interesting to elaborate a bit more on these differences; is there a trend, eg. with bulletins showing systematically larger or smaller depths?

l.345: This is not surprising, since no search for new events beyond those in national catalogs has been performed

l.358: mention the magnitudes of these events

l.398: See General Comment: how well can one argue for the absence of seismicity ("seismic gap") based on a catalog that was only a choice of 384 out of 2639 events? A map showing where the remainder of events (those that were not chosen) were located is essential if such a statement is attempted

l.405ff: spelling: Engadin vs. Engadine Fault

l.427ff: These (at least the first three dotpoints) are not results but the analysis steps that were carried out to retrieve the results. Either only results should be listed, or two separate listings for analysis steps and results are needed.

In the name list of AlpArray people, all those containing special characters have formatting issues (LaTeX syntax?)

Figures:

Figure 1: Typo in Tectonic units legend (forland should be foreland)

Figure 2: The fault lines in this plot are really hard to see. Choosing a larger linewidth would be helpful. A color scale for the topography would also be nice.

Figure 4: Please be more specific about what the "various depths" are that were used to obtain the travel-time corridor. Then, I do not see red dashed lines in the plot (as mentioned in the caption). Lastly, I would prefer if the meaning and implications of this nice plot were elaborated a bit more in the text. Is my interpretation of a change from a Pg-like to a Pn-like trend at around 150-200 km correct? How does this fit to theoretical overtaking distances, does this mean that the first arrival is picked everywhere?

Figure 6: missing values in the histograms (Mu and sigma)

Figure 7: should be 110 m in depth (not km)

Figure 8: What is the reason for the wide spread of models at very shallow depths? Looks like they did not converge there (same in Fig. 10). Thicker lines for the plusses and crosses in subfigure b would be helpful.

Figure 9: Typo in label: pahse should be phase; also, I guess the green line marks the end of the burn-in phase. I don't really see anything in the right subplot (which also has no axis-labels)

Figure 10: Would it be meaningful to compare the station correction pattern to the one from the synthetic test (Fig. 8)? In the caption, abbreviations CA and EA should be

Central and Eastern Alps (not Alpine)

Figure 11: Caption: remove one of the two "and" in fourth line; see level should be sea level

Figure 12: Comparing to Figure 2, it seems that the vast majority of events is at the edge or slightly outside of the SWATH-D network. This should be mentioned, and maybe the rectangle shown in Figure 2 can be added here?

Figure 13: Caption: "for a better clarity the depth and length scales of cross section A are magnified by a factor of 1.5"; does this mean vertical exaggeration of 1.5? Or only that it was upscaled by a factor 1.5 relative to profiles B and C (without any distortion)?
* * *

---

## Author Comment (AC1) · 25 Mar 2021

Dear Referee,

We would like to thank you for your insightful comments on the manuscript. We believe that your comments are well justified and thanks for your positive and encouraging review. In the following, we provide our responses to the remarks raised by you as blue text.

In this paper, the authors relocate 344 earthquakes in the southern and eastern Alps by exploiting arrival time data from the temporary SWATH-D network, supplemented by Alp-Array stations. Overall, the paper is well written, the methods well explained and tested, and the results carefully obtained and discussed. I suppose one could argue that the work they carried out would normally be integrated into a local earthquake tomography study, in which preliminary location of events is undertaken and a robust 1-D reference model is generated prior to the tomography. However, given that this is a new and large dataset from a region of significant interest, that cutting edge methods were used for the hypocenter locations and 1-D velocity structure determination, and that the final earthquake distribution does provide insight into active faults in the region, I would be happy to see the paper published following minor revisions.

**(1)** Line 9: I would be tempted to replace "precise" with "robust".
Done

**(2)** Line 43: I'm not sure I would say "...they depend not only on proper choices of initial values for hypocenter coordinates..." - what is meant by "proper" in this context – that they are close enough to the "true" location to make the inverse problem locally linear?
Yes, by proper we mean close enough to the true location. This explanation is now added in brackets in the updated manuscript.

**(3)** Line 44: I don't think I would describe damping as a "technical parameter". It is better described as regularisation.
Agree – changed to "regularization"

**(4)** Lines 115-119: In a way it's a pity that automatic event detection was skipped, since presumably the combined array used in this study provides a much denser data coverage of the region compared to what national seismological agencies have access to. Consequently, there is probably quite a lot of seismicity that has been overlooked, and if the purpose of this paper is to examine earthquake distribution and its relationship to active faulting in the region, then that is somewhat unfortunate. However, on reading the next lines, it appears that an automated picking algorithm was applied, but presumably only to data windowed by the pre-existing catalogue? It would also be interesting to know why the combined national

seismological agencies were able to detect and presumably locate 2,639 local events, yet only 384 were deemed good enough for the current study. I understand the issue of seismic gaps and noise, but are the national networks more dense than the network used in this study in some regions?

Yes, you are right, the automatic picking is applied only to the data windowed by the pre-existing catalog events. We totally agree that running a detection routine on the data of this dense and large network will probably yield a much more complete set of events (down to smaller magnitudes). However, our aim was to concentrate on locating the events with high precision rather than obtaining a comprehensive catalog. To locate the events with high precision they have to have a minimum magnitude and – in turn – a relatively large number of observations. We are confident that these (larger) earthquakes are contained in the permanent networks' catalogs. Additionally, one reason for this selection – besides the high-precision – is that we plan to use this dataset also for local earthquake tomography. We mention this now more frequently in the text.

Considering that no information on the precision of national catalog events is available and some of their events are poorly locatable (due to smaller number of stations and larger inter-station distances), we do an event selection based on our own location information after automatic picking:

We used the origin-time of 2,639 local events from national catalogs (now in appendix A of the manuscript) to start the automatic picking with (the automatic picking was not skipped). After the automatic procedure we saw that lots of events are either on the periphery of the network or too weak or too noisy to be detected by more than 5 stations. Moreover, automatic procedure ignored some of the clear picks in the nearest stations and had some suspected picks in the farthest stations. Therefore, we decided to manually/visually check the picks (semi-automated picking). For this purpose, we selected the events with gap<200 and RMS <1s (not too conservative) for further consideration. We explain later in the text that from these 384 events, some are blasts, and some are unclear to us, and some with very small numbers of picks.

**(5)** Line 140: Should be "..and thus is easily mispicked."
Done

**(6)** Line 175: It is not clear to me how station terms can account for 3-D variations in velocity structure that are not considered in the inversion for 1-D velocity structure.
We think our formulation was misleading. Therefore, we rephrased it as following:
"Moreover, the model **m** comprises station–corrections for P and S waves ($\tau_P$ and $\tau_S$), which account for travel-time effects (delayed or earlier arrivals) due to deviations of the 1–D model from the real 3–D velocity structure in the shallow subsurface beneath the stations."

**(7)** Line 188: "...for a very large..." - should be "...a very large...".
Done

**(8)** Lines 188-193: Perhaps I misunderstand something, but the velocity model is defined by a series of horizontal layers, each with constant Vp and Vp/Vs? So why does a 3-D Voronoi mesh with 1 km spacing vertically and horizontally come into it? While it may be technically correct to use this terminology, isn't it less confusing to describe this as a regular mesh in 3-D with 1 km spacing? Also, is there any need to correct for Earth's sphericity, since I believe that the Podvin and Lecomte method is in Cartesian coordinates?
For efficiency, we use a fast 2-D Eikonal solver to calculate the travel times. Therefore, we do not use a 3-D mesh but indeed a 2-D mesh with 1x1 km grid node spacing (vertically and

horizontally). The irregular 1-D velocity model (defined by the set of $Vp_i$, $Vp/Vs_i$) is converted into this fine 2-D mesh by assigning the velocity value of the nearest model node ($Vp_i$ or $Vp/Vs_i$, respectively) to the fine grid nodes (which is in fact some kind of Voronoi cell). Nevertheless, in order to avoid confusion, we modified the sentence to:

"Therefore, the irregular velocity model is converted to a fine and uniform mesh by setting the velocity at each mesh point to the value of the nearest point from the irregular model ($Vp_i$ or $Vp/Vs_i$, respectively). The fine mesh used by the Eikonal solver has a cell spacing of 1 km vertically and horizontally. "

Based on similar earlier studies, the dimension of our network seems to be small enough to neglect the sphericity, and similar inversion codes for local earthquakes (Velest; Kissling et al., 1994) or simul2000 (Thurber 1977) make this simplification as well (and numerous studies with similar-sized networks)

The actual model is a one-dimensional one, i.e., a set of n layers with constant P-velocities and Vp/Vs ratios. The model is described as a system of equivalent one-dimensional Voronoi cells (=layers). These Voronoi cells are degenerated: typically, Voronoi cells are 2- or 3-dimensional objects.

**(9)** Lines 193-195: This is perhaps slightly confusing, because the first part seems to indicate an L1 measure of misfit, but with a Gaussian likelihood function, the actual misfit would be L2.
For the Markov Chain method, we used the L2 norm and corrected this in the manuscript accordingly:
"a misfit function, particularly for each model, is defined as the summed squared differences between the observed (d) and calculated travel-times."

**(10)** Line 215: It would be interesting to have some numbers on how many iterations constitute the burn-in phase, and how many subsequent iterations were used to build the posterior PDF.
In this section (Method), we only explain the methodology and how the inversion works. We provide these numbers later in the text. As the burn-in phase and also the number of iterations is data-driven, these statistics are shown in figures 8 and 9a for the real data and also in the corresponding text. In particular, we used ~15,000 of the final models (every 1000[th] of all models after the burn-in phase) for the calculation of the posterior PDF; we describe this in Section 6.

**(11)** Line 263: Should be "earthquake", not "earthquakes".
Agree - changed

**(11)** Line 273: I find it interesting that the Vp uncertainty is almost zero in the 0-20 km depth range, which the authors put down to dense ray coverage. What values do these uncertainty estimates take, and are they comparable with, say, the standard deviation of the lateral heterogeneity of the synthetic model input at that depth?
The velocity uncertainty can depend on both the data (ray coverage and pick errors) and the lateral heterogeneity. A comparison between the lateral heterogeneity of the synthetic model and velocity uncertainty shows that they follow almost a similar pattern, although, not exactly equal. The lateral heterogeneity of the synthetic model is ~0.05 km/s between 2 and 20 km depth. It varies between 0.1 and 0.5 km/s below 20 km. These values are now added to the text.

**(12)** Line 284: Should be "Results and discussion".
Done

**(13)** Line 288: What happens if all model unknowns are allowed to vary in the initial tranche of iterations?

Actually, there is no difference in the final models (Vp, Vp/Vs, quake locations, etc.) when running the inversion with or without the "first phase" (where we do not change the initial velocity model but only quake locations). When using no "first phase" the run-time (CPU time) is only significantly increased. So, we introduced the first phase for practical reasons to accelerate the computation. Note that we start our Markov chains with completely random velocity models and initial quake locations, i.e., these initial values are potentially "very far away" from the final results, spanning a very wide range. Keeping the initial velocity models fixed during the "first phase" moves the initial quake locations to their approximate epicenters, thus accelerates the inversion.

**(14)** Line 319: Should be "A detailed interpretation of the pattern of corrections. . .."
Done

**(15)** Line 323: Should be "Estimation of hypocenter accuracy. . .."
Done

**(16)** Section 6.3: This section is essentially fine, although with a relatively modest database of 344 earthquakes, it is not entirely clear what new insights are brought to the table beyond what might be gleaned from national catalogues that have been accumulated over periods of decades and involve many more earthquakes (albeit not as well located). To some extent this brings us back to the question of trying to use auto-detection methods that take advantage of this large array to find potentially large numbers of small earthquakes missed by the national agencies.

Our database is mainly aimed for high location accuracy of the occurring earthquakes and not focusing on relatively small earthquakes (with only a few observations). Especially the focal depth-estimates are very sensitive to the quantity and quality of the picks and the velocity model. Furthermore, we intend to use this dataset for the calculation of the Local Earthquake Tomography (now this is emphasized more frequently in the manuscript). Therefore, instead of creating a comprehensive catalog, we focused on having a selection of the most consistent and precise hypocenters. We can also add that the microseismicity of this region is being further studied by other groups within the research project.

We think that the earthquake locations that we established in this study have the highest precision in the region (at this time) thus enables us to interpret for example the depths of the earthquakes at the southern Alpine front (section 6.3).

On behalf of the authors,
Azam Jozi Najafabadi

---

## Author Comment (AC2) · 25 Mar 2021

Edi Kissling (Referee)
kiss@tomo.ig.erdw.ethz.ch

Dear Prof. Kissling,

We would like to appreciate the time you have invested and your insightful comments on the manuscript. We take your comments very seriously and fully appreciate them during the revision of the manuscript. In the following, please find our response to your thoughtful remarks as blue texts.

The manuscript regards the compilation of a local earthquakes catalog of 16 months period with the application of a few modifications and improvements to standard location procedure using the dense AlpArray and SWATH-D temprary station network. The study comprises different topics-procedural steps and results- each of potential interest to a wide range of readership.

Interesting enough the first such topic addressed in the abstract is the description and attempted correlation of the seismicity with the regional geology and tectonics. While certainly precisely relocated, the 344 local earthquakes of a 16 months period by no means could be taken as representative for the seismicity in the region and it should not come as a surprise - and not be seen as regional "characteristics"- that it appears in clusters. For a seismotectonic interpretation linking such observed clusters with tectonic faults to conclude, f.e., that "the general pattern of seismicity reflects head-on convergence of the Adriatic indenter with the Alpine orogenic crust." one would have hoped the authors to take advantage of the great data set with on average 36 P observation per event to complement the hypocenter locations with focal mechanisms at least for the larger magnitude events.

We fully agree that for an in-depth seismotectonic study, focal mechanisms are essential. This issue was covered in other projects within the SPP. A manuscript dealing with focal mechanism was submitted to Solid Earth (Petersen et al., 2021), which we now reference in our manuscript. We also added some more information about the mechanisms in some sub-regions. Please see our response below.

Furthermore, a comparison and thourough discussion of the relation of the presented high-precision short-period seismicity with the long-term seismicity pattern reveiled by the official catalogs over the past 30 years is not only possible but necessary.

The general pattern of the seismicity agrees with previous studies, e.g., Reiter et al., 2018, which we mentioned in the section "Discussion" of the manuscript. Furthermore, in the discussion section we compare the seismicity distribution of our study with those from previous studies in several sub-regions (based on local networks or national agency data). In the updated manuscript we added some sentences regarding the similarity of the seismicity pattern to long-term seismic catalogs such as the SHARE catalog as well.

The main topic and work of the study regards the successful application of a Markov chain Monte Carlo inversion of the 12,534 P and 7,258 S observations from 344 local earthquakes to

obtain a 1D velocity model and station delays for the region that allows high-precision hypocenter locations. The derivation of the model is well explained and complemented with the description of a synthetic test to provide a statistical estimate of the location uncertainties. In addition, a "ground truth" location experiment with quarry blasts is presented and discussed. This part of the manuscript is very clearly presented and contains a lot of technical details that allow the interested specialist to follow most steps. Considering the readership that might be interested in the seismicity and their tectonic interpretation though, I suggest to most of the chapter 5 could be moved to the supplementary material. What is missing, however, is a critical discussion of the results and, in particular, their relevance and meaning for the seismic catalog of 344 events presented. Considering that with the Markov chain MC inversion the authors address the coupled hypocenter-velocity model problem for the complet (very high-quality in terms of number of observations per event) 344 event data set, I do not understand why there is no mentioning about the internal consistency of the hypocenter solution or about the great potential of these results as initial data (hypocenters and model alike) for 3D seismic tomography. Rather, the list of relocated earthquakes is presented simply as a higher-precision-"than INGV/ZAMG" catalog for the region.

We agree with your comment regarding the internal consistency of our hypocenter solution and the ultimate goal of creating this dataset and therefore, we added the following text to the section "introduction" of the updated manuscript:

"The dense, high-quality travel-time picks created in this study potentially lead to constrained hypocenter solutions with high internal consistency. This will enable us to identify the general pattern of seismicity on the surface and at depth throughout the region and contribute to the understanding of active tectonic processes. A further aim of the study is to derive a high-quality dataset suitable to be used in Local Earthquake Tomography (LET)."

A similar text is in the updated section "conclusion".

We totally agree that the study provides a very well-suited dataset for further studies such as local earthquake tomography. We emphasize this aspect now more clearly throughout the text.

In chapter 5 we intended to conduct a synthetic recovery test. With this test, we can study how (well) the synthetic hypocenters are recovered by the inversion routine and how the 1-D velocity (output) looks like the 3-D model (input). Also, how shallow velocity anomalies are "mapped" by the station-corrections and whether the input random noise is recovered by the McMC method. We think that this is an interesting test for showing the performance of the method, especially when keeping in mind that we apply a relatively novel McMC search. We report the outcome of this test (please see also our response to your point number 17). Similar tests are regularly employed in the inversions for (3-D) subsurface structure (i.e., tomography). Therefore, we decided to keep chapter 5 in the main text of the manuscript. Nevertheless, we modified the text in the updated manuscript and hope that the reason behind this test is now better clarified.

Finally, the study also contains a section interesting for seismologists (observatory tasks and seismic tomography) about the semi-automated picking of such a large data set. Much of the description of this work part is already allocated in the supplementary material and it should remain there. It would be logic though if the reference to this work and the presentation and discussion of the results would be appearing before and not after the Markov chain MC inversion of the data set (see Figure 7). Furthermore, some details important for the specialist are missing (see individual points 10 to 15 below).

In section 3.2 of the updated manuscript, we added a table and histograms (table 1 and figure 4) regarding the results (travel-time dataset) of the semi-automatic picking procedure. Further detail about the event list from national catalogs is now added to the updated manuscript as a new appendix A and it is referenced in section 3.2.

The chapters 6 (Results in discussion) and 7 (conclusions) read like they were written by different people with totally different interests and perspectives and the only connection between the two parts are the 344 precisely located hypocenters. There is no geologic-lithologic interpretation or at least comment about correlation presented between the other results (notabene of great importance for the claimed reliability and accuracy of the hypocenters) of the coupled problem, the velocity model and station delays. The tectonic interpretation of the seismicity presented in chapter 6.3 (pages 21 to 25) is missing taking explicitly into account (and explaining to the reader how and why this is used as arguments for the interpretation) the great advantage of this study (having an event data set of high internal consistency and high precision hypocenter location of quantitatively known uncertainties) and the significant limitations (pre-selected events of unspecified magnitude of completeness and only 16 months of observation period).

Our intention is to keep the conclusion (and in general the manuscript) more technical than interpretational (in terms of seismotectonic). However, we rephrased the whole conclusion in order to have a reasonable balance between all chapters of the manuscript. In this regard, we rephrased the abstract as well so that our main aim of the manuscript is better conceived.

In consideration of the above general remarks on quality and deficiancies , I suggest moderate revision of the manuscript before publication.

**Specific points:**
Below are our responses to your specific points:

**(1)** Line 12. Replace "accuracy" with "precision estimate". Note that in line 13 you correctly assess the "accuracy" with the blast location test.
Agree – changed

**(2)** Line 15. Delete the rest of the sentence after:" ..1.7 km in depth."
Done

**(3)** Line 27. Replace "accuracy" with "precision"
Agree – changed

**(4)** Line 48. "... has the advantage of being" largely "independent …"
Done

**(5)** Line 51. "best model(s)." a note on ambiguity would be useful
This part is rephrased to:
"Moreover, the results can be statistically analyzed, and thus errors and ambiguities can be estimated. The method extends the probabilistic relocation approaches (Lomax et al., 2000) by inverting for a set of velocity models well explaining the data. "
However, more details are provided in the method chapter of the manuscript

**(6)** Line 64. Please outline Adriatic microplate in one of the Figures.
Done

**(7)** Lines 67-75. Needs a figure to show the strain if you keep the introduction as is and the chapter 6.3.
We think that we have already provided all references to our statements regarding the strain (mainly in section 2). Nevertheless, now the companion paper by Verwater et al. 2021 is

submitted (and accessible) in which this topic (including a Figure) is covered in more detail. Therefore, we prefer not to add an additional Figure to our paper. We add a reference to Verwater et al. 2021 to Section 2.

**(8)** Figure 1. See point 6 above. Red box in bottom figure does not correspond with bounds of upper figure. This figure is not providing all necessary tectonic information mentioned in the manuscript. You should note that the seismic catalog presented by ISC is by far not complete down to magnitude 2. If you want to show the big picture use either EMSC catalog likely complete to M3 or ISC likely complete to M3.5. Otherwise you could use a composite of the various national catalogs that probably are complete to M2.5.
The figure bounds are corrected.
All the information which is mentioned in the text is now added to figure 1.
The big picture of seismicity (Figure 1) is now taken from EMSC with M3.

**(9)** Line 85. Actually there are earlier catalogs that were compiled: European Geotraverse Blundell et al. 1992, Solarino et al. 1997
Agree – two references are inserted in the text.

**(10)** Lines 115 to 119. You need to elaborate in detail (this can be done in supplementary material but it is absolutely necessary to have this information) how you identified the "common" events and how in the end you established the event list of the 2619 events.
An appendix A with the information regarding how the event list is formed is added to the updated manuscript.

**(11)** Lines 120 – 126. The discussion of the results of this semi-automated picking (that is well described in suppl.) needs to be more extensive and detailed. On what basis did you define the selection criteria (gap<200 , why not <180') (RMS <1s), why no mentioning of number of P obs? Did you check all 12534 P obs manually?
We used the origin-time of 2,639 local events from national catalogs (now in appendix A) as initial data to start the automatic picking with. However, we don't have any information about the precision of these events beforehand. Therefore, in the beginning, we decided to select as many events as possible from the national catalogs, apply an automatic picking procedure for these events, and then make a further selection based on our own picks and location information. After the automatic procedure (2,639 events, 68,099 P- and 17,151 S-Picks), we noticed that many of the events are either on the periphery of the network, too weak, or too noisy to be detected by more than 5 stations. Moreover, we believe that the automatic picker missed some of the good picks or introduced some suspected picks. Therefore, we had an early selection criterion (not too conservative) in order to do a manual/visual inspection of the picks. These selection criteria (gap<200 and RMS <1s which gives 384 events with 18,390 P- and 7,762 S-picks) were applied to the automatic picking results.  We manually/visually inspected the P and S picks of these 384 events, with careful consideration for picks in the distances of the triplication zone and farther than that (we inspected a very large amount of the picks from these 384 events, maybe 90% of them). We explain later in text that from these 384 events, some are blasts, and some are unclear to us. Later on, this dataset was further selected for simultaneous inversion (301 local earthquakes with gap<180° and minimum 10 P-picks and 5 S-picks).
We added the statistics about the number of picks and their quality classes of 2,639 events after automatic picking to Appendix B (Table B1) of the updated manuscript. Detailed

information of the selected dataset for the simultaneous inversion is now given in the main text (section 3.2; table 1 and figure 4) of the updated manuscript.

**(12)** Line 134-6 and Figure 3. The Wadati diagram shows significant numbers of observations with +/-3s residual relative to constant Vp/Vs ratio. Note the the Vp/Vs ratio varies within the crust and at Moho. You may see this in the Figure as the straight line is systematically shifted onto the side of the highest point density after about 25s P travel time. What residual range do you define as corresponding to the sum of 3D, lithologies and regular Gaussian observation uncertainty effects and what value denotes an outlier?

We agree that the points are systematically shifted to above the straight line indicating that the Vp/Vs ratio (slightly) increases at larger distances. This could be related to different (average) Vp/Vs ratios in the crust and upper mantle (which is e.g., also suggested by global models such as ak135). We mention this now in the text: "We notice that at P travel-times larger than ~25 s the observations tend to slightly larger S-P travel-time differences, potentially indicating a higher Vp/Vs ratio at larger depth, i.e., in the upper mantle."

According to own synthetic tests (using the source and receiver geometry of this study and including (1) a Moho topography based on a simplified Moho from Spada et al., 2013 (see section 5), (2) assumed (moderate) differences in average Vp/Vs ratio in the crust and upper mantle, (3) crustal Vp/Vs ratio anomalies which we expect for the region (e.g., Vigano et al.,2015) and generally from other regions of the world, and (4) picking uncertainties) we expect a scatter of no more than ± 2 to 3s around the linear trend of the points in the Wadati diagram. We would like to emphasize that we only considered values with $t_s$-$t_p$ > 0.72 $t_p$ ± 4s as "outliers" and removed them. These outliers were only 0.3% of all observations (which should not make a difference even if they were incorrectly flagged as "outliers"). We state this now more clearly in the text. After this removal, only 2% of the whole observations have $t_s$ – $t_p$ > 0.72 $t_p$±3 s.

**(13)** Line 144 and Figure 4. "Checking the phase-type is extremly important." I fully agree and record section display is a good zero-order approach. However, I do not think your figure 4 is of help for doing this. How realistic is the ak135 global model for P phase identification considering the Moho topography by Spada et al. 2013 (your Fig.5c) or the 3D LET model by Diehl et al. 2009 but most important the literally more than a dozen refraction seismic lines that have been published (for a review see Kissling et al. 2006).

We agree. The comparison of the cumulated picks (from earthquakes with a variety of different earthquake depths) with the bounds of the synthetic traveltime curves (from a global model for different earthquake depths) was too coarse, too unspecific, and obviously not helpful. As we mentioned in the text (both in section 3.2 and in the appendix), we visually/manually inspected the waveforms of all earthquakes and their automatically determined arrival-times, and added, deleted, or modified picks where applicable. This inspection was done mainly on individual traces and was most helpful for identifying wrongly picked noise burst etc. Arrivals of stronger events, for which we expect XmX or Xn arrivals, were additionally assessed with the help of record sections (epidistance plots of individual earthquakes; based on preliminary locations). So, we are quite confident, that we identified most of the phases correctly. Nevertheless, we acknowledge - and are very much aware of - that even these detailed visual/manual checks cannot prevent some amount of wrongly identified picks in our dataset (a general difficulty for manual picks e.g., already pointed out by Diehl et al., 2009). We state this now more clearly in the text. In this context, we think that Figure 4 of the original manuscript is indeed not helpful for the manuscript and thus we decided to delete it, as we deleted parts of the text of section 3.3 in the original manuscript (some parts were moved to the newly formed section 3.2). In any

case, we would like to emphasize that we never used Figure 4 for identifying and removing individual picks (potentially misidentified).

**(14)** Line 152. Please provide clear evidence and explain in detail strategy to identify PmP phase by using a totally inadequat 1D model.
We agree, based on this plot we are not able to do this. We deleted this statement (together with the whole paragraph) – please see our comment above.

**(15)** Lines 152/3. "the number of outliers, ..., is not significant to the total number of picks." What value do you define as being an outlier? (analog question to point 12) Note there are dozens of observations +/-2s from the main intensity of data points (that by the way is totally off your Pg line) and that individual hypocenter location precision (and even more important for accuracy) is in truth measured by the fit of just those observations that refer to the specific event.
Yes, we agree on this point. Our discussion of "outliers" was not specific enough. Similar to the phase-identification issue (see our comment above), we are very much aware that our dataset probably still contains some number of "outliers" (howsoever defined) and we stated this now more clearly. However, as we also state above, we did not use Figure 4 or any related (automatic) mechanism for identifying and removing individual picks (potentially misidentified). We have just used it as a – probably too simple – statement regarding the existence or non-existence of "outliers".
As stated above, we deleted the whole corresponding paragraph (and Figure 4) and modified the text.

**(16)** Line 218. "... does not depend on initial hypocenters, ..." I seriously doubt this (does not depend) and suggest to phrase it differently. Consider how you would identify an outlier with Figure 4 if you do not have a rather good idea the initial hypocenter! Furthermore, consider that you were using a priori information from existing catalogs for your semi-automated picking and that even with all this information you apparently found mispicks and had to select the 384 best events.
We only partly agree. We are still convinced, that our inversion method itself does not depend on the initial hypocenters (as is the case for the traditional inversion routines using a linearized approach, DLSQ inversion, etc.), see Fig. 8 in Ryberg & Haberland, 2019. However, we agree that some dependency might be introduced through the selection of picks (e.g., "outlier removal"), especially in the case of automatic picking (involving e.g., sequential intermediate location steps based on initial velocity models). Nevertheless, we think that even this influence is minimized because we are using a visually inspected pick-dataset.
We modified the part in the following way:
"Therefore, the McMC method only uses the travel times and does not directly depend on initial hypocenters, origin times, velocity models, or even the model parametrization (e.g., grid node spacing). Nevertheless, because in the (semi-automatic) picking procedure the selection of picks is involving the comparison with travel times based on preliminary hypocenters (which depend on initial velocity models), the results of the inversion depend *stricto sensu* to some degree on initial values."

**(17)** Line 226. This does not provide an "accuracy" estimate! May be internal consistency, precision.
Agree - It is changed to consistency.
And you need to provide reasons why this should be expected to be of relevance for the real individual

event locations.

The event locations (and the velocity model) derived by the inversion procedure depend on a large number of parameters such as the quantity and spatial distribution of earthquakes and receivers, noise, quality of travel-time readings, the class of the model used (in the inversion) and so forth. Furthermore, the test is not only interesting in respect to the hypocenters but also to the derived velocity model. We think that it is appropriate to test the recovery by synthetic tests as it is standard for the recovery of subsurface structure (e.g., in LET studies). Since the McMC approach for solving the coupled hypocenter-(1-D) velocity problem is quite novel we think that this kind of test is particularly interesting. We already described our reasoning for conducting the test in section 5, however, we added/modified the text slightly. Furthermore, we added some sentences to the discussion of the results of the test.

"Comparing input event locations (synthetic) and the inverted (output) ones allows us to study the recovery of the hypocenters, location consistency, and potential systematic errors related to the use of a 1-D model, which we can generally expect for the derived real hypocenters. For example, it can be studied whether events at the periphery or in certain parts of the model have systematically larger uncertainties (e.g., due to their location and/or spatial distribution of picks). Furthermore, we can study how the (output) 1-D model looks in comparison to the (input) 3-D model, how large the derived noise is in relation to the synthetic input noise and how the pattern of station-corrections corresponds to the shallow velocity anomalies. Similar tests are standard in structural studies (i.e., LET) to study the recovery of certain features. "

By the way, your blast test shows otherwise!

We think that the simultaneous/joint hypocenter relocation is a very powerful concept. We agree that our blast test indeed shows very good performance of this test.

**(18)** Line 230. This statement about Moho velocities is simply wrong. No 7km/s velocity has been reported in the Alps. Please check the literature.

While the reviewer is certainly right regarding the reported seismic velocity values in the Alps, we would like to point out that this is a synthetic test with the main aim of checking whether and to which extent a 3-D velocity variation (i.e., Moho topography and shallow velocity variations) influences the recovery of the hypocenters (and which 1-D model is derived). We think that the exact values of the velocities are not that important for this test as long as they are in the "usual" range. We used also for example an angular shape of the Tauern Window and the sedimentary basins which is also not close to reality. We never claimed that these are reported values, it's a description of our synthetic model. Therefore, we prefer to leave the synthetic model as it is. Nevertheless (and in response to issues 17 – 21), we modified the whole paragraph describing the synthetic model.

**(19)** Line 232. The 19km are just along the flank of the Ivrea body and not relevant for the Moho topography beneath the Po plain.

Agree - changed

**(20)** Line 233. -5km is much to high and you are doing something wrong if you need to avoid rays through air by such model top elevation. Note that the increase in pressure within the earth causes a velocity increase with depth and the seismic waves to show a downward curvature You should use the average station elevation for ray tracing.

During the McMC inversion many different – also weird and unrealistic - velocity values, i.e., also models with a velocity increase with elevation, are (randomly) tested. This is totally different e.g., compared to velest. We are very much aware of the fact that usually on Earth there is a velocity increase with depth. These unrealistic models obviously have to be suppressed

because otherwise, we obtain a somehow "mirrored" (artificial) velocity model above the surface (in our case, only models without low-velocity zones are accepted). Furthermore, the method does not use rays (or raytracing) at all but instead calculates the travel time field with an FD Eikonal solver. More information can be found in Ryberg & Haberland (2019) (exact reference in the bibliography of the manuscript).

**(21)** Line 245. Choosing a constant Vp/Vs ratio of SQR(3) is very problematic as we know it is wrong because it varies and likely the average is different.
This is a synthetic test, and the goal is mainly to assess the performance of the method to recover the earthquake locations when using a simplified 1-D velocity model in the inversion. We agree that the reality is more complex and maybe our test is not complete and might not cover all aspects which should be checked for. In Ryberg & Haberland (2019) a much simpler recovery test with a 1-D synthetic model was performed, now we wanted to add some more complexity (although not claiming that this is the ultimate test). We modified the section describing the synthetic model (please see also our response to issues 17 – 20 above and 24 below).

**(22)** Line 250/1. Not only refer to table but provide correct value here.
Done

**(23)** Figure 5. Your model extent in Figure 5a does not correspond with your map extent in Figure 5c
AND either of these extents differ from your study region shown in Figure 1 AND all of these are different than your Figure 2. Make certain you everywhere show the same study region extent, if you want to show more area around, then mark the study region.
Done

**(24)** Figure 5b. The Moho topography is wrong. There is a Moho offset across the plate boundary but you show a vertical Moho interface!
Yes, the reviewer is right, the Moho shown in the Figure and used in our synthetic test is only a very simplified version of the Spada et al. 2013 Moho (Moho offset, double Moho, etc. are not reproduced). Furthermore, the crustal structure is very much simplified. We state this now more clearly in the text. However, we still think that simplifications for the synthetic test model are acceptable, and a simplified synthetic model is still useful for studying the recovery of some features (hypocenters, 1-D model).
Please see also our comments on similar issues 18 and 21.

**(25)** Lines 259/60. "average uncertainty of 240m in longitude, 270m in latitude ...." How did you determine that? In such way this information is not usefull. With what probability do we have what location uncertainty for any single hypocenter?
In the previous paragraph, it is explained that the model parameters are defined based on the average and standard deviation of the final models. In the old version of the manuscript, we showed in figure 6b the histograms of the uncertainties (1σ) of all earthquakes. However, for avoiding any misunderstanding, we decided to not mention these uncertainties here and only talk about the misfits which are important for recovery assessment.

**(26)** Figure 7. Move to supplementary material.
We think figure 7a of the original manuscript is quite important to display in the main text because it is the best information showing the recovery of the synthetic test. Therefore, we

removed figure 6b (not as important as Figure 7) and merged figures 6a and 7 into one figure (in the updated manuscript it is figure 6).

**(27)** Figure 8. Again a DIFFERENT STUDY REGION SHOWN!.
The figure frame is now set correctly.
What about the stations to the West of the Tauern window (as example, there are other regions with no visible symbols)? Do they have all zero station delay values or did you not obtain any values for them?
There are stations that have very small delays and it's probably hard to see them (they are like a point in the figure). The stations with zero delay are indicated now with a different symbol in the updated manuscript. There are also some stations without any delays, which were considered in the early part of the study but not included in the final inversion (e.g., no high-quality picks, station problems, etc.). However, all the stations to the west of Tauern Window have symbols (comparison between figures 2 and 7b of the updated manuscript).
Please explain in more detail what the velocity-depth function shows. In may view, it documents the data set is not capable to resolve the velocity structure below 30km depth and certainly not the Moho. This does not come as a surprise as it is well known that you loose vertical resolution below your deepest hypocenters.
We totally agree that resolution typically degrades below the (deepest) earthquakes. However, we would like to point out that the synthetic (3-D) velocity model in the depth range between ~30 and ~55 km is characterized by a strong Moho topography, which *per se* cannot be exactly recovered by a 1-D model (used in our inversion). We think that is not (only) an effect of the data (less ray coverage and resolution below the deepest earthquakes) but also of the inability of a 1-D model to capture a 3-D variation. So, the input model is indeed not well resolved in this depth range, but it is most likely an effect of the data and the much simpler model class used in the inversion. Within the model complexity (in each depth level) the input model is recovered in average (with some variability indicated by the standard deviations). Of course, upper mantle velocities are reached at a large depth. We think this is a nice result of our test which also sheds light on the interpretability of such 1-D models. We modified the text and hope that this issue is somewhat clearer.

**(28)** Lines 297 to 304. This model discussion is inadequate with regards to the previously published information about the crustal structure. If your model does not allow to resolve it, then say so and it is OK. But do not claim it is in agreement with prior independent knowledge if it is obviously not.
We agree. For example, our model is relatively close to the Diehl et al. (2009) model in the upper part, however, at larger depth it is different. We describe this now better. Furthermore, the resolution of our model is only good down to about 45km depth, below that the standard deviations (1 sigma) are getting significantly larger indicating fading resolution. Also, this aspect is now described in more detail in the updated manuscript.

**(29)** Line 320. "it proved to be useful for accurately localizing earthquakes ..". I am missing the prove. Please explain how this was proved.
Our phrasing was obviously misleading. We wanted to add a more general statement that - although it might be difficult to interpret the station corrections in a straightforward ("physical") way - the whole concept of the simultaneous /joint inversion (including the inversion for these station corrections) is very powerful. We reformulated this part in the text to:

"Nevertheless, the general concept of simultaneously inverting local earthquake datasets for a (simplified) 1-D velocity model, hypocenter positions/origin times, and station corrections proved to be very powerful for accurately localizing earthquakes (see e.g., Kissling,1988)."

**(30)** Figure 10. Figure 10b I would again derive the conclusion from this figure that you are lacking resolution power below 30km for Vp and below 20km for Vp/Vs ratio.

The resolution for Vp is starting to fade below 30km depth, however, standard deviations of around 0.02 in the depth range between 30 and 40 km (yellow ranges in Figure 9 of the updated manuscript) still indicate fair resolution. For Vp/Vs, we see this limit at around 30 km. We write this now in the text.

Figure 10c. Now this looks like the study region. Why not always marking this extent where you do have data from?

We homogenized the lat/long in all the maps.

How do you interpret the distribution of the station delays?

We discussed the pattern of station corrections and possible qualitative relations with geological features in lines 313 – 318 of the original manuscript.

Note that there is a single station delay strongly different from all others within its vicinity located near 11.7E/47N. If I obtained such result I would check if it is real or caused by bad data of some sort. Note that otherwise your largest station delays are all within the periphery of the study region as this is well known from minimum 1Dmodel applications.

Yes, the reviewer points out a totally valid point, thanks. We checked the data and identified this station as having wrong picks. Since we generally have to expect influences of these wrong picks on all other parameters in our simultaneous inversion (velocity model, other station corrections, and hypocenters), we removed all picks of this station, reran the simultaneous inversion, and located again all events with the updated velocity model. So, Figure 9 (of the updated manuscript) shows now the updated data. Please note that the changes in the hypocenters and station corrections were very marginal, and all inferences in the paper are not changed.

**(31)** Lines 325 to 335 and Figure 11. What hypocenter depth did you test these blast locations with?

How do you define the mis-location vector? Relative to the center of the quarries or do you know the precise location within the quarry for each blast? This accuracy test shows that your previous precision estimates were a bit to optimistic but the accuracy is still very good.

We appreciate this question. It motivated us to calculate the misfit vectors for the blasts similar to Husen et al. (1999) using the centers of the quarry areas (since we do not have the exact blast locations). We, therefore, added the following text to the manuscript:

"Based on the average mislocations of the blasts (relative to the centers of the quarry areas) and also their location uncertainties, we estimate the absolute location errors in the range of 1 km horizontally and 500 m vertically."

**(32)** Line 331. Please explain in theory why you suggest that including S observations improve the hypocenter location solution?

According to Gomberg et al. (1990) S-picks provide powerful additional constraints on hypocentre location, especially regarding the focal depth. They demonstrated that the partial derivatives of S-travel-times are larger than for P-waves and that they act as a unique constraint especially in cases of S-picks recorded within 1.4 focal depth's distance. However, precisely measuring the S phases is absolutely necessary and we carefully and conservatively identified the first arriving S-picks on the 3C high-quality recordings. Nevertheless, we think that our

statement in Line 331 (of original manuscript) was not really precise and we decided to delete the statement regarding the S-picks. We replaced it by:
"It is expected that the errors of earthquakes (which are potentially deeper than blasts) are smaller than those estimated from this test because they are less affected by the heterogeneous shallow structure which was poorly accounted for by the model (Kissling 1988 & Husen et al., 1999)."

**(33)** Lines 340/1. Your absolute depth uncertainty has been documented by your ground truth accuracy test (Fig.11) to be a few km and the epicenter location uncertainty is about 1km. Please correct your numbers.
The location uncertainties that we report here are the standard deviations from the mean value of all the models of McMC. We provided now new estimates for the mislocation (based on quarry blasts test; see point 31) that we can now mention here as well. We rephrased the sentence to:
"Based on statistical analyses of the McMC inversion results, the average epicentral and depth uncertainties (1σ) are ~500 m and ~1.7 km which are compatible with precision estimates by the synthetic test (Sect. 5). However, the absolute location errors estimated by quarry blasts test (Sect. 6.2), are 1 km horizontally and 500 km vertically. "

**(34)** Lines 343/4. These differences are indeed significant. However, as it just regards a selected best event data set with on average 36 P observations this is comparing apples and grapes. Your data set is excellent for seismic tomography but absolutely not representative for a seismicity catalog (magnitude of completeness? Just 16 months). On the other hand, the national seismic catalogs contain many poorly locatable –or you could also say difficult to locate- events that need extra processing time to obtain a complete catalog and there is the significant difference in number of stations. I believe it would be useful to discuss thourougly these difference in addition to presenting just the numbers.
Our intention was not to show poor hypocenter solutions from the national agencies. Most of the large differences between location of the blasts reported by the agencies and our solutions (and the mislocations) are indeed related to the amount of data (stations). We added the following paraphrase to the text:
"Since the national agencies are (probably) using much less data for the location (smaller number of stations used, larger inter-station distances), a significant difference between their hypocenter solutions and those obtained in this study is expected (average of 2.4 km in epicenter and 3.7 km in depth). The earthquake depths calculated by McMC are systematically shallower than those by national agencies (by an average of 1.1 km). The maximum and minimum differences in the epicenters and depths (between McMC and national catalogs) are seen for the earthquakes from the INGV and SED, respectively.
The derived hypocenters in this study do not represent a representative seismicity catalog of the region (the national catalogs contain also many small, poorly constrained events in a much longer period) but form excellent data for further seismological studies e.g., Local Earthquake Tomography (LET). Moreover, this highly precise hypocentral data allows further tectonic inferences."

**(35)** Figure 12. Figure 12b is not needed, just provide the uncertainty estimates. Note that for cluster interpretation relative hypocenter location uncertainty estimates (and that is what you obtain with your Markov chain MC inversion of the coupled problem that includes a joint hypocenter determination approach) are most important while for absolute location obviously the accuracy is key.

We agree and delete this part of the figure. We now mention the uncertainty values in the text.

**(36)** Figure 13. For seismotectonic interpretation of clusters along a fault system, you should definitely
employ focal mechanisms.
We totally agree. In the discussion, we wanted to compare our derived seismicity, which is now captured in a consistent way throughout the study area with a dense network, with the seismicity known from other studies and datasets (e.g., from permanent, though coarse, networks running for many years). In this context we also discussed some occurrences along fault zones, however, we did not go into too much detail here. We are aware that for a conclusive investigation, focal mechanisms as well as a more complete catalog is necessary. Within the research program, more studies are underway focusing on exactly these issues. In the meanwhile, a manuscript (Petersen et al., 2021) was submitted (to SE) showing focal mechanisms from the AlpArray and SWATH-D networks. We refer to this study now in the text, however, results were not available at the time of initial submission of our manuscript.

On behalf of the authors,
Azam Jozi Najafabadi

---

## Author Comment (AC3) · 25 Mar 2021

Dear Referee,

We would like to thank you for your thoughtful comments on the manuscript. Your valuable feedback will help us to improve the quality of the manuscript. Following, please find our response to your comments as blue texts.

The present manuscript by Jozi Najafabadi et al. presents the compilation of a seismicity dataset that will, I presume, eventually be used for a local earthquake tomography study of the Eastern Alps. Using recordings from the dense SWATH-D deployment, they present a careful procedure of obtaining and verifying arrival time picks, derivation of optimal hypocenter locations and a best-fit 1D velocity model. The Bayesian approach for the inversion of hypocenters, velocity model and station corrections is something new, and the present manuscript provides a nice case study for its application. Lastly, the obtained hypocentral locations are compared to mapped faults, from which the apparent activity or non-activity of a number of structures in the Eastern Alps is inferred. This last part is where I see some potential problems that will make some changes to the manuscript necessary. Overall, the paper is well written and definitely of interest to the readership of Solid Earth and the special volume "New insights on the tectonic evolution of the Alps and the adjacent orogens". I recommend moderate revisions before publication.

General comment:

In section 3.2, it is briefly mentioned that the events for which arrival times on SWATH-D stations were obtained and that were then relocated, used for deriving the 1D velocity model etc. were selected from a synthesized catalog that was based on the bulletins of national agencies. For the tectonic interpretation to be viable, this part needs to be made much more transparent. While it is fine to choose a subset of events based on network criteria when working towards a tomography study, it is a completely different thing when the activity or (more crucial) non-activity of faults is inferred from such a subset. In clearer words: the authors need to convincingly show that their chosen subset of events is representative, and does not systematically miss events from certain regions. I suggest to provide a map with all 2639 events from the different national catalogs, in which the chosen 384 are marked. It would likely be even better if the station distribution from the different national networks could be shown as well. I also suggest to better describe the reasoning behind this approach of choosing a subset of events from national bulletins. Are the national bulletins complete enough that one can exclude that the dense SWATH-D network contains signals from small, previously undetected events? Or was the focus on the larger events that would generate arrival times at a larger number of stations?

You are correctly pointing to the main aim of our work which is local earthquake tomography, and this is the reason to focus on creating a catalog with the most consistent earthquake hypocenters (and corresponding travel time picks). Our hypocenters have the highest precision in the region (at this time) and our clustered seismicity agrees very well with seismotectonic characterizations based on long-term records. Therefore, our hypocenters can shed light on active faulting and enables us for a convincing tectonic interpretation.

We totally agree that the dense SWATH-D network will make it possible to detect very small earthquakes not contained in the national catalogs. However, for the purpose of locating the earthquake with high accuracy - and in turn using the dataset (hypocenters, travel-time picks) for local earthquake tomography - we were mainly interested in earthquakes of some magnitude so that they are observed by a number of stations (see section 3.2 of the updated manuscript). We assume that these earthquakes are to a very large extent listed in the national catalogs (example: minimum magnitude -0.8 by ZAMG for the Austrian part of the study region).

[Figure]

Here is a map containing all 2639 events from the national catalogs and the chosen 384 events are indicated by white dots. Unfortunately, we have no information regarding the stations for the national catalogs. The number of figures in our article is already rather large, thus we would like to not include this figure in the manuscript. The events by national catalogs are publicly available.

I agree with a previous reviewer that fault plane solutions would be nice to have for a more detailed tectonic interpretation. However, I can see that the main aim of the manuscript is the description of the dataset that will be used for tomography, and speculate that the tectonics part was added mainly for the sake of the Special Volume topic. I believe the careful derivation of the hypocenters and velocity model, using a rather novel approach and performing many quality checks, is in itself enough material, so that a deeper-going tectonic interpretation employing focal mechanisms is not strictly necessary here,

Thank you very much for this comment. In the meanwhile, a manuscript dealing with the focal mechanisms was submitted to Solid Earth (Petersen et al., 2021). We mention this now in our manuscript and added the reference.

Specific comments:

**l.25:** Why were only data from 2017/2018 used when the stations ran into 2019? Should be mentioned with a word or two.

At the time of doing this analysis, the data from 2019 was not available. We will include an updated dataset in the LET study.

Also, mentioning the total number of SWATHD stations here could be useful, especially since the number of AlpArray stations that were also used is brought up in the next paragraph.

Done

**l.33/34:** "to identify the status of the seismically active volume...". This is a strange formulation, and should be changed.

Modified to:

"... to identify the general pattern of seismicity on the surface and at depth"

**l.36:** remove the

Done

**l.55ff:** I would recommend to use fewer abbreviations, this is making the manuscript unnecessarily hard to read. Best limit abbreviation to a handful of terms that really show up a lot throughout the manuscript, and write out the rest (this is maybe also my personal taste...).

We actually use these abbreviations several times in the manuscript, especially in the section "Results and discussion", and also in the figures. We think that using the whole word makes the sentence long (in some cases) and also reading the figures alongside the text would be more difficult!

**l.82:** this bracket is not closing again

Agree - corrected

**l.91:** stuck should be struck; corrected

about what time interval are we talking for the ML>6 earthquakes? Last decades, centuries, millennia?

"in the last several centuries (Slejko, 2018)" – This is added to the text

**l.96/97:** This statement is problematic, because while the present study is using a denser seismic network, the chosen approach of using a subset of events from agency bulletins (see General Comments) makes it impossible that previously missed events (if they exist) will be detected. Thus, the present study can do nothing to address the problem that is hinted at here (inactive region maybe because not well instrumented).

Agree – The improved coverage of the region with seismic stations was one of the main aims of the deployment of the dense SWATH-D network, however, the issue towards a more complete catalog is not directly covered in our study. Other studies dealing with this issue are underway, but results are not yet available. Accordingly, we think that this statement is not necessary here and thus we deleted the last two sentences.

**l.105:** The "however" doesn't fit here

Agree - corrected

**l.111:** remove "stations"

Done

**ll.115-119:** For me, this paragraph is the main problem of the manuscript as is. At least for the tectonic interpretation part, the authors need to convince the reader that no selection bias of earthquakes exists, i.e. that regions interpreted as aseismic based on the chosen 344 events are also aseismic if one looks at the entire >2600 events in the original database (see recommendatiosn in General comments). Also, the statement here seems to indicate that the national bulletins were deemed complete, which stands in contrast to l.96/97.

Our statement in lines 96-97 (which is eliminated in the updated manuscript) was related to individual seismological studies in the region (lines 85-86 of the original manuscript), and not national bulletins. However, if we consider the national bulletins with 2,639 events to be complete, it also indicates rare seismicity in the aforementioned region. We are not claiming for a complete earthquake catalog, but for a high-precise catalog of seismicity which agrees very well with previous studies in the region and also national bulletins.

**l.135/136:** how were outliers defined, and where can I see outliers in Figure 3 (can they be marked?)

All the observations that have $t_s - t_p$ >0.72 $t_p$ ± 4s were considered as "outlier" and removed from the data which formed only 0.3% of whole observations. After this removal, only 2% of the whole observations have ts – tp > 0.72 tp±3 s.

Based on experiences with similar datasets and according to own synthetic tests (using the source and receiver geometry of this study and including 1- a Moho topography based on a simplified Moho from Spada et al., 2013 (see section 5), 2- assumed (moderate) differences in average Vp/Vs ratio in the crust and upper mantle, 3-crustal Vp/Vs ratio anomalies which we see (in respect to spatial dimension and amplitude) both in our first tomographic results and in other regions of the world, and 4- picking uncertainties) we expect a scatter of no more than ± 2 to 3s around the linear trend of the points in the Wadati diagram.

More detailed information is provided in response to point (12) of Reviewer 2.

**l.138:** If a part of the goal audience are people mainly interested in the activity of structures in the (south)eastern Alps, the three phases and the triplication distance should be briefly explained, e.g. in a brief sketch that could be added to Figure 4. Also, giving an estimate of the overtaking distance, e.g. with the crustal thickness and velocity given in ll.147/148, would be beneficial.

Motivated by the critical questions and comments regarding Figure 4 by reviewer 2, we modified the text dealing with the phase-type identification considerably. By doing this, we also deleted Figure 4 which seemed not helpful in this context. Please see more details on this issue in the response to reviewer 2.

**l.140:** be (remove ing)

Done

**l.158:** is indicated the right word here?

Agree – the sentence is modified to:

"… a simultaneous inversion for hypocenters and velocity structure (and/or station corrections) is needed."

**l.160:** Well, a Bayesian-type approach has been used for all these geophysical studies. As it is written, it sounds like this was always the exact same approach (which it wasn't)

We agree. We modified the text slightly:

"Different to the conventional approach of damped least squares, we use a Bayesian approach (Bayes, 1763). Bayesian approaches have been applied in a number of geophysical studies (Tarantola et al., 1982; Duijndam, 1988a,b; Mosegaard and Tarantola, 1995; Gallagher et al., 2009; Bodin et al., 2012a,b; Ryberg and Haberland, 2018). Ryberg and Haberland (2019) recently implemented a hierarchical, transdimensional Markov chain Monte Carlo approach for the joint inversion of hypocenters, 1–D velocity structure and station–corrections for the local earthquake case."

**l.176:** structure (-s)
Done

**l.212:** reformulate that first sentence Section 5: I am not completely sure I understand the reasoning behind this test. The authors construct a first-order 3D velocity model of the Alps based on published data and perform a retrieval check using the real data (hypocenters, stations) as input. Thus any misfit in the output should stem from 3D structure and/or general uncertainty, but only with the assumption of this specific 3D model...since the true 3D structure of the Alps will almost certainly differ from the utilized model (presumably only to secondorder differences?), do we have any idea if the retrieved 3D effects are similar for a (subtly, substantially) different reality? I think the paragraph needs a more detailed description about the purpose of the test, what it is supposed to show and what it can not show. Nevertheless, I appreciate the effort that went into performing it!
With this test, we want to study how (well) the synthetic hypocenters (and the velocity) model are recovered by our inversion routine. We think this is interesting because we use a simplified framework in the inversion (i.e., 1-D model, station corrections) obviously not capable to capture the full 3-D conditions which we expect for the Earth. For this, we designed a 3-D model (based on existing information) which resembles some first-order features in our study area such as Moho topography or shallow low-velocity sedimentary basins. Furthermore, we can study how the (output) 1-D model looks in comparison to the (input) 3-D model, how large the derived noise is in relation to the synthetic input noise and how the pattern of station-corrections corresponds to the shallow velocity anomalies. In inversions for (3-D) subsurface structure (i.e., tomography) people regularly employ synthetic recovery tests.
In order to make this clearer we modified the text. Please see also our comment regarding the issue (17) of Reviewer 2.

**l.268:** Figure (-s)
Corrected

**l.271:** reformulate "rather slight", add km after 50
The sentence is reformulated to:
"it is modeled by a gradual increase of the velocities at depths from 30 to 50 km"

**l.275:** reformulate "fluctuating"
"fluctuating" is replaced by "irregular". And by the way, this sentence is moved to section 6.1 where we talk about velocity models of real data.

**l.319:** "contains an overlay site effect" I'm not sure I understand what exactly is meant here.
The sentence is reformulated to:
"contain a superposition of site effects and/or effects from 3–D structural variations"

**l.324:** This is not really a sentence

Agree – There was a mistake in the formulation of the sentence. It's reformulated to:
"To validate the localization procedure, the detected 15 quarry blasts (based on manual/visual inspections; see 3.2) were independently relocated by McMC …"

**l.332/333:** Can hypocenters from the INGV/ZAMG catalogs also be shown in Figure 11, to better illustrate the improvement in hypocenter location?

The large differences in the location of the blasts reported by the agencies to our solutions (and the mislocations) are most likely related to much less data (stations), so it is not surprising that they are different (and show some mislocation). We think that the differences in location are already mentioned in the text. Furthermore, in order not to overload the (old) Figure 11 we prefer not to add the hypocenters to the Figure.

**l.344:** it would be interesting to elaborate a bit more on these differences; is there a trend, eg. with bulletins showing systematically larger or smaller depths?

In response to this comment and also to reviewer 2, we added the following paraphrase to the text:
"Since the national agencies are (probably) using much less data for the location (smaller number of stations used, larger inter-station distances), a significant difference between their hypocenter solutions and those obtained in this study is expected (average of 2.3 km in epicenter and 2.9 km in depth). The earthquake depths calculated by McMC are systematically shallower than those by national agencies (by an average of 1.1 km). The maximum and minimum differences in the epicenters and depths (between McMC and national catalogs) are seen for the earthquakes from the INGV and SED, respectively.
The derived hypocenters in this study do not represent a representative seismicity catalog of the region (the national catalogs contain also many small, poorly constrained events in a much longer period) but form excellent data for further seismological studies e.g., Local Earthquake Tomography (LET). Moreover, this highly precise hypocentral data allows further tectonic inferences. "

**l.345:** This is not surprising, since no search for new events beyond those in national catalogs has been performed

We agree with your point. However, other seismological studies in the region (based on local networks or national agency data) confirm a similar seismicity pattern. Although, considering high-accurate and high-constrained hypocenters throughout the region in our study, a comparison with national catalogs could be meaningful too.

**l.358:** mention the magnitudes of these events
Done

**l.398:** See General Comment: how well can one argue for the absence of seismicity ("seismic gap") based on a catalog that was only a choice of 384 out of 2639 events? A map showing where the remainder of events (those that were not chosen) were located is essential if such a statement is attempted

In the updated manuscript, we emphasize that the seismicity pattern of our study is similar to long-term seismicity by, e.g., Reiter et al., 2018, or seismic catalogs such as the SHARE. Furthermore, we always (for every sub-region) compared our seismicity with previous long-term seismicity studies. Our seismotectonic interpretation is based on the high-quality hypocenters

that are derived in our study. However, we confirmed the sparse seismicity in part of the region with other studies and thereupon made an interpretation.

**l.405ff:** spelling: Engadin vs. Engadine Fault
Corrected

**l.427ff:** These (at least the first three dotpoints) are not results but the analysis steps that were carried out to retrieve the results. Either only results should be listed, or two separate listings for analysis steps and results are needed.
The conclusion is totally rewritten. Please look at the updated manuscript.
In the name list of AlpArray people, all those containing special characters have formatting issues (LaTeX syntax?)
Corrected

**Figures:**

Figure 1: Typo in Tectonic units legend (forland should be foreland)
corrected

Figure 2: The fault lines in this plot are really hard to see. Choosing a larger linewidth would be helpful. A color scale for the topography would also be nice.
Both done

Figure 4: Please be more specific about what the "various depths" are that were used to obtain the travel-time corridor. Then, I do not see red dashed lines in the plot (as mentioned in the caption). Lastly, I would prefer if the meaning and implications of this nice plot were elaborated a bit more in the text. Is my interpretation of a change from a Pg-like to a Pn-like trend at around 150-200 km correct? How does this fit to theoretical overtaking distances, does this mean that the first arrival is picked everywhere?
We think that the change from Pg to Pn (as the first arrival) is somewhere between 100 and 200 km. This distance depends on both earthquake depth and Moho depth. Since earthquake depths vary between 0 to 20 km and also Moho in this region is very variable, we think that showing the travel-times cumulatively in one graph is indeed not helpful for the manuscript. Therefore, we decided to remove this figure from the manuscript. We also removed some parts of section 3.3 of the original manuscript and moved some parts to section 3.2 (no section 3.3 in the updated manuscript).

Figure 6: missing values in the histograms (Mu and sigma)
The values are by mistake missed in this plot. However, we decided to remove these histograms and showing only the misfits (figure 6b of the updated manuscript) that are more important for synthetic recovery test (also in response to the point (26) of Reviewer 2).

Figure 7: should be 110 m in depth (not km)
corrected

Figure 8: What is the reason for the wide spread of models at very shallow depths? Looks like they did not converge there (same in Fig. 10).
We think that the larger standard deviations of the models shown in Figure 8 (in the original manuscript) are due to the shallow anomalies introduced in our synthetic model (lower

velocities to resemble sedimentary basins, higher velocities for "TW"). Obviously, models with different velocities exist explaining the data more or less equally well.

For the real data (Figure 9 of the updated manuscript), we see actually very small standard deviations down to 25km depth. Maybe the shallow low velocities are also due to shallow lateral heterogeneity (sedimentary basins).

Thicker lines for the plusses and crosses in subfigure b would be helpful.

Done

Figure 9: Typo in label: pahse should be phase;

corrected

also, I guess the green line marks the end of the burn-in phase.

Right - corrected

I don't really see anything in the right subplot (which also has no axis-labels)

The histogram is shown with the color blue. This is a sharp line centered at 0.36 s. The x-axis is now labeled.

Figure 10: Would it be meaningful to compare the station correction pattern to the one from the synthetic test (Fig. 8)?

To some extend yes. For example, the shallow low-velocity anomalies that we considered for the MoB and PoB in the synthetic model are not too far from the real structure. However, the station-correction pattern related to the TW anomaly in the synthetic test is not seen so clearly in the real data. We modified the text slightly:"

"... Besides, extreme positive corrections are seen in the PoB and MoB as it is expected for sedimentary basins, also consistent with the results of the synthetic test (see above). A pronounced pattern specifically related to the TW as seen in our synthetic test is not observed so clearly in the real data suggesting a different structure in the shallow subsurface. The pattern of stations in the ESA and a few stations in the WSA and CA agree very well with results by Diehl et al. (2009b)."

In the caption, abbreviations CA and EA should be Central and Eastern Alps (not Alpine)

Agree - changed

Figure 11: Caption: remove one of the two "and" in fourth line; see level should be sea level

Both done

Figure 12: Comparing to Figure 2, it seems that the vast majority of events is at the edge or slightly outside of the SWATH-D network. This should be mentioned, and maybe the rectangle shown in Figure 2 can be added here?

We agree, most of the events used in our study are on the edge of the SWATH-D network and some events outside. That's why we decided to add also data of AlpArray surrounding the SWATH-D network so that all events used in our study are well surrounded by stations (gap<180°). This information is given in our manuscript.

The SWATH-D rectangle is added to this figure.

Figure 13: Caption: "for a better clarity the depth and length scales of cross section A are magnified by a factor of 1.5"; does this mean vertical exaggeration of 1.5? Or only that it was upscaled by a factor 1.5 relative to profiles B and C (without any distortion)?

Both vertical and horizontal axes in cross section A are upscaled (without any distortion) compared to the other two cross sections. This is clarified in the caption as well.

On behalf of the authors,
Azam Jozi Najafabadi

---

## Author Response (AR2)

We would like to thank Prof. Edi Kissling and two other anonymous reviewers for their insightful comments on our manuscript. The constructive comments helped us to further improve the manuscript. We edited the manuscript carefully and addressed all comments of reviewers. Please find below the individual comments of the reviewers as black text and our answers as blue text. In the appended manuscript, the color red refers to old texts that are removed or replaced, whereas new expressions and changes are denoted by color blue.

On behalf of the authors,
Azam Jozi Najafabadi

****** **Answers to Referee #1** ******

In this paper, the authors relocate 344 earthquakes in the southern and eastern Alps by exploiting arrival time data from the temporary SWATH-D network, supplemented by Alp-Array stations. Overall, the paper is well written, the methods well explained and tested, and the results carefully obtained and discussed. I suppose one could argue that the work they carried out would normally be integrated into a local earthquake tomography study, in which preliminary location of events is undertaken and a robust 1-D reference model is generated prior to the tomography. However, given that this is a new and large dataset from a region of significant interest, that cutting edge methods were used for the hypocenter locations and 1-D velocity structure determination, and that the final earthquake distribution does provide insight into active faults in the region, I would be happy to see the paper published following minor revisions.

**(1)** Line 9: I would be tempted to replace "precise" with "robust".
Done

**(2)** Line 43: I'm not sure I would say "...they depend not only on proper choices of initial values for hypocenter coordinates..." - what is meant by "proper" in this context – that they are close enough to the "true" location to make the inverse problem locally linear?
Yes, by proper we mean close enough to the true location. This explanation is now added in brackets in the updated manuscript.

**(3)** Line 44: I don't think I would describe damping as a "technical parameter". It is better described as regularisation.
Agree – changed to "regularization"

**(4)** Lines 115-119: In a way it's a pity that automatic event detection was skipped, since presumably the combined array used in this study provides a much denser data coverage of the region compared to what national seismological agencies have access to. Consequently, there is probably quite a lot of seismicity that has been overlooked, and if the purpose of this paper is to examine earthquake distribution and its relationship to active faulting in the region, then that is somewhat unfortunate. However, on reading the next lines, it appears that an automated picking algorithm was applied, but presumably only to data windowed by the pre-existing catalogue? It would also be interesting to know why the combined national seismological agencies were able to detect and presumably locate 2,639 local events, yet only 384 were deemed good enough for the current study. I understand the issue of seismic gaps

and noise, but are the national networks more dense than the network used in this study in some regions?

Yes, you are right, the automatic picking is applied only to the data windowed by the pre-existing catalog events. We totally agree that running a detection routine on the data of this dense and large network will probably yield a much more complete set of events (down to smaller magnitudes). However, our aim was to concentrate on locating the events with high precision rather than obtaining a comprehensive catalog. To locate the events with high precision they have to have a minimum magnitude and – in turn – a relatively large number of observations. We are confident that these (larger) earthquakes are contained in the permanent networks' catalogs. Additionally, one reason for this selection – besides the high-precision – is that we plan to use this dataset also for local earthquake tomography. We mention this now more frequently in the text.

Considering that no information on the precision of national catalog events is available and some of their events are poorly locatable (due to smaller number of stations and larger inter-station distances), we do an event selection based on our own location information after automatic picking:

We used the origin-time of 2,639 local events from national catalogs (now in appendix A of the manuscript) to start the automatic picking with (the automatic picking was not skipped). After the automatic procedure we saw that lots of events are either on the periphery of the network or too weak or too noisy to be detected by more than 5 stations. Moreover, automatic procedure ignored some of the clear picks in the nearest stations and had some suspected picks in the farthest stations. Therefore, we decided to manually/visually check the picks (semi-automated picking). For this purpose, we selected the events with gap<200 and RMS <1s (not too conservative) for further consideration. We explain later in the text that from these 384 events, some are blasts, and some are unclear to us, and some with very small numbers of picks.

**(5)** Line 140: Should be "..and thus is easily mispicked."
Done

**(6)** Line 175: It is not clear to me how station terms can account for 3-D variations in velocity structure that are not considered in the inversion for 1-D velocity structure.
We think our formulation was misleading. Therefore, we rephrased it as following:
"Moreover, the model **m** comprises station–corrections for P and S waves ($T_P$ and $T_S$), which account for travel-time effects (delayed or earlier arrivals) due to deviations of the 1–D model from the real 3–D velocity structure in the shallow subsurface beneath the stations."

**(7)** Line 188: "...for a very large..." - should be "...a very large...".
Done

**(8)** Lines 188-193: Perhaps I misunderstand something, but the velocity model is defined by a series of horizontal layers, each with constant Vp and Vp/Vs? So why does a 3-D Voronoi mesh with 1 km spacing vertically and horizontally come into it? While it may be technically correct to use this terminology, isn't it less confusing to describe this as a regular mesh in 3-D with 1 km spacing? Also, is there any need to correct for Earth's sphericity, since I believe that the Podvin and Lecomte method is in Cartesian coordinates?

For efficiency, we use a fast 2-D Eikonal solver to calculate the travel times. Therefore, we do not use a 3-D mesh but indeed a 2-D mesh with 1x1 km grid node spacing (vertically and horizontally). The irregular 1-D velocity model (defined by the set of $Vp_i$, $Vp/Vs_i$) is converted into this fine 2-D mesh by assigning the velocity value of the nearest model node ($Vp_i$ or $Vp/Vs_i$,

respectively) to the fine grid nodes (which is in fact some kind of Voronoi cell). Nevertheless, in order to avoid confusion, we modified the sentence to:

"Therefore, the irregular velocity model is converted to a fine and uniform mesh by setting the velocity at each mesh point to the value of the nearest point from the irregular model ($Vp_i$ or $Vp/Vs_i$, respectively). The fine mesh used by the Eikonal solver has a cell spacing of 1 km vertically and horizontally. "

Based on similar earlier studies, the dimension of our network seems to be small enough to neglect the sphericity, and similar inversion codes for local earthquakes (Velest; Kissling et al., 1994) or simul2000 (Thurber 1977) make this simplification as well (and numerous studies with similar-sized networks)

The actual model is a one-dimensional one, i.e., a set of n layers with constant P-velocities and Vp/Vs ratios. The model is described as a system of equivalent one-dimensional Voronoi cells (=layers). These Voronoi cells are degenerated: typically, Voronoi cells are 2- or 3-dimensional objects.

**(9)** Lines 193-195: This is perhaps slightly confusing, because the first part seems to indicate an L1 measure of misfit, but with a Gaussian likelihood function, the actual misfit would be L2.

For the Markov Chain method, we used the L2 norm and corrected this in the manuscript accordingly:

"a misfit function, particularly for each model, is defined as the summed squared differences between the observed (d) and calculated travel-times."

**(10)** Line 215: It would be interesting to have some numbers on how many iterations constitute the burn-in phase, and how many subsequent iterations were used to build the posterior PDF.

In this section (Method), we only explain the methodology and how the inversion works. We provide these numbers later in the text. As the burn-in phase and also the number of iterations is data-driven, these statistics are shown in figures 8 and 9a for the real data and also in the corresponding text. In particular, we used ~15,000 of the final models (every 1000[th] of all models after the burn-in phase) for the calculation of the posterior PDF; we describe this in Section 6.

**(11)** Line 263: Should be "earthquake", not "earthquakes".
Agree - changed

**(11)** Line 273: I find it interesting that the Vp uncertainty is almost zero in the 0-20 km depth range, which the authors put down to dense ray coverage. What values do these uncertainty estimates take, and are they comparable with, say, the standard deviation of the lateral heterogeneity of the synthetic model input at that depth?

The velocity uncertainty can depend on both the data (ray coverage and pick errors) and the lateral heterogeneity. A comparison between the lateral heterogeneity of the synthetic model and velocity uncertainty shows that they follow almost a similar pattern, although, not exactly equal. The lateral heterogeneity of the synthetic model is ~0.05 km/s between 2 and 20 km depth. It varies between 0.1 and 0.5 km/s below 20 km. These values are now added to the text.

**(12)** Line 284: Should be "Results and discussion".
Done

**(13)** Line 288: What happens if all model unknowns are allowed to vary in the initial tranche of iterations?

Actually, there is no difference in the final models (Vp, Vp/Vs, quake locations, etc.) when running the inversion with or without the "first phase" (where we do not change the initial velocity model but only quake locations). When using no "first phase" the run-time (CPU time) is only significantly increased. So, we introduced the first phase for practical reasons to accelerate the computation. Note that we start our Markov chains with completely random velocity models and initial quake locations, i.e., these initial values are potentially "very far away" from the final results, spanning a very wide range. Keeping the initial velocity models fixed during the "first phase" moves the initial quake locations to their approximate epicenters, thus accelerates the inversion.

**(14)** Line 319: Should be "A detailed interpretation of the pattern of corrections. . .."
Done

**(15)** Line 323: Should be "Estimation of hypocenter accuracy. . .."
Done

**(16)** Section 6.3: This section is essentially fine, although with a relatively modest database of 344 earthquakes, it is not entirely clear what new insights are brought to the table beyond what might be gleaned from national catalogues that have been accumulated over periods of decades and involve many more earthquakes (albeit not as well located). To some extent this brings us back to the question of trying to use auto-detection methods that take advantage of this large array to find potentially large numbers of small earthquakes missed by the national agencies.
Our database is mainly aimed for high location accuracy of the occurring earthquakes and not focusing on relatively small earthquakes (with only a few observations). Especially the focal depth-estimates are very sensitive to the quantity and quality of the picks and the velocity model. Furthermore, we intend to use this dataset for the calculation of the Local Earthquake Tomography (now this is emphasized more frequently in the manuscript). Therefore, instead of creating a comprehensive catalog, we focused on having a selection of the most consistent and precise hypocenters. We can also add that the microseismicity of this region is being further studied by other groups within the research project.
We think that the earthquake locations that we established in this study have the highest precision in the region (at this time) thus enables us to interpret for example the depths of the earthquakes at the southern Alpine front (section 6.3).

******Answers to Referee #2 – Prof. Edi Kissling ******

The manuscript regards the compilation of a local earthquakes catalog of 16 months period with the application of a few modifications and improvements to standard location procedure using the dense AlpArray and SWATH-D temprary station network. The study comprises different topics-procedural steps and results- each of potential interest to a wide range of readership.
Interesting enough the first such topic addressed in the abstract is the description and attempted correlation of the seismicity with the regional geology and tectonics. While certainly precisely relocated, the 344 local earthquakes of a 16 months period by no means could be taken as representative for the seismicity in the region and it should not come as a surprise - and not be seen as regional "characteristics"- that it appears in clusters. For a seismotectonic interpretation linking such observed clusters with tectonic faults to conclude, f.e., that "the

general pattern of seismicity reflects head-on convergence of the Adriatic indenter with the Alpine orogenic crust." one would have hoped the authors to take advantage of the great data set with on average 36 P observation per event to complement the hypocenter locations with focal mechanisms at least for the larger magnitude events.

We fully agree that for an in-depth seismotectonic study, focal mechanisms are essential. This issue was covered in other projects within the SPP. A manuscript dealing with focal mechanism was submitted to Solid Earth (Petersen et al., 2021), which we now reference in our manuscript. We also added some more information about the mechanisms in some sub-regions. Please see our response below.

Furthermore, a comparison and thourough discussion of the relation of the presented high-precision short-period seismicity with the long-term seismicity pattern reveiled by the official catalogs over the past 30 years is not only possible but necessary.

The general pattern of the seismicity agrees with previous studies, e.g., Reiter et al., 2018, which we mentioned in the section "Discussion" of the manuscript. Furthermore, in the discussion section we compare the seismicity distribution of our study with those from previous studies in several sub-regions (based on local networks or national agency data). In the updated manuscript we added some sentences regarding the similarity of the seismicity pattern to long-term seismic catalogs such as the SHARE catalog as well.

The main topic and work of the study regards the successful application of a Markov chain Monte Carlo inversion of the 12,534 P and 7,258 S observations from 344 local earthquakes to obtain a 1D velocity model and station delays for the region that allows high-precision hypocenter locations. The derivation of the model is well explained and complemented with the description of a synthetic test to provide a statistical estimate of the location uncertainties. In addition, a "ground truth" location experiment with quarry blasts is presented and discussed. This part of the manuscript is very clearly presented and contains a lot of technical details that allow the interested specialist to follow most steps. Considering the readership that might be interested in the seismicity and their tectonic interpretation though, I suggest to most of the chapter 5 could be moved to the supplementary material. What is missing, however, is a critical discussion of the results and, in particular, their relevance and meaning for the seismic catalog of 344 events presented. Considering that with the Markov chain MC inversion the authors address the coupled hypocenter-velocity model problem for the complet (very high-quality in terms of number of observations per event) 344 event data set, I do not understand why there is no mentioning about the internal consistency of the hypocenter solution or about the great potential of these results as initial data (hypocenters and model alike) for 3D seismic tomography. Rather, the list of relocated earthquakes is presented simply as a higher-precision-"than INGV/ZAMG" catalog for the region.

We agree with your comment regarding the internal consistency of our hypocenter solution and the ultimate goal of creating this dataset and therefore, we added the following text to the section "introduction" of the updated manuscript:

"The dense, high-quality travel-time picks created in this study potentially lead to constrained hypocenter solutions with high internal consistency. This will enable us to identify the general pattern of seismicity on the surface and at depth throughout the region and contribute to the understanding of active tectonic processes. A further aim of the study is to derive a high-quality dataset suitable to be used in Local Earthquake Tomography (LET)."

A similar text is in the updated section "conclusion".

We totally agree that the study provides a very well-suited dataset for further studies such as local earthquake tomography. We emphasize this aspect now more clearly throughout the text. In chapter 5 we intended to conduct a synthetic recovery test. With this test, we can study how (well) the synthetic hypocenters are recovered by the inversion routine and how the 1-D velocity

(output) looks like the 3-D model (input). Also, how shallow velocity anomalies are "mapped" by the station-corrections and whether the input random noise is recovered by the McMC method. We think that this is an interesting test for showing the performance of the method, especially when keeping in mind that we apply a relatively novel McMC search. We report the outcome of this test (please see also our response to your point number 17). Similar tests are regularly employed in the inversions for (3-D) subsurface structure (i.e., tomography). Therefore, we decided to keep chapter 5 in the main text of the manuscript. Nevertheless, we modified the text in the updated manuscript and hope that the reason behind this test is now better clarified.

Finally, the study also contains a section interesting for seismologists (observatory tasks and seismic tomography) about the semi-automated picking of such a large data set. Much of the description of this work part is already allocated in the supplementary material and it should remain there. It would be logic though if the reference to this work and the presentation and discussion of the results would be appearing before and not after the Markov chain MC inversion of the data set (see Figure 7). Furthermore, some details important for the specialist are missing (see individual points 10 to 15 below).

In section 3.2 of the updated manuscript, we added a table and histograms (table 1 and figure 4) regarding the results (travel-time dataset) of the semi-automatic picking procedure. Further detail about the event list from national catalogs is now added to the updated manuscript as a new appendix A and it is referenced in section 3.2.

The chapters 6 (Results in discussion) and 7 (conclusions) read like they were written by different people with totally different interests and perspectives and the only connection between the two parts are the 344 precisely located hypocenters. There is no geologic-lithologic interpretation or at least comment about correlation presented between the other results (notabene of great importance for the claimed reliability and accuracy of the hypocenters) of the coupled problem, the velocity model and station delays. The tectonic interpretation of the seismicity presented in chapter 6.3 (pages 21 to 25) is missing taking explicitly into account (and explaining to the reader how and why this is used as arguments for the interpretation) the great advantage of this study (having an event data set of high internal consistency and high precision hypocenter location of quantitatively known uncertainties) and the significant limitations (pre-selected events of unspecified magnitude of completeness and only 16 months of observation period).

Our intention is to keep the conclusion (and in general the manuscript) more technical than interpretational (in terms of seismotectonic). However, we rephrased the whole conclusion in order to have a reasonable balance between all chapters of the manuscript. In this regard, we rephrased the abstract as well so that our main aim of the manuscript is better conceived.

In consideration of the above general remarks on quality and deficiancies , I suggest moderate revision of the manuscript before publication.

**Specific points:**
Below are our responses to your specific points:

**(1)** Line 12. Replace "accuracy" with "precision estimate". Note that in line 13 you correctly assess the "accuracy" with the blast location test.
Agree – changed

**(2)** Line 15. Delete the rest of the sentence after:" ..1.7 km in depth."
Done

**(3)** Line 27. Replace "accuracy" with "precision"
Agree – changed

**(4)** Line 48. "... has the advantage of being" largely "independent …"
Done

**(5)** Line 51. "best model(s)." a note on ambiguity would be useful
This part is rephrased to:
"Moreover, the results can be statistically analyzed, and thus errors and ambiguities can be estimated. The method extends the probabilistic relocation approaches (Lomax et al., 2000) by inverting for a set of velocity models well explaining the data. "
However, more details are provided in the method chapter of the manuscript

**(6)** Line 64. Please outline Adriatic microplate in one of the Figures.
Done

**(7)** Lines 67-75. Needs a figure to show the strain if you keep the introduction as is and the chapter 6.3.
We think that we have already provided all references to our statements regarding the strain (mainly in section 2). Nevertheless, now the companion paper by Verwater et al. 2021 is submitted (and accessible) in which this topic (including a Figure) is covered in more detail. Therefore, we prefer not to add an additional Figure to our paper. We add a reference to Verwater et al. 2021 to Section 2.

**(8)** Figure 1. See point 6 above. Red box in bottom figure does not correspond with bounds of upper figure. This figure is not providing all necessary tectonic information mentioned in the manuscript. You should note that the seismic catalog presented by ISC is by far not complete down to magnitude 2. If you want to show the big picture use either EMSC catalog likely complete to M3 or ISC likely complete to M3.5. Otherwise you could use a composite of the various national catalogs that probably are complete to M2.5.
The figure bounds are corrected.
All the information which is mentioned in the text is now added to figure 1.
The big picture of seismicity (Figure 1) is now taken from EMSC with M3.

**(9)** Line 85. Actually there are earlier catalogs that were compiled: European Geotraverse Blundell et al. 1992, Solarino et al. 1997
Agree – two references are inserted in the text.

**(10)** Lines 115 to 119. You need to elaborate in detail (this can be done in supplementary material but it is absolutely necessary to have this information) how you identified the "common" events and how in the end you established the event list of the 2619 events.
An appendix A with the information regarding how the event list is formed is added to the updated manuscript.

**(11)** Lines 120 – 126. The discussion of the results of this semi-automated picking (that is well described in suppl.) needs to be more extensive and detailed. On what basis did you define the selection criteria (gap<200 , why not <180') (RMS <1s), why no mentioning of number of P obs? Did you check all 12534 P obs manually?

We used the origin-time of 2,639 local events from national catalogs (now in appendix A) as initial data to start the automatic picking with. However, we don't have any information about the precision of these events beforehand. Therefore, in the beginning, we decided to select as many events as possible from the national catalogs, apply an automatic picking procedure for these events, and then make a further selection based on our own picks and location information. After the automatic procedure (2,639 events, 68,099 P- and 17,151 S-Picks), we noticed that many of the events are either on the periphery of the network, too weak, or too noisy to be detected by more than 5 stations. Moreover, we believe that the automatic picker missed some of the good picks or introduced some suspected picks. Therefore, we had an early selection criterion (not too conservative) in order to do a manual/visual inspection of the picks. These selection criteria (gap<200 and RMS <1s which gives 384 events with 18,390 P- and 7,762 S-picks) were applied to the automatic picking results.  We manually/visually inspected the P and S picks of these 384 events, with careful consideration for picks in the distances of the triplication zone and farther than that (we inspected a very large amount of the picks from these 384 events, maybe 90% of them). We explain later in text that from these 384 events, some are blasts, and some are unclear to us. Later on, this dataset was further selected for simultaneous inversion (301 local earthquakes with gap<180° and minimum 10 P-picks and 5 S-picks).

We added the statistics about the number of picks and their quality classes of 2,639 events after automatic picking to Appendix B (Table B1) of the updated manuscript. Detailed information of the selected dataset for the simultaneous inversion is now given in the main text (section 3.2; table 1 and figure 4) of the updated manuscript.

**(12)** Line 134-6 and Figure 3. The Wadati diagram shows significant numbers of observations with +/-3s residual relative to constant Vp/Vs ratio. Note the the Vp/Vs ratio varies within the crust and at Moho. You may see this in the Figure as the straight line is systematically shifted onto the side of the highest point density after about 25s P travel time. What residual range do you define as corresponding to the sum of 3D, lithologies and regular Gaussian observation uncertainty effects and what value denotes an outlier?

We agree that the points are systematically shifted to above the straight line indicating that the Vp/Vs ratio (slightly) increases at larger distances. This could be related to different (average) Vp/Vs ratios in the crust and upper mantle (which is e.g., also suggested by global models such as ak135). We mention this now in the text: "We notice that at P travel-times larger than ~25 s the observations tend to slightly larger S-P travel-time differences, potentially indicating a higher Vp/Vs ratio at larger depth, i.e., in the upper mantle."

According to own synthetic tests (using the source and receiver geometry of this study and including (1) a Moho topography based on a simplified Moho from Spada et al., 2013 (see section 5), (2) assumed (moderate) differences in average Vp/Vs ratio in the crust and upper mantle, (3) crustal Vp/Vs ratio anomalies which we expect for the region (e.g., Vigano et al.,2015) and generally from other regions of the world, and (4) picking uncertainties) we expect a scatter of no more than ± 2 to 3s around the linear trend of the points in the Wadati diagram. We would like to emphasize that we only considered values with ts-$t_p$ > 0.72 $t_p$ ± 4s as "outliers" and removed them. These outliers were only 0.3% of all observations (which should not make a difference even if they were incorrectly flagged as "outliers"). We state this now more clearly in the text. After this removal, only 2% of the whole observations have ts – tp > 0.72 tp±3 s.

**(13)** Line 144 and Figure 4. "Checking the phase-type is extremly important." I fully agree and record section display is a good zero-order approach. However, I do not think your figure 4 is of help for doing this. How realistic is the ak135 global model for P phase identification

considering the Moho topography by Spada et al. 2013 (your Fig.5c) or the 3D LET model by Diehl et al. 2009 but most important the literally more than a dozen refraction seismic lines that have been published (for a review see Kissling et al. 2006).

We agree. The comparison of the cumulated picks (from earthquakes with a variety of different earthquake depths) with the bounds of the synthetic traveltime curves (from a global model for different earthquake depths) was too coarse, too unspecific, and obviously not helpful. As we mentioned in the text (both in section 3.2 and in the appendix), we visually/manually inspected the waveforms of all earthquakes and their automatically determined arrival-times, and added, deleted, or modified picks where applicable. This inspection was done mainly on individual traces and was most helpful for identifying wrongly picked noise burst etc. Arrivals of stronger events, for which we expect XmX or Xn arrivals, were additionally assessed with the help of record sections (epidistance plots of individual earthquakes; based on preliminary locations). So, we are quite confident, that we identified most of the phases correctly. Nevertheless, we acknowledge - and are very much aware of - that even these detailed visual/manual checks cannot prevent some amount of wrongly identified picks in our dataset (a general difficulty for manual picks e.g., already pointed out by Diehl et al., 2009). We state this now more clearly in the text. In this context, we think that Figure 4 of the original manuscript is indeed not helpful for the manuscript and thus we decided to delete it, as we deleted parts of the text of section 3.3 in the original manuscript (some parts were moved to the newly formed section 3.2). In any case, we would like to emphasize that we never used Figure 4 for identifying and removing individual picks (potentially misidentified).

**(14)** Line 152. Please provide clear evidence and explain in detail strategy to identify PmP phase by using a totally inadequat 1D model.

We agree, based on this plot we are not able to do this. We deleted this statement (together with the whole paragraph) – please see our comment above.

**(15)** Lines 152/3. "the number of outliers, ..., is not significant to the total number of picks." What value do you define as being an outlier? (analog question to point 12) Note there are dozens of observations +/-2s from the main intensity of data points (that by the way is totally off your Pg line) and that individual hypocenter location precision (and even more important for accuracy) is in truth measured by the fit of just those observations that refer to the specific event.

Yes, we agree on this point. Our discussion of "outliers" was not specific enough. Similar to the phase-identification issue (see our comment above), we are very much aware that our dataset probably still contains some number of "outliers" (howsoever defined) and we stated this now more clearly. However, as we also state above, we did not use Figure 4 or any related (automatic) mechanism for identifying and removing individual picks (potentially misidentified). We have just used it as a – probably too simple – statement regarding the existence or non-existence of "outliers".

As stated above, we deleted the whole corresponding paragraph (and Figure 4) and modified the text.

**(16)** Line 218. "... does not depend on initial hypocenters, ..." I seriously doubt this (does not depend) and suggest to phrase it differently. Consider how you would identify an outlier with Figure 4 if you do not have a rather good idea the initial hypocenter! Furthermore, consider that you were using a priori information from existing catalogs for your semi-automated picking and that even with all this information you apparently found mispicks and had to select the 384 best events.

We only partly agree. We are still convinced, that our inversion method itself does not depend on the initial hypocenters (as is the case for the traditional inversion routines using a linearized approach, DLSQ inversion, etc.), see Fig. 8 in Ryberg & Haberland, 2019. However, we agree that some dependency might be introduced through the selection of picks (e.g., "outlier removal"), especially in the case of automatic picking (involving e.g., sequential intermediate location steps based on initial velocity models). Nevertheless, we think that even this influence is minimized because we are using a visually inspected pick-dataset.

We modified the part in the following way:

"Therefore, the McMC method only uses the travel times and does not directly depend on initial hypocenters, origin times, velocity models, or even the model parametrization (e.g., grid node spacing). Nevertheless, because in the (semi-automatic) picking procedure the selection of picks is involving the comparison with travel times based on preliminary hypocenters (which depend on initial velocity models), the results of the inversion depend *stricto sensu* to some degree on initial values."

**(17)** Line 226. This does not provide an "accuracy" estimate! May be internal consistency, precision.

Agree - It is changed to consistency.

And you need to provide reasons why this should be expected to be of relevance for the real individual
event locations.

The event locations (and the velocity model) derived by the inversion procedure depend on a large number of parameters such as the quantity and spatial distribution of earthquakes and receivers, noise, quality of travel-time readings, the class of the model used (in the inversion) and so forth. Furthermore, the test is not only interesting in respect to the hypocenters but also to the derived velocity model. We think that it is appropriate to test the recovery by synthetic tests as it is standard for the recovery of subsurface structure (e.g., in LET studies). Since the McMC approach for solving the coupled hypocenter-(1-D) velocity problem is quite novel we think that this kind of test is particularly interesting. We already described our reasoning for conducting the test in section 5, however, we added/modified the text slightly. Furthermore, we added some sentences to the discussion of the results of the test.

"Comparing input event locations (synthetic) and the inverted (output) ones allows us to study the recovery of the hypocenters, location consistency, and potential systematic errors related to the use of a 1-D model, which we can generally expect for the derived real hypocenters. For example, it can be studied whether events at the periphery or in certain parts of the model have systematically larger uncertainties (e.g., due to their location and/or spatial distribution of picks). Furthermore, we can study how the (output) 1-D model looks in comparison to the (input) 3-D model, how large the derived noise is in relation to the synthetic input noise and how the pattern of station-corrections corresponds to the shallow velocity anomalies. Similar tests are standard in structural studies (i.e., LET) to study the recovery of certain features. "

By the way, your blast test shows otherwise!

We think that the simultaneous/joint hypocenter relocation is a very powerful concept. We agree that our blast test indeed shows very good performance of this test.

**(18)** Line 230. This statement about Moho velocities is simply wrong. No 7km/s velocity has been reported in the Alps. Please check the literature.

While the reviewer is certainly right regarding the reported seismic velocity values in the Alps, we would like to point out that this is a synthetic test with the main aim of checking whether and to which extent a 3-D velocity variation (i.e., Moho topography and shallow velocity variations)

influences the recovery of the hypocenters (and which 1-D model is derived). We think that the exact values of the velocities are not that important for this test as long as they are in the "usual" range. We used also for example an angular shape of the Tauern Window and the sedimentary basins which is also not close to reality. We never claimed that these are reported values, it's a description of our synthetic model. Therefore, we prefer to leave the synthetic model as it is. Nevertheless (and in response to issues 17 – 21), we modified the whole paragraph describing the synthetic model.

**(19)** Line 232. The 19km are just along the flank of the Ivrea body and not relevant for the Moho topography beneath the Po plain.
Agree - changed

**(20)** Line 233. -5km is much to high and you are doing something wrong if you need to avoid rays through air by such model top elevation. Note that the increase in pressure within the earth causes a velocity increase with depth and the seismic waves to show a downward curvature You should use the average station elevation for ray tracing.
During the McMC inversion many different – also weird and unrealistic - velocity values, i.e., also models with a velocity increase with elevation, are (randomly) tested. This is totally different e.g., compared to velest. We are very much aware of the fact that usually on Earth there is a velocity increase with depth. These unrealistic models obviously have to be suppressed because otherwise, we obtain a somehow "mirrored" (artificial) velocity model above the surface (in our case, only models without low-velocity zones are accepted). Furthermore, the method does not use rays (or raytracing) at all but instead calculates the travel time field with an FD Eikonal solver. More information can be found in Ryberg & Haberland (2019) (exact reference in the bibliography of the manuscript).

**(21)** Line 245. Choosing a constant Vp/Vs ratio of SQR(3) is very problematic as we know it is wrong because it varies and likely the average is different.
This is a synthetic test, and the goal is mainly to assess the performance of the method to recover the earthquake locations when using a simplified 1-D velocity model in the inversion. We agree that the reality is more complex and maybe our test is not complete and might not cover all aspects which should be checked for. In Ryberg & Haberland (2019) a much simpler recovery test with a 1-D synthetic model was performed, now we wanted to add some more complexity (although not claiming that this is the ultimate test). We modified the section describing the synthetic model (please see also our response to issues 17 – 20 above and 24 below).

**(22)** Line 250/1. Not only refer to table but provide correct value here.
Done

**(23)** Figure 5. Your model extent in Figure 5a does not correspond with your map extent in Figure 5c
AND either of these extents differ from your study region shown in Figure 1 AND all of these are different than your Figure 2. Make certain you everywhere show the same study region extent, if you want to show more area around, then mark the study region.
Done

**(24)** Figure 5b. The Moho topography is wrong. There is a Moho offset across the plate boundary but you show a vertical Moho interface!

Yes, the reviewer is right, the Moho shown in the Figure and used in our synthetic test is only a very simplified version of the Spada et al. 2013 Moho (Moho offset, double Moho, etc. are not reproduced). Furthermore, the crustal structure is very much simplified. We state this now more clearly in the text. However, we still think that simplifications for the synthetic test model are acceptable, and a simplified synthetic model is still useful for studying the recovery of some features (hypocenters, 1-D model).
Please see also our comments on similar issues 18 and 21.

**(25)** Lines 259/60. "average uncertainty of 240m in longitude, 270m in latitude ...." How did you determine that? In such way this information is not usefull. With what probability do we have what location uncertainty for any single hypocenter?
In the previous paragraph, it is explained that the model parameters are defined based on the average and standard deviation of the final models. In the old version of the manuscript, we showed in figure 6b the histograms of the uncertainties (1σ) of all earthquakes. However, for avoiding any misunderstanding, we decided to not mention these uncertainties here and only talk about the misfits which are important for recovery assessment.

**(26)** Figure 7. Move to supplementary material.
We think figure 7a of the original manuscript is quite important to display in the main text because it is the best information showing the recovery of the synthetic test. Therefore, we removed figure 6b (not as important as Figure 7) and merged figures 6a and 7 into one figure (in the updated manuscript it is figure 6).

**(27)** Figure 8. Again a DIFFERENT STUDY REGION SHOWN!.
The figure frame is now set correctly.
What about the stations to the West of the Tauern window (as example, there are other regions with no visible symbols)? Do they have all zero station delay values or did you not obtain any values for them?
There are stations that have very small delays and it's probably hard to see them (they are like a point in the figure). The stations with zero delay are indicated now with a different symbol in the updated manuscript. There are also some stations without any delays, which were considered in the early part of the study but not included in the final inversion (e.g., no high-quality picks, station problems, etc.). However, all the stations to the west of Tauern Window have symbols (comparison between figures 2 and 7b of the updated manuscript).
Please explain in more detail what the velocity-depth function shows. In may view, it documents the data set is not capable to resolve the velocity structure below 30km depth and certainly not the Moho. This does not come as a surprise as it is well known that you loose vertical resolution below your deepest hypocenters.
We totally agree that resolution typically degrades below the (deepest) earthquakes. However, we would like to point out that the synthetic (3-D) velocity model in the depth range between ~30 and ~55 km is characterized by a strong Moho topography, which *per se* cannot be exactly recovered by a 1-D model (used in our inversion). We think that is not (only) an effect of the data (less ray coverage and resolution below the deepest earthquakes) but also of the inability of a 1-D model to capture a 3-D variation. So, the input model is indeed not well resolved in this depth range, but it is most likely an effect of the data and the much simpler model class used in the inversion. Within the model complexity (in each depth level) the input model is recovered in average (with some variability indicated by the standard deviations). Of course, upper mantle velocities are reached at a large depth. We think this is a nice result of our test which also sheds

light on the interpretability of such 1-D models. We modified the text and hope that this issue is somewhat clearer.

**(28)** Lines 297 to 304. This model discussion is inadequate with regards to the previously published information about the crustal structure. If your model does not allow to resolve it, then say so and it is OK. But do not claim it is in agreement with prior independent knowledge if it is obviously not.

We agree. For example, our model is relatively close to the Diehl et al. (2009) model in the upper part, however, at larger depth it is different. We describe this now better. Furthermore, the resolution of our model is only good down to about 45km depth, below that the standard deviations (1 sigma) are getting significantly larger indicating fading resolution. Also, this aspect is now described in more detail in the updated manuscript.

**(29)** Line 320. "it proved to be useful for accurately localizing earthquakes ..". I am missing the prove. Please explain how this was proved.

Our phrasing was obviously misleading. We wanted to add a more general statement that - although it might be difficult to interpret the station corrections in a straightforward ("physical") way - the whole concept of the simultaneous /joint inversion (including the inversion for these station corrections) is very powerful. We reformulated this part in the text to:

"Nevertheless, the general concept of simultaneously inverting local earthquake datasets for a (simplified) 1-D velocity model, hypocenter positions/origin times, and station corrections proved to be very powerful for accurately localizing earthquakes (see e.g., Kissling,1988)."

**(30)** Figure 10. Figure 10b I would again derive the conclusion from this figure that you are lacking resolution power below 30km for Vp and below 20km for Vp/Vs ratio.

The resolution for Vp is starting to fade below 30km depth, however, standard deviations of around 0.02 in the depth range between 30 and 40 km (yellow ranges in Figure 9 of the updated manuscript) still indicate fair resolution. For Vp/Vs, we see this limit at around 30 km. We write this now in the text.

Figure 10c. Now this looks like the study region. Why not always marking this extent where you do have data from?

We homogenized the lat/long in all the maps.

How do you interpret the distribution of the station delays?

We discussed the pattern of station corrections and possible qualitative relations with geological features in lines 313 – 318 of the original manuscript.

Note that there is a single station delay strongly different from all others within its vicinity located near 11.7E/47N. If I obtained such result I would check if it is real or caused by bad data of some sort. Note that otherwise your largest station delays are all within the periphery of the study region as this is well known from minimum 1Dmodel applications.

Yes, the reviewer points out a totally valid point, thanks. We checked the data and identified this station as having wrong picks. Since we generally have to expect influences of these wrong picks on all other parameters in our simultaneous inversion (velocity model, other station corrections, and hypocenters), we removed all picks of this station, reran the simultaneous inversion, and located again all events with the updated velocity model. So, Figure 9 (of the updated manuscript) shows now the updated data. Please note that the changes in the hypocenters and station corrections were very marginal, and all inferences in the paper are not changed.

**(31)** Lines 325 to 335 and Figure 11. What hypocenter depth did you test these blast locations with?

How do you define the mis-location vector? Relative to the center of the quarries or do you know the precise location within the quarry for each blast? This accuracy test shows that your previous precision estimates were a bit to optimistic but the accuracy is still very good.

We appreciate this question. It motivated us to calculate the misfit vectors for the blasts similar to Husen et al. (1999) using the centers of the quarry areas (since we do not have the exact blast locations). We, therefore, added the following text to the manuscript:

"Based on the average mislocations of the blasts (relative to the centers of the quarry areas) and also their location uncertainties, we estimate the absolute location errors in the range of 1 km horizontally and 500 m vertically."

**(32)** Line 331. Please explain in theory why you suggest that including S observations improve the hypocenter location solution?

According to Gomberg et al. (1990) S-picks provide powerful additional constraints on hypocentre location, especially regarding the focal depth. They demonstrated that the partial derivatives of S-travel-times are larger than for P-waves and that they act as a unique constraint especially in cases of S-picks recorded within 1.4 focal depth's distance. However, precisely measuring the S phases is absolutely necessary and we carefully and conservatively identified the first arriving S-picks on the 3C high-quality recordings. Nevertheless, we think that our statement in Line 331 (of original manuscript) was not really precise and we decided to delete the statement regarding the S-picks. We replaced it by:

"It is expected that the errors of earthquakes (which are potentially deeper than blasts) are smaller than those estimated from this test because they are less affected by the heterogeneous shallow structure which was poorly accounted for by the model (Kissling 1988 & Husen et al., 1999)."

**(33)** Lines 340/1. Your absolute depth uncertainty has been documented by your ground truth accuracy test (Fig.11) to be a few km and the epicenter location uncertainty is about 1km. Please correct your numbers.

The location uncertainties that we report here are the standard deviations from the mean value of all the models of McMC. We provided now new estimates for the mislocation (based on quarry blasts test; see point 31) that we can now mention here as well. We rephrased the sentence to:

"Based on statistical analyses of the McMC inversion results, the average epicentral and depth uncertainties (1σ) are ~500 m and ~1.7 km which are compatible with precision estimates by the synthetic test (Sect. 5). However, the absolute location errors estimated by quarry blasts test (Sect. 6.2), are 1 km horizontally and 500 km vertically. "

**(34)** Lines 343/4. These differences are indeed significant. However, as it just regards a selected best event data set with on average 36 P observations this is comparing apples and grapes. Your data set is excellent for seismic tomography but absolutely not representative for a seismicity catalog (magnitude of completeness? Just 16 months). On the other hand, the national seismic catalogs contain many poorly locatable –or you could also say difficult to locate- events that need extra processing time to obtain a complete catalog and there is the significant difference in number of stations. I believe it would be useful to discuss thourougly these difference in addition to presenting just the numbers.

Our intention was not to show poor hypocenter solutions from the national agencies. Most of the large differences between location of the blasts reported by the agencies and our solutions

(and the mislocations) are indeed related to the amount of data (stations). We added the following paraphrase to the text:

"Since the national agencies are (probably) using much less data for the location (smaller number of stations used, larger inter-station distances), a significant difference between their hypocenter solutions and those obtained in this study is expected (average of 2.4 km in epicenter and 3.7 km in depth). The earthquake depths calculated by McMC are systematically shallower than those by national agencies (by an average of 1.1 km). The maximum and minimum differences in the epicenters and depths (between McMC and national catalogs) are seen for the earthquakes from the INGV and SED, respectively.

The derived hypocenters in this study do not represent a representative seismicity catalog of the region (the national catalogs contain many small, poorly locatable events in a much longer period) but form excellent data for further seismological studies e.g., Local Earthquake Tomography (LET). Moreover, this highly precise hypocentral data allows further tectonic inferences."

**(35)** Figure 12. Figure 12b is not needed, just provide the uncertainty estimates. Note that for cluster interpretation relative hypocenter location uncertainty estimates (and that is what you obtain with your Markov chain MC inversion of the coupled problem that includes a joint hypocenter determination approach) are most important while for absolute location obviously the accuracy is key.

We agree and delete this part of the figure. We now mention the uncertainty values in the text.

**(36)** Figure 13. For seismotectonic interpretation of clusters along a fault system, you should definitely
employ focal mechanisms.

We totally agree. In the discussion, we wanted to compare our derived seismicity, which is now captured in a consistent way throughout the study area with a dense network, with the seismicity known from other studies and datasets (e.g., from permanent, though coarse, networks running for many years). In this context we also discussed some occurrences along fault zones, however, we did not go into too much detail here. We are aware that for a conclusive investigation, focal mechanisms as well as a more complete catalog is necessary. Within the research program, more studies are underway focusing on exactly these issues. In the meanwhile, a manuscript (Petersen et al., 2021) was submitted (to SE) showing focal mechanisms from the AlpArray and SWATH-D networks. We refer to this study now in the text, however, results were not available at the time of initial submission of our manuscript.

******Answers to Referee #3 ******

The present manuscript by Jozi Najafabadi et al. presents the compilation of a seismicity dataset that will, I presume, eventually be used for a local earthquake tomography study of the Eastern Alps. Using recordings from the dense SWATH-D deployment, they present a careful procedure of obtaining and verifying arrival time picks, derivation of optimal hypocenter locations and a best-fit 1D velocity model. The Bayesian approach for the inversion of hypocenters, velocity model and station corrections is something new, and the present manuscript provides a nice case study for its application. Lastly, the obtained hypocentral locations are compared to mapped faults, from which the apparent activity or non-activity of a number of structures in the Eastern Alps is inferred. This last part is where I see some potential

problems that will make some changes to the manuscript necessary. Overall, the paper is well written and definitely of interest to the readership of Solid Earth and the special volume "New insights on the tectonic evolution of the Alps and the adjacent orogens". I recommend moderate revisions before publication.

General comment:

In section 3.2, it is briefly mentioned that the events for which arrival times on SWATH-D stations were obtained and that were then relocated, used for deriving the 1D velocity model etc. were selected from a synthesized catalog that was based on the bulletins of national agencies. For the tectonic interpretation to be viable, this part needs to be made much more transparent. While it is fine to choose a subset of events based on network criteria when working towards a tomography study, it is a completely different thing when the activity or (more crucial) non-activity of faults is inferred from such a subset. In clearer words: the authors need to convincingly show that their chosen subset of events is representative, and does not systematically miss events from certain regions. I suggest to provide a map with all 2639 events from the different national catalogs, in which the chosen 384 are marked. It would likely be even better if the station distribution from the different national networks could be shown as well. I also suggest to better describe the reasoning behind this approach of choosing a subset of events from national bulletins. Are the national bulletins complete enough that one can exclude that the dense SWATH-D network contains signals from small, previously undetected events? Or was the focus on the larger events that would generate arrival times at a larger number of stations?

You are correctly pointing to the main aim of our work which is local earthquake tomography, and this is the reason to focus on creating a catalog with the most consistent earthquake hypocenters (and corresponding travel time picks). Our hypocenters have the highest precision in the region (at this time) and our clustered seismicity agrees very well with seismotectonic characterizations based on long-term records. Therefore, our hypocenters can shed light on active faulting and enables us for a convincing tectonic interpretation.

We totally agree that the dense SWATH-D network will make it possible to detect very small earthquakes not contained in the national catalogs. However, for the purpose of locating the earthquake with high accuracy - and in turn using the dataset (hypocenters, travel-time picks) for local earthquake tomography - we were mainly interested in earthquakes of some magnitude so that they are observed by a number of stations (see section 3.2 of the updated manuscript). We assume that these earthquakes are to a very large extent listed in the national catalogs (example: minimum magnitude -0.8 by ZAMG for the Austrian part of the study region).

[Figure]

Here is a map containing all 2639 events from the national catalogs and the chosen 384 events are indicated by white dots. Unfortunately, we have no information regarding the stations for the national catalogs. The number of figures in our article is already rather large, thus we would like to not include this figure in the manuscript. The events by national catalogs are publicly available.

I agree with a previous reviewer that fault plane solutions would be nice to have for a more detailed tectonic interpretation. However, I can see that the main aim of the manuscript is the description of the dataset that will be used for tomography, and speculate that the tectonics part was added mainly for the sake of the Special Volume topic. I believe the careful derivation of the hypocenters and velocity model, using a rather novel approach and performing many quality checks, is in itself enough material, so that a deeper-going tectonic interpretation employing focal mechanisms is not strictly necessary here,

Thank you very much for this comment. In the meanwhile, a manuscript dealing with the focal mechanisms was submitted to Solid Earth (Petersen et al., 2021). We mention this now in our manuscript and added the reference.

Specific comments:

**I.25:** Why were only data from 2017/2018 used when the stations ran into 2019? Should be mentioned with a word or two.

At the time of doing this analysis, the data from 2019 was not available. We will include an updated dataset in the LET study.

Also, mentioning the total number of SWATHD stations here could be useful, especially since the number of AlpArray stations that were also used is brought up in the next paragraph.

Done

**I.33/34:** "to identify the status of the seismically active volume...". This is a strange formulation, and should be changed.

Modified to:

"… to identify the general pattern of seismicity on the surface and at depth"

**I.36:** remove the

Done

**l.55ff:** I would recommend to use fewer abbreviations, this is making the manuscript unnecessarily hard to read. Best limit abbreviation to a handful of terms that really show up a lot throughout the manuscript, and write out the rest (this is maybe also my personal taste...).
We actually use these abbreviations several times in the manuscript, especially in the section "Results and discussion", and also in the figures. We think that using the whole word makes the sentence long (in some cases) and also reading the figures alongside the text would be more difficult!

**l.82:** this bracket is not closing again
Agree - corrected

**l.91:** stuck should be struck; corrected
about what time interval are we talking for the ML>6 earthquakes? Last decades, centuries, millennia?
"in the last several centuries (Slejko, 2018)" – This is added to the text

**l.96/97:** This statement is problematic, because while the present study is using a denser seismic network, the chosen approach of using a subset of events from agency bulletins (see General Comments) makes it impossible that previously missed events (if they exist) will be detected. Thus, the present study can do nothing to address the problem that is hinted at here (inactive region maybe because not well instrumented).
Agree – The improved coverage of the region with seismic stations was one of the main aims of the deployment of the dense SWATH-D network, however, the issue towards a more complete catalog is not directly covered in our study. Other studies dealing with this issue are underway, but results are not yet available. Accordingly, we think that this statement is not necessary here and thus we deleted the last two sentences.

**l.105:** The "however" doesn't fit here
Agree - corrected

**l.111:** remove "stations"
Done

**ll.115-119:** For me, this paragraph is the main problem of the manuscript as is. At least for the tectonic interpretation part, the authors need to convince the reader that no selection bias of earthquakes exists, i.e. that regions interpreted as aseismic based on the chosen 344 events are also aseismic if one looks at the entire >2600 events in the original database (see recommendatiosn in General comments). Also, the statement here seems to indicate that the national bulletins were deemed complete, which stands in contrast to l.96/97.
Our statement in lines 96-97 (which is eliminated in the updated manuscript) was related to individual seismological studies in the region (lines 85-86 of the original manuscript), and not national bulletins. However, if we consider the national bulletins with 2,639 events to be complete, it also indicates rare seismicity in the aforementioned region. We are not claiming for a complete earthquake catalog, but for a high-precise catalog of seismicity which agrees very well with previous studies in the region and also national bulletins.

**l.135/136:** how were outliers defined, and where can I see outliers in Figure 3 (can they be marked?)

All the observations that have $t_s$-$t_p$ >0.72 $t_p$ ± 4s were considered as "outlier" and removed from the data which formed only 0.3% of whole observations. After this removal, only 2% of the whole observations have ts – tp > 0.72 tp±3 s.

Based on experiences with similar datasets and according to own synthetic tests (using the source and receiver geometry of this study and including 1- a Moho topography based on a simplified Moho from Spada et al., 2013 (see section 5), 2- assumed (moderate) differences in average Vp/Vs ratio in the crust and upper mantle, 3-crustal Vp/Vs ratio anomalies which we see (in respect to spatial dimension and amplitude) both in our first tomographic results and in other regions of the world, and 4- picking uncertainties) we expect a scatter of no more than ± 2 to 3s around the linear trend of the points in the Wadati diagram.

More detailed information is provided in response to point (12) of Reviewer 2.

**l.138:** If a part of the goal audience are people mainly interested in the activity of structures in the (south)eastern Alps, the three phases and the triplication distance should be briefly explained, e.g. in a brief sketch that could be added to Figure 4. Also, giving an estimate of the overtaking distance, e.g. with the crustal thickness and velocity given in ll.147/148, would be beneficial.

Motivated by the critical questions and comments regarding Figure 4 by reviewer 2, we modified the text dealing with the phase-type identification considerably. By doing this, we also deleted Figure 4 which seemed not helpful in this context. Please see more details on this issue in the response to reviewer 2.

**l.140:** be (remove ing)

Done

**l.158:** is indicated the right word here?

Agree – the sentence is modified to:

"… a simultaneous inversion for hypocenters and velocity structure (and/or station corrections) is needed."

**l.160:** Well, a Bayesian-type approach has been used for all these geophysical studies. As it is written, it sounds like this was always the exact same approach (which it wasn't)

We agree. We modified the text slightly:

"Different to the conventional approach of damped least squares, we use a Bayesian approach (Bayes, 1763). Bayesian approaches have been applied in a number of geophysical studies (Tarantola et al., 1982; Duijndam, 1988a,b; Mosegaard and Tarantola, 1995; Gallagher et al., 2009; Bodin et al., 2012a,b; Ryberg and Haberland, 2018). Ryberg and Haberland (2019) recently implemented a hierarchical, transdimensional Markov chain Monte Carlo approach for the joint inversion of hypocenters, 1–D velocity structure and station–corrections for the local earthquake case."

**l.176:** structure (-s)

Done

**l.212:** reformulate that first sentence Section 5: I am not completely sure I understand the reasoning behind this test. The authors construct a first-order 3D velocity model of the Alps based on published data and perform a retrieval check using the real data (hypocenters,

stations) as input. Thus any misfit in the output should stem from 3D structure and/or general uncertainty, but only with the assumption of this specific 3D model...since the true 3D structure of the Alps will almost certainly differ from the utilized model (presumably only to secondorder differences?), do we have any idea if the retrieved 3D effects are similar for a (subtly, substantially) different reality? I think the paragraph needs a more detailed description about the purpose of the test, what it is supposed to show and what it can not show. Nevertheless, I appreciate the effort that went into performing it!

With this test, we want to study how (well) the synthetic hypocenters (and the velocity) model are recovered by our inversion routine. We think this is interesting because we use a simplified framework in the inversion (i.e., 1-D model, station corrections) obviously not capable to capture the full 3-D conditions which we expect for the Earth. For this, we designed a 3-D model (based on existing information) which resembles some first-order features in our study area such as Moho topography or shallow low-velocity sedimentary basins. Furthermore, we can study how the (output) 1-D model looks in comparison to the (input) 3-D model, how large the derived noise is in relation to the synthetic input noise and how the pattern of station-corrections corresponds to the shallow velocity anomalies. In inversions for (3-D) subsurface structure (i.e., tomography) people regularly employ synthetic recovery tests.

In order to make this clearer we modified the text. Please see also our comment regarding the issue (17) of Reviewer 2.

**l.268:** Figure (-s)
Corrected

**l.271:** reformulate "rather slight", add km after 50
The sentence is reformulated to:
"it is modeled by a gradual increase of the velocities at depths from 30 to 50 km"

**l.275:** reformulate "fluctuating"
"fluctuating" is replaced by "irregular". And by the way, this sentence is moved to section 6.1 where we talk about velocity models of real data.

**l.319:** "contains an overlay site effect" I'm not sure I understand what exactly is meant here.
The sentence is reformulated to:
"contain a superposition of site effects and/or effects from 3–D structural variations"

**l.324:** This is not really a sentence
Agree – There was a mistake in the formulation of the sentence. It's reformulated to:
"To validate the localization procedure, the detected 15 quarry blasts (based on manual/visual inspections; see 3.2) were independently relocated by McMC …"

**l.332/333:** Can hypocenters from the INGV/ZAMG catalogs also be shown in Figure 11, to better illustrate the improvement in hypocenter location?
The large differences in the location of the blasts reported by the agencies to our solutions (and the mislocations) are most likely related to much less data (stations), so it is not surprising that they are different (and show some mislocation). We think that the differences in location are already mentioned in the text. Furthermore, in order not to overload the (old) Figure 11 we prefer not to add the hypocenters to the Figure.

**I.344:** it would be interesting to elaborate a bit more on these differences; is there a trend, eg. with bulletins showing systematically larger or smaller depths?

In response to this comment and also to reviewer 2, we added the following paraphrase to the text:

"Since the national agencies are (probably) using much less data for the location (smaller number of stations used, larger inter-station distances), a significant difference between their hypocenter solutions and those obtained in this study is expected (average of 2.3 km in epicenter and 2.9 km in depth). The earthquake depths calculated by McMC are systematically shallower than those by national agencies (by an average of 1.1 km). The maximum and minimum differences in the epicenters and depths (between McMC and national catalogs) are seen for the earthquakes from the INGV and SED, respectively.

The derived hypocenters in this study do not represent a representative seismicity catalog of the region (the national catalogs contain many small, poorly locatable events in a much longer period) but form excellent data for further seismological studies e.g., Local Earthquake Tomography (LET). Moreover, this highly precise hypocentral data allows further tectonic inferences. "

**I.345:** This is not surprising, since no search for new events beyond those in national catalogs has been performed

We agree with your point. However, other seismological studies in the region (based on local networks or national agency data) confirm a similar seismicity pattern. Although, considering high-accurate and high-constrained hypocenters throughout the region in our study, a comparison with national catalogs could be meaningful too.

**I.358:** mention the magnitudes of these events

Done

**I.398:** See General Comment: how well can one argue for the absence of seismicity ("seismic gap") based on a catalog that was only a choice of 384 out of 2639 events? A map showing where the remainder of events (those that were not chosen) were located is essential if such a statement is attempted

In the updated manuscript, we emphasize that the seismicity pattern of our study is similar to long-term seismicity by, e.g., Reiter et al., 2018, or seismic catalogs such as the SHARE.

Furthermore, we always (for every sub-region) compared our seismicity with previous long-term seismicity studies. Our seismotectonic interpretation is based on the high-quality hypocenters that are derived in our study. However, we confirmed the sparse seismicity in part of the region with other studies and thereupon made an interpretation.

**I.405ff:** spelling: Engadin vs. Engadine Fault

Corrected

**I.427ff:** These (at least the first three dotpoints) are not results but the analysis steps that were carried out to retrieve the results. Either only results should be listed, or two separate listings for analysis steps and results are needed.

The conclusion is totally rewritten. Please look at the updated manuscript.

In the name list of AlpArray people, all those containing special characters have formatting issues (LaTeX syntax?)

Corrected

**Figures:**

Figure 1: Typo in Tectonic units legend (forland should be foreland)
corrected

Figure 2: The fault lines in this plot are really hard to see. Choosing a larger linewidth would be helpful. A color scale for the topography would also be nice.
Both done

Figure 4: Please be more specific about what the "various depths" are that were used to obtain the travel-time corridor. Then, I do not see red dashed lines in the plot (as mentioned in the caption). Lastly, I would prefer if the meaning and implications of this nice plot were elaborated a bit more in the text. Is my interpretation of a change from a Pg-like to a Pn-like trend at around 150-200 km correct? How does this fit to theoretical overtaking distances, does this mean that the first arrival is picked everywhere?
We think that the change from Pg to Pn (as the first arrival) is somewhere between 100 and 200 km. This distance depends on both earthquake depth and Moho depth. Since earthquake depths vary between 0 to 20 km and also Moho in this region is very variable, we think that showing the travel-times cumulatively in one graph is indeed not helpful for the manuscript. Therefore, we decided to remove this figure from the manuscript. We also removed some parts of section 3.3 of the original manuscript and moved some parts to section 3.2 (no section 3.3 in the updated manuscript).

Figure 6: missing values in the histograms (Mu and sigma)
The values are by mistake missed in this plot. However, we decided to remove these histograms and showing only the misfits (figure 6b of the updated manuscript) that are more important for synthetic recovery test (also in response to the point (26) of Reviewer 2).

Figure 7: should be 110 m in depth (not km)
corrected

Figure 8: What is the reason for the wide spread of models at very shallow depths? Looks like they did not converge there (same in Fig. 10).
We think that the larger standard deviations of the models shown in Figure 8 (in the original manuscript) are due to the shallow anomalies introduced in our synthetic model (lower velocities to resemble sedimentary basins, higher velocities for "TW"). Obviously, models with different velocities exist explaining the data more or less equally well.
For the real data (Figure 9 of the updated manuscript), we see actually very small standard deviations down to 25km depth. Maybe the shallow low velocities are also due to shallow lateral heterogeneity (sedimentary basins).
Thicker lines for the plusses and crosses in subfigure b would be helpful.
Done

Figure 9: Typo in label: pahse should be phase;
corrected
also, I guess the green line marks the end of the burn-in phase.
Right - corrected
I don't really see anything in the right subplot (which also has no axis-labels)

The histogram is shown with the color blue. This is a sharp line centered at 0.36 s. The x-axis is now labeled.

Figure 10: Would it be meaningful to compare the station correction pattern to the one from the synthetic test (Fig. 8)?
To some extend yes. For example, the shallow low-velocity anomalies that we considered for the MB and PoB in the synthetic model are not too far from the real structure. This can be perceived by looking at the station corrections in these sedimentary basins in both figures 7 and 9 (of the updated manuscript).
In the caption, abbreviations CA and EA should be Central and Eastern Alps (not Alpine)
Agree - changed

Figure 11: Caption: remove one of the two "and" in fourth line; see level should be sea level
Both done

Figure 12: Comparing to Figure 2, it seems that the vast majority of events is at the edge or slightly outside of the SWATH-D network. This should be mentioned, and maybe the rectangle shown in Figure 2 can be added here?
We agree, most of the events used in our study are on the edge of the SWATH-D network and some events outside. That's why we decided to add also data of AlpArray surrounding the SWATH-D network so that all events used in our study are well surrounded by stations (gap<180°). This information is given in our manuscript.
The SWATH-D rectangle is added to this figure.

Figure 13: Caption: "for a better clarity the depth and length scales of cross section A are magnified by a factor of 1.5"; does this mean vertical exaggeration of 1.5? Or only that it was upscaled by a factor 1.5 relative to profiles B and C (without any distortion)?
Both vertical and horizontal axes in cross section A are upscaled (without any distortion) compared to the other two cross sections. This is clarified in the caption as well.

**Relocation of earthquakes in the Southern and Eastern Alps (Austria, Italy) recorded by the dense, temporary SWATH-D network using a Markov chain Monte Carlo inversion**

Azam Jozi Najafabadi[1,2], Christian Haberland[1], Trond Ryberg[1], Vincent F. Verwater[3], Eline Le Breton[3], Mark R. Handy[3], Michael Weber[1], and  the AlpArray and AlpArray SWATH-D working groups[*]

[1]GFZ German Research Centre for Geosciences, Potsdam, Germany
[2]Institute of Geosciences, Potsdam University, Potsdam, Germany
[3]Institute of Geological Sciences, Freie Universität Berlin, Berlin, Germany
[*]For further information regarding the teams, please visit the link which appears at the end of the paper

**Correspondence:** Azam Jozi Najafabadi (azam@gfz-potsdam.de)

**Abstract.** Local earthquakes with magnitudes in the range of 1 - 4.2 ($M_L$) in the Southern and Eastern Alps (2017 - 2018)
registered by the dense, temporary SWATH-D network and the AlpArray network reveal seismicity in the upper crust (0-20
km). The seismicity is characterized by pronounced clusters along the Alpine frontal thrust, e.g., Friuli-Venetia (FV) region,
in the Giudicarie-Lessini (GL) and Schio-Vicenza domains, as well as in the Austroalpine Nappes and the Inntal area. Some
seismicity also occurs along the Periadriatic Fault. The general pattern of seismicity reflects head-on convergence of the Adri-
atic Indenter with the Alpine orogenic crust. The deeper seismicity in the FV and GL regions indicate southward propagation
of the Southern Alpine deformation front (blind thrusts). The first arrival-times of P- and S-waves of earthquakes are deter-
mined by an automatic workflow and then visually/manually checked and corrected. We applied a Markov chain Monte Carlo
inversion method to achieve precise hypocenter locations of the 344 local earthquakes. This approach simultaneously calcu-
lates hypocenters, 1-D velocity model, and station-corrections without prior assumptions such as initial velocity models and
earthquake locations. A further advantage of the method is the derivation of the model parameter uncertainties and noise levels
of the data. The accuracy of the localization procedure is checked by inverting a synthetic travel-time dataset from a complex
3-D velocity model and using the real stations and earthquakes geometry. The location accuracy is further investigated by the
relocation of quarry blasts. The average uncertainties of the locations of the earthquakes data are below 500 m in the epicenter
and ∼1.7 km in depth when using the average $V_P$ and $V_P/V_S$ models and the station-corrections from the simultaneous in-
version. In this study, we analyzed a large seismological dataset from temporary and permanent networks in the Southern and
Eastern Alps to establish high-precision hypocenters and 1-D $V_P$ and $V_P/V_S$ models. The waveform data of a subset of local
earthquakes with magnitudes in the range of 1 - 4.2 $M_L$ is recorded by the dense, temporary SWATH-D network and selected
stations of the AlpArray network between September 2017 and the end of 2018. The first arrival-times of P- and S-waves
of earthquakes are determined by a semi-automatic procedure. We applied a Markov chain Monte Carlo inversion method to
simultaneously calculate robust hypocenters, 1-D velocity model, and station-corrections without prior assumptions such as
initial velocity models or earthquake locations. A further advantage of this method is the derivation of the model parameter uncertainties and noise levels of the data. The precision estimates of the localization procedure is checked by inverting a synthetic travel-time dataset from a complex 3-D velocity model and using the real stations and earthquakes geometry. The location accuracy is further investigated by a quarry blast test. The average uncertainties of the locations of the earthquakes are below 500 m in epicenter and ∼1.7 km in depth. The earthquake distribution reveals seismicity in the upper crust (0-20 km) which is characterized by pronounced clusters along the Alpine frontal thrust, e.g., Friuli-Venetia (FV) region, in the Giudicarie-Lessini (GL) and Schio-Vicenza domains, as well as in the Austroalpine Nappes and the Inntal area. Some seismicity also occurs along the Periadriatic Fault. The general pattern of seismicity reflects head-on convergence of the Adriatic indenter with the Alpine orogenic crust. The seismicity in the FV and GL regions is deeper than the modeled frontal thrusts, which we interpret as indication for southward propagation of the Southern Alpine deformation front (blind thrusts).

**1 Introduction**

[revised manuscript text omitted]

We noticed that the number of mispicks and also ignored picks was quite large (Sect. B2). Therefore, a visual/manual inspection on the selected events (384 events with azimuthal gap less than 200° and RMS less than 1 s) was subsequently done to remove or modify obvious mispicks (mainly at large epicentral distances) and also repick the missed arrivals (mostly S-picks of very close stations).As described in Sect. B2, the number of mispicks and picks ignored by the automatic procedure was quite large. Therefore, a dataset with relatively highly-constrained events were selected for visual/manual inspection. Since the number of picks in this stage (after the automatic procedure) is not the most reliable information on the location precision

180 (see Sect. B2), and also considering that this data is aimed for high location accuracy of the occurring earthquakes, we selected events with azimuthal gap less than 200° and RMS less than 1 s which resulted in 384 event (18,390 P- and 7,762 S-picks). The inspection of this dataset was carried out with removing or modifying obvious mispicks (mainly at large epicentral distances) and also re-picking the missed arrivals (mostly S-picks at small epicentral distances).

This manual/visual inspection was mainly performed on individual station data. as well as on seismogram sections of all

185 observations of larger events to check for the correct phase identification (see also Sect. 3.3). The direct Pg (Sg), Moho-reflected PmP (SmS), and Moho-refracted Pn (Sn) phases arrive closely spaced in time, especially in the triplication zone, potentially leading to phase-type misidentification. The PmP (SmS) is always a secondary arrival, but its amplitude can dominate the first arrival and thus be easily mispicked. On the other hand, for epicentral distances larger than the triplication zone, the Pn (Sn) amplitude can be so weak, that the Pg (Sg) or PmP (SmS) be wrongly identified as the first arrival. In the local earthquake case,

190 and particularly for regions with varying Moho topography and significant lateral variations in the crustal structure, something to be expected for the Alps, the phase-type identification is even more challenging (Diehl et al., 2009b). Since the inversion is based on the first arrival-times we looked at the waveforms in epicentral distance plots for larger events for which we expect the later phases, inspected the picks and corrected/adjusted where needed. However, we would like to point out that even with these procedures we cannot rule out a certain amount of misidentified phases.

195 An inspection for identifying potential quarry blasts (and other anthropogenic sources) was simultaneously doneduring the visual inspection performed. Criteria for potential quarry blasts were: 1) relatively small number of S-picks, 2) relatively small S/P amplitude ratios (see, e.g., Walter et al., 2018), and 3) large surface waves (observed dispersive waveform characteristics). Based on our assessment, we classified the events into 344 earthquakes, 15 potential blasts, and 25 unclear events.

A Wadati diagram (Wadati, 1933; Kisslinger and Engdahl, 1973; Diehl, 2008) was used to identify and correct obvious

200 outliers. As seen in Fig. 3, the number of outliers after correction is quite low. The $V_P/V_S$ ratio is 1.72 (calculated by linear least squares) in our dataset.calculate the $V_P/V_S$ ratio of the earthquakes, which is 1.72 (calculated by linear least squares; Fig. 3). Furthermore, S-picks with S-P traveltime differences larger than 4 s off the main trend were removed (however, this was only 0.3% of all picks). After this removal, only 2% of the whole observations have $t_s - t_p > 0.72\, t_p \pm 3$ s. We notice that at P travel-times larger than ~25 s the observations tend to slightly larger S-P travel-time differences, potentially indicating a

205 higher $V_P/V_S$ ratio at larger depth, i.e., in the upper mantle.

[Figure]

**Figure 3.** Modified Wadati diagram based on P- and S-picks after visual/manual analysis (the color indicates number of picks; please see the color palette table on the right). The origin-times ($t_0$) of the earthquakes are provided by the single-event localization using HYPO71 hypocenter determination program (Lee and Lahr, 1975). The black solid line shows the fit to the data by linear least squares and its slope indicates the $V_P/V_S$ ratio of 1.72. The dashed lines show values with $t_s - t_p = 0.72\, t_p \pm 3$ s.

For the simultaneous inversion of hypocenters, 1-D velocity model, and station-corrections, we used the events with at least 10 P-picks and 5 S-picks and an azimuthal gap less than 180°, which comprises 301 local earthquakes. The arrival-time dataset consists of 11,084 P- and 6,496 S-picks (quality classes of 0 to 3) with average picking errors of 0.12 s and 0.21 s for P and S observations, respectively (Table 1). The average number of picks per event is 32 for P and 18 for S with maximum of 157 and 210 118, respectively (Fig. 4).

**Table 1.** Statistical parameters of high-quality P- and S-pick dataset of 301 well-constrained local earthquakes used for simultaneous inversion of hypocenters, 1-D velocity model, and station-corrections.

| Quality class | P picking uncertainty (s) | Number of P-picks | S picking uncertainty (s) | Number of S-picks |
|---|---|---|---|---|
| 0 | ±0.05 | 4,170 | ±0.1 | 2,715 |
| 1 | ±0.1 | 3,462 | ±0.2 | 1,254 |
| 2 | ±0.2 | 2,327 | ±0.3 | 1,563 |
| 3 | ±0.3 | 1,125 | ±0.4 | 964 |
| Sum (#) | | 11,084 | | 6,496 |
| Average picking error (s) | | 0.12 | | 0.21 |

[Figure]

**Figure 4.** histograms of high-quality P- and S-picks per event for 301 well-constrained local earthquakes used for simultaneous inversion of hypocenters, 1-D velocity model, and station-corrections.

In order to investigate the seismicity pattern, we use the travel-time data of additional 43 local earthquakes with slightly fewer picks (minimum 8 P- and 2 S-picks) to the main dataset and relocate them using the $V_P$, $V_P/V_S$, station–corrections, and noise levels from the simultaneous inversion (Sec. 6.3). This updated dataset contains 344 earthquakes, 12,420 P-picks, and 7,192 S-picks.

215

[revised manuscript text omitted]
). Nevertheless, the selection of picks in the (semi-automatic) picking procedure is involving the comparison of the observed travel-times with those based on preliminary hypocenters (which depend on initial velocity models). Therefore, the results of the inversion can *stricto sensu* depend to some degree on initial values.

**5   Uncertainty estimates using synthetic tests**

As a variable Moho topography and complex lithosphere structure are expected for our study region, the 1-D model derived by and used in our inversion can only be a rough abstraction of the real conditions. Nevertheless, this simplification could potentially have influence on earthquake location correctness and accuracy. To assess the performance of our inversion strategy in this respect, using a 3-D velocity model and earthquakes and station distribution of the real data, we calculated the synthetic travel-times, added synthetic noise and finally inverted them in the same way as the real data (see below). Comparing input event locations (synthetic) and the inverted (output) ones allows us to study the location and potential systematic errors related to the use of a 1-D model, which we can also expect for the derived real hypocenters. recovery of the hypocenters, location consistency and potential systematic errors related to the use of a 1-D model, which we can generally expect for the derived real hypocenters. For example, it can be studied whether events at the periphery or in certain parts of the model have systematically larger uncertainties (e.g., due to their location and/or spatial distribution of picks). Furthermore, we can study how the (output)

1-D model looks in comparison to the (input) 3-D model how large the derived noise is in relation to the synthetic input noise and how the pattern of station-corrections correspond to the shallow velocity anomalies. Similar tests are standard in structural studies (i.e., LET) to study the recovery of certain features.

320    The background 3-D velocity model is designed based on the Moho topography of the European and Adriatic plates from Spada et al. (2013). The $V_P$ starts with 6 km s$^{-1}$ at -5 km, gradually increases down to the Moho depth, and reaches 7 km s$^{-1}$. At Moho depth, the velocity increases from 7 km s$^{-1}$ to 8 km s$^{-1}$. The velocity below the Moho increases again gradually until it reaches a velocity of 8.05 km s$^{-1}$ at 90 km depth (Kennett et al., 1995; Christensen and Mooney, 1995). The Moho depth in the target area changes from 22 km in the PoB to more than 55 km along the Adria-Europe plate boundary (Spada et al., 2013). We considered the model's surface at -5 km to avoid wave propagation through the air.

In our synthetic 3-D background model the course of the crust-mantle boundary is inspired by the Moho topography of the European and Adriatic plates from Spada et al. (2013), however, we adopted it in a simplified way without features such as crustal stacks (overlapping Mohos of the two involved plates) or the proposed Moho hole. Our main aim of this simplified adaption is studying the influence of a 1-D model class on the derived hypocenters after the inversion and the goal is not recovering the exact velocity structure. We assigned typical crustal velocities to the upper layer, starting with 6 km s$^{-1}$ and increasing to 7 km s$^{-1}$ at the base of this crustal layer. Thereafter, the velocity increases from 7 km s$^{-1}$ to 8 km s$^{-1}$ within a thin zone of 3 km with a strong gradient. Then, the velocity increases again gradually until it reaches 8.05 km s$^{-1}$ at 90 km depth (Kennett et al., 1995; Christensen and Mooney, 1995). The Moho depth in the target area changes from 22 km along the flank of the Ivrea body to more than 55 km along the Adria-Europe plate boundary (Spada et al., 2013). We considered the model's surface at -5 km to avoid wave propagation through the air.

[revised manuscript text omitted]

375 Figure 7a shows the derived $V_P$ and $V_P/V_S$ models as heat-maps (posterior distribution of all the models). In addition, it shows the average value (red line), standard deviation of all the inferred models (yellow zone), and a reference model with maximum likelihood (blue line), i.e., maximum posterior probability (similar to Ryberg and Haberland, 2019). As seen, no clear, single Moho velocity jump is recovered, but there are rather slight velocity jumps at different depths from 30 to 50 it is modeled by a gradual increase of the velocities at depths from 30 to 50 km. We attribute this to. This reflects the variable

380 Moho topography, which *per se* cannot be modeled inby the 1-D velocity model, and potentially less resolution due to less ray coverage in this depth range. Because of a dense ray-coverage from the surface down to a depth of $\sim$20 km, the $V_P$ uncertainty is almost zero in this depth range. Below 20 km In the depth interval with strong Moho topography, the uncertainty varies between 0.1 and 0.66 km s$^{-1}$. The uncertainty values are compatible with the standard deviation of the lateral heterogeneity in the synthetic 3-D velocity model ($\sim$0.8 km s$^{-1}$ above 2 km, $\sim$0.05 km s$^{-1}$ between 2 and 20 km, and between 0.1 and 0.5

385 km s$^{-1}$ below 20 km depth). Thus, we think that the increased standard deviations beneath $\sim$30 km depth indicating reduced and fading resolution are due to both variable Moho topography and reduced ray coverage, however, their exact contribution is hard to tell.

The $V_P$ model is resolved until 63 km depth corresponding to the maximum ray penetration. However, as a 1-D model cannot be representative of the 3-D structure, especially in a region with expected fluctuating Moho and complex crustal structure, a

390 geologically meaningful interpretation of the derived $V_P$ model is hardly possible.

The $V_P/V_S$ ratio was fixed to the square root of 3 for the whole region in the forward modeling and the same value is recovered down to $\sim$35 km depth with almost zero uncertainty (less than 0.02). The uncertainty of $V_P/V_S$ increases below 35 km depth to a maximum value of 0.046 indicating that this deeper part of the model is not well resolved.

The stations-corrections (Fig. 7b) correspond, to large extent, to the shallow anomalies of the 3-D synthetic model (Fig.

[revised manuscript text omitted]

The derived V$_P$ model starts with a rather high value of 5.94 km s$^{-1}$ at the surface and increase only very gently down to a depth of around 20 km. After a zone of higher velocity gradient in the average model around (and a step-like increase of the reference model at) ∼28 km depth, velocities reach ∼6.8 km s$^{-1}$ at ca. 30 km (down to ca. 45 km depth). Down to this depth the uncertainties (1σ) indicating the resolution increase from 0.01 km s$^{-1}$ (very good) to 0.28 km s$^{-1}$ (fair). Our derived V$_P$ model in the upper crust – except for the uppermost layer shallower than 3 km – is similar to the Diehl et al. (2009b) model, an increase to values above 6.5 km s$^{-1}$ is observed deeper than in the latter model.

Below ca. 45 km depth velocities increase further up to 7.85 km s$^{-1}$ at 60 km depth, however, standard deviations (1σ) between 0.3 km s$^{-1}$ and 0.42 km s$^{-1}$ indicate only poor resolution in this depth range. A single step-like increase from „crustal" to „upper mantle" velocities (as at 35 km depth in the Diehl et al. (2009b) model) cannot be observed, which we attribute either to 1) the expected strong depth variability of the Moho in the study area (see discussion of the synthetic model in Sec. 5) and a gradual course of the 1-D model due to lateral averaging in this depth range, 2) the relatively small number of Pn arrivals and/or 3) limited maximum observation distances. Influences by wrongly-picked late-arriving Pg arrivals at large distances (instead of Pn phase) cannot totally be ruled out.

As a 1-D model cannot be representative of the 3-D structure, especially in a region with expected irregular Moho and complex crustal structure, a geologically meaningful interpretation of the derived V$_P$ model is hardly possible.

The V$_P$/V$_S$ model starts with high values at the surface (to ∼5 km depth), shows reduced values of around 1.70 down to 22 km depth before reaching 1.77 at greater depths (with uncertainty between 0 and 0.050.02). However, 1σ values of 0.02 to 0.045 indicate only poor resolution at depth larger than around 30km, which we associate to only a few Sn arrivals. This was basically expected from the Wadati diagram (Fig. 3) and is in agreement with values previously derived, e.g., by Viganò et al. (2013). Both V$_P$ and V$_P$/V$_S$ models are available in the supplementary material S1.

The station-corrections derived from the McMC simultaneous inversion potentially indicate local, shallow 3-D velocity anomalies in the subsurface, which cannot be accounted for by the 1-D model. The McMC inversion assumes that P and S station-corrections have an average of zero. Negative corrections indicate earlier wave arrival and thus higher velocities in the (shallow) subsurface, whereas, positive correction implies delayed arrival indicative of lower velocities.

The pattern of corrections (Fig. 9c) shows coherent negative corrections associated with the EA, ESA, and CA. The large negative values in the eastern part (east of 15° E) might be related to proximity to the edge of the network and thus be dominated by mantle phases (faster arrivals). Surprisingly, in between the negative values in the Alpine Chain, a pattern of slight positive corrections in the WSA is observed. Besides, extreme positive corrections are seen in the PoB and MoB as it is expected for

450 sedimentary basins. The pattern of stations in the ESA and a few stations in the WSA and CA agree very well with results by Diehl et al. (2009b).

A detailed interpretation of corrections pattern is highly ambiguous because it contains an overlay site effect and/or other 3-D variations such as Moho topography. It proved to be useful for accurately localizing earthquakes, for subsequent 3-D inversion (i.e., local earthquake tomography), and to show the consistency of phase data through inspection of the general pattern.

455 A detailed interpretation of the pattern of the station corrections is highly ambiguous because they contain a superposition of site effects and/or effects from 3–D structural variations as well as a lot of smearing. Nevertheless, the general concept of simultaneously inverting local earthquake datasets for a (simplified) 1-D velocity model, hypocenter positions/origin times and station corrections proved to be very powerful for accurately localizing earthquakes (see e.g., Kissling, 1988). Moreover, the general pattern of corrections can indicate the consistency of phase data.

[Figure]

**Figure 9. (a)** Left panel: probability of number of layers of the random models introduced by Markov chains during the evolution (color indicates number of models; please see color palette table on the left). Right panel: histogram of number of layers of the models after burn-in phase. **(b)** $V_P$, $V_P/V_S$, and $V_S$ models after McMC simultaneous inversion. Figure characteristics are similar to Fig. 7. The velocity models are well recovered down to ∼63 km depth. **(c)** P-wave station-corrections of all the stations that were involved in the inversion corresponding to the 1-D velocity model in **(b)**. Negative corrections (red circles) indicate earlier arrivals (indicative for higher velocities in the shallow subsurface) and positive corrections (blue crosses) indicate delayed arrivals (representative for lower velocities underneath the station). The faults are same as in Fig. 2. Abbreviations: **CA** - Central AlpineAlps; **EA** - Eastern AlpineAlps; **ESA** - Eastern Southern Alps; **WSA** - Western Southern Alps; **MoB** - Molasse Basin; **PoB** - Po Basin.

**6.2 Estimation of hypocenters accuracy based on relocation of quarry blasts**

To validate the localization procedure, the detected 15 quarry blasts (based on manual/visual inspections; see Sect. 3.2) , which were independently relocated by McMC using the previously derived 1-D velocity model and station-corrections from the simultaneous inversion (Sect. 6.1). Fig. 10 focuses on the blasts distribution associated with two quarr< areas close to the villages of Albiano (Italy) and Gummern (Villach, Austria). After the relocation of the blasts, we see that the epicenters are located within the quarry areas and the depths are in the range of the quarry topography (considering the average and theuncertainty of $1\sigma$), some of them are offset by a maximum of hundreds of meters. Based on the average mislocations of the blasts (relative to the centers of the quarry areas) and also their location uncertainties, we estimate the absolute location errors in the range of 1 km horizontally and 500 m vertically. This indicates that, although the number of picks (especially S-picks) is generally lower for blasts, the McMC routine provides high precision hypocentral solutions. We expect that the accuracy of earthquake hypocentersis even better because they have usually a higher number of S-observations. It is expected that the errors of earthquakes (which are potentially deeper than blasts) are smaller than those estimated from this test because they are less affected by the heterogeneous shallow structure which was poorly accounted for by the model (Kissling, 1988; Husen et al., 1999).

The hypocentral solution of the blasts, in comparison with those obtained by INGV/ZAMG, have an average difference of $\sim$160 m in longitude, $\sim$1 km in latitude, and $\sim$4.6 km in depth, so the depths are considerably better resolved and recovered in our study, due to the availability of the denser SWATH-D network.

[Figure]

**Figure 10. (a)** Map view showing the location of quarry blasts relocated using single-event mode of McMC, with solutions of the 1-D velocity model and station-corrections from the simultaneous inversion of earthquakes. **(b)** and **(c)** Epicenter and depth distribution of the blasts associated to two quarries close to Albiano in Italy and Gummern in Austria, respectively. The light gray band in the cross sections shows the surface elevation variation within the map-view boundary. The dark gray zones show the location  and surface elevations of the quarries. The depth histogram is also shown for each quarry (the bar in the depth histograms displays the surface elevation within the quarry area). Please note that elevations above the sea level are shown with negative values.

**6.3 Seismicity distribution**

The distribution of seismicity in the Southern and Eastern Alps is shown in map view and cross sections in Figs 11 and 12. This includes the same 301 events used for the simultaneous inversion (Sect. 6.1) as well as 43 additional earthquakes with slightly

480   fewer picks for a total of 344 earthquakes, 12,534 P-picks, and 7,265 S-picks lower quality (see Sec. 3.2). For this relocation, we used the modified average $V_P$ and $V_P/V_S$ models, station-corrections, and noise levels resulted from the simultaneous McMC inversion. After relocation, the depth probability distribution of 9 earthquakes were not Gaussian and therefore, due to their high depth uncertainty we decided to remove these earthquakes from our catalog. The hypocentral uncertainties ($1\sigma$) based on statistical analyses of the results of the McMC inversion are 1.75 km in depth and ∼500 m for the epicenters. This

485   is in agreement with the synthetic tests (Sect. 5) and the relocation of the quarry blasts (Sect. 6.2). The list of earthquakes is available in the supplementary material S2. The average differences between the epicentral positions derived in this study and those provided by the agencies are2.3 km, with maximum differences of 11.5 km. The differences are even larger for depth (average 2.9 km and maximum 12.5 km).Based on statistical analyses of the McMC inversion, the average epicentral and

depth uncertainties (1$\sigma$) are $\sim$500 m and $\sim$1.7 km which are compatible with precision estimates by the synthetic test (Sect. 5). However, the absolute location errors estimated by quarry blasts test (Sect. 6.2) are $\sim$1 km for epicenter and $\sim$500 km for depth.

Since the national agencies are (probably) using much less data for the location (smaller number of stations used, larger inter-station distances), a significant difference between their hypocenter solutions and those obtained in this study is expected (average of 2.4 km in epicenter and 3.7 km in depth). The earthquake depths calculated by McMC are systematically shallower than those by national agencies (by an average of 1.1 km). The maximum and minimum differences in the epicenters and depths (between McMC and national catalogs) are seen for the earthquakes from the INGV and SED, respectively.

The derived hypocenters in this study are not a representative seismicity catalog for the region (the national catalogs contain many small, poorly locatable events in a much longer period) but form excellent data for further seismological studies, e.g., Local Earthquake Tomography (LET). Moreover, this highly-precise hypocentral data allows further tectonic inferences. The hypocenter solutions of constrained earthquakes derived in this study is available in the supplementary material S2.

[revised manuscript text omitted]

In this study, a semi-automatized picking procedure was applied to the waveform data recorded by SWATH-D and selected AlpArray network stations of earthquakes listed in national seismic catalogs. During the visual inspection of the picks, several quarry blasts were identified and separated from the earthquakes. In order to establish a very high-quality, consistent earthquake catalog in the study region, the arrival-time data of the best locatable events (in terms of azimuthal gap, RMS, and number of 610 picks) were inverted simultaneously for hypocenters, 1-D velocity model, and station-corrections using a McMC approach. A synthetic recovery test , using a close-to-realistic 3-D velocity model, validated the reliability of the method for yielding highly precise hypocenters even in complex 3-D velocity conditions. A quarry blast test demonstrated that the blast hypocenters are located very close to the quarry areas with average horizontal and vertical mislocations (relative to the centers of the quarry areas) of 740 m and 376 m, respectively. The average uncertainties calculated by McMC for the earthquake locations are ∼500 615 m in epicenter and ∼1.7 km in depth. The derived $V_P$ and $V_P/V_S$ models are well-resolved down to a depth of 30 km with an uncertainty below 0.45 km s$^{-1}$ and 0.05, respectively. At greater depth the resolution decreases. Below 30 km depth, velocities gradually increase to mantle velocity which is reached at around 50 km depth. The station-corrections demonstrate coherent

negative values associated with the EA, ESA, and CA and slight positive values in the WSA. Moreover, the PoB and MoB represent extreme positive corrections, as it is expected for sedimentary basins.

620    The study is based on a very dense and large station network, however, it is limited to 16 months. Nevertheless, our precise hypocenter solutions with high internal consistency allow us to have insight into the active tectonic situation throughout the study region. The seismicity pattern confirms previous seismotectonic studies based on long-term records or local networks. Seismicity is found in diffuse clusters in the upper crust (0-20 km) and within broader zones of the Alpine frontal thrust, e.g., FV region, along the GL and Schio-Vicenza domains, and in the Internal Alpine and Inntal areas. The northeastern corner of

625    Adriatic or Dolomites Indenter is obviously inactive or low-active. We think that the overall pattern of seismicity reflects the head-on convergence of the Adriatic Indenter with the Alpine orogenic crust, accommodated in the FV region and along the Giudicarie Belt. The seismicity in the FV and GL regions shows activities deeper than the modeled frontal thrusts, which we interpret as an indication for southward propagation of the Southern Alpine deformation front (blind thrusts). The seismicty in FV and GL indicates similar earthquake depths for both regions, while the sparse seismicity between them may indicate that

630    the deformation within this area is most likely occurring aseismically.

The 1-D $V_P$ and $V_P/V_S$ models and the reassessed precise hypocenters will be utilized for investigating 3-D seismic velocities in the region using LET. The earthquake database is provided in the supplementary materials S1 and S2.

**Appendix A:**  **Integration of the seismic event catalogs**

The process of integrating events from the national seismological agencies is summarized in Fig. A1. It starts with getting

635    events between September 2017 and the end of 2018 and confined to our study area with $M_L \geq 1$ from catalogs of INGV, OGS, ZAMG, and SED. Thereafter, three individual event lists of A for Italy, Slovenia, and Croatia from INGV, B for Germany and Austria from ZAMG, and C for Switzerland from SED are selected. The lists will be subsequently updated by other 3 catalogs (one at a time as described in Fig. A1). Eventually, the three lists are merged in to one list which forms 2,639 local events in the study region.

[Figure]

**Figure A1.** Flow chart showing the process of merging seismic catalogs of INGV, OGS, ZAMG, and SED in order to derive an integrated seismic event list in the study region.

**Appendix B: Automatized Arrival-Time Picking and Earthquake Relocation**

**B1 Workflow**

Our modified workflow consists of three main stages: 1) the NonLinLoc ray-tracer (Lomax et al., 2000) for predicting initial P arrival-times based on preliminary origin-time and hypocenters (integrated earthquake catalog from national agencies) and the minimum 1-D velocity model of Diehl (2008); 2) the MPX algorithm (Aldersons, 2004; DiStefano et al., 2006) for improving the first alerts of P-picks and classifying them into four quality classes (0 to 3, Table B2) based on Fisher statistics (Diehl, 2008, ;Appendix D: Users guide for MPX picking system); 3) the Spicker algorithm (Diehl et al., 2009a) for picking S-onsets based on P-picks, event location, and a preliminary S-pick (determined by NonLinLoc ray-tracing; Lomax et al., 2000). Spicker also rates picks into four quality classes (Table B2).

After each stage, in order to remove the effect of low-quality picks on hypocenter locations, a multiple relocation procedure using HYPO71 (Lee and Lahr, 1975) and various subsets of picks is performed. The procedure starts with localization of the events using the most reliable picks (higher quality classes and closer distance to the event) and this first location is considered to be the reference for testing further picks. Thereafter, the picks with lower quality and those from distant stations are incorporated one-by-one. The newly added pick is kept if its residual is less than a distance-dependent, user-defined value and also if the root mean square (RMS) residual of the event location is still acceptable (less than 1 s). Picks leading to not

655 acceptable RMS residuals are removed from the dataset. Moreover, in this relocation procedure, not only low-quality picks but also events with a small number of high-quality picks are removed. This procedure results in not dominating the location by low-quality picks or not sorting out better picks. Since MPX and Spicker depend on the correctness of the predicted arrival-times (the predicted arrival-time itself depends on the event location), therefore picks and location are highly inter-dependent. This can be controlled by iteratively improving the event location, repicking arrival-times based on the updated location, and

660 adding or removing dubious picks in each step.

The implementation of this automatic workflow for the event list by national catalogs (appendix A) resulted in a dataset with 2,639 events, 68,099 P- and 17,151 S-Picks. The statistics of the picking quality classes are given in Table B1.

**Table B1.** Distribution of the picks and their quality classes after implementation of the automatic picking algorithms for the event list created by national catalogs.

| Quality class | P-picks (#) | S-picks (#) |
|:---:|:---:|:---:|
| 0 | 5,362 | 880 |
| 1 | 18,353 | 1,624 |
| 2 | 14,055 | 2,418 |
| 3 | 30,329 | 12,229 |
| sum (#) | 68,099 | 17,151 |

**B2  Uncertainty assessment of arrival-time picks**

665 To assess the performance of the picking algorithms, a dataset consisting of 14 earthquakes with various magnitude ranges and randomly scattered in the region, are manually analyzed. Based on the arrival-time uncertainty, the P- and S-picks are qualified into 5 classes (0, 1 ,2, 3, and 4 meaning respectively 100%, 75%, 50%, 25%, and 0% of contribution into location) and they are considered as reference for calibration (Table B2).

**Table B2.** Statistics of the automatic and reference picks associated with the quality classes for a dataset of 14 earthquakes.

| pick quality class | #auto picks | #reference picks | manual arrival-time uncertainty (s) | auto-picking $\mu^*$ (s) | auto-picking SD** (s) |
|---|---|---|---|---|---|
| P-all | 757 | 871 | - | 0.071 | 0.189 |
| P-0 | 195 | 299 | <0.05 | 0.001 | 0.039 |
| P-1 | 245 | 156 | 0.1 | 0.063 | 0.123 |
| P-2 | 160 | 171 | 0.2 | 0.158 | 0.283 |
| P-3 | 157 | 120 | 0.3 | 0.061 | 0.384 |
| P-4 | - | 125 | >0.3 | - | - |
| S-all | 343 | 445 | - | 0.103 | 0.642 |
| S-0 | 21 | 69 | <0.1 | 0.056 | 0.053 |
| S-1 | 56 | 61 | 0.2 | 0.07 | 0.132 |
| S-2 | 77 | 108 | 0.3 | 0.115 | 0.235 |
| S-3 | 189 | 121 | 0.4 | 0.223 | 0.817 |
| S-4 | - | 86 | >0.4 | - | - |

\* The mean of misfits between automatic and reference picks

\*\* The standard deviation of misfits between automatic and reference picks

The misfits between reference and automatic arrival-times are calculated (individually for each quality class of the P- and S-picks) and the mean and standard deviation values are shown in Table B2. The picks with higher qualities have smaller mean values which means their misfits are lower (*i.e.,* automatic picks are closer to reference picks). Moreover, except for S-3, the standard deviation of each class is very well comparable with manual arrival-time uncertainty.

Another way of assessing the performance of the automatic picking algorithms is the so-called confusion matrix. The confusion matrix, in general, shows the performance of predicted vs. the actual data of different classifications in machine learning (Kohavi and Provost, 1998). For our dataset, it shows automatic picks vs. reference picks (similar to DiStefano et al., 2006; Diehl et al., 2009a; Sippl et al., 2013). In the confusion matrices displayed in Fig. B1, three values represent the performance of the automatic pickers in each quality class: 1) number of picks, 2) hit-rate (fraction of reference picks of a certain class that are assigned to a specific class by the automatic picking) that is summed up to 100% in each row, and 3) the standard deviation of the residuals between automatic and reference picks. The red and orange colors refer to picks that are upgraded to higher qualities by the automatic algorithms (unfavorable), whereas the green sectors show the picks that are underestimated by the automatic picker (more eligible). Having low hit-rates in the orange and red cells and high hit-rates in the green fields point toward a conservative assessment of the picking algorithms. The white fields in the confusion matrices illustrate the picks that are ideally qualified based on the reference dataset.

| Reference P-picks | Automatic P-picks from MPX | | | | |
|---|---|---|---|---|---|
| | 0 | 1 | 2 | 3 | ignored |
| 0 | N=158 53.38% σ=0.039 | N=81 27.36% σ=0.124 | N=13 4.39% σ=0.069 | N=19 6.42% σ=0.301 | N=25 8.45% - |
| 1 | N=26 16.77% σ=0.089 | N=61 39.35% σ=0.123 | N=31 20.01% σ=0.124 | N=19 12.26% σ=0.300 | N=18 11.61% - |
| 2 | N=6 3.59% σ=0.009 | N=38 22.75% σ=0.173 | N=41 24.55% 0.283 | N=30 17.96% σ=0.273 | N=52 31.15% - |
| 3 | N=2 1.69% σ=0.070 | N=21 17.8% σ=0.214 | N=23 19.49% σ=0.166 | N=21 17.8% σ=0.384 | N=51 43.22% - |
| 4 or ignored | N=1 0.43% - | N=40 17.17% - | N=51 21.89% - | N=65 27.89% - | N=76 32.62% - |

| Reference S-picks | Automatic S-picks from Spicker | | | | |
|---|---|---|---|---|---|
| | 0 | 1 | 2 | 3 | ignored |
| 0 | N=9 13.04% σ=0.053 | N=8 11.60% σ=0.088 | N=6 8.69% σ=0.250 | N=8 11.60% σ=0.665 | N=38 55.07% - |
| 1 | N=3 5% σ=0.063 | N=11 18.33% σ=0.132 | N=10 16.67% σ=0.319 | N=7 11.67% σ=0.586 | N=29 48.33% - |
| 2 | N=4 3.74% σ=0.081 | N=15 14.02% σ=0.140 | N=17 15.89% σ=0.235 | N=19 17.75% σ=0.882 | N=52 48.60% - |
| 3 | N=2 1.67% σ=0.074 | N=7 5.83% σ=0.115 | N=15 12.5% σ=0.227 | N=32 26.67% σ=0.817 | N=64 53.33% - |
| 4 or ignored | N=2 0.91% - | N=15 6.85% - | N=28 12.79% - | N=121 55.25% - | N=53 24.20% - |

**Figure B1.** Confusion Matrices of P- and S-picks, comparing the automatic picks with reference picks. Columns are quality classes of the automatic P-picks from MPX (upper table) and S-picks from Spicker (lower table), and rows represent the pick weighting classes provided by the human analyst for reference data. The parameters in each cell of the matrices are explained in the text.

The confusion matrices of both MPX and Spicker show relatively high values of hit-rate (alternatively N) in some orange and red cells that we interpret as mispicks. For instance, the MPX (Spicker) picked 157 P-picks (166 S-picks) that are either ignored or qualified as class 4 by a human analyst. As the human analyst adopted class 4 for far stations with unclear arrivals, far automatic picks are unreliable. On the other hand, it seems that Spicker ignored a large number of picks (high hit-rates in the most right column of the Spicker matrix). This is eminent for picks with manual quality class 0 that are potentially associated with closest stations based on human analysis.

*Data availability.* The SWATH-D data is archived at the GEOFON datacenter (http://eida.gfz-potsdam.de/webdc3/; FDSN network code ZS, 2017-2019; Heit et al., 2017). Detailed information for accessing the data of permanent and temporary networks of AASN (network codes: BW, CH, GR, IV, MN, NI, OE, OX, RF, SI, SL, ST, Z3) are available in http://www.alparray.ethz.ch/en/seismic_network/backbone/data-access/. The earthquake hypocenters together with the $V_P$ and $V_P/V_S$ models established in this study are utilized in the attached .zip archive.

*Team list.* **The AlpArray Seismic Network Team:** György HETÉNYI, Rafael ABREU, Ivo ALLEGRETTI, Maria-Theresia APOLONER, Coralie AUBERT, Simon BESANÇON, Maxime BÈS DE BERC, Götz BOKELMANN, Didier BRUNEL, Marco CAPELLO, Martina CARMAN, Adriano CAVALIERE, Jérôme CHÈZE, Claudio CHIARABBA, John CLINTON, Glenn COUGOULAT, Wayne C. CRAWFORD, Luigia CRISTIANO, Tibor CZIFRA, Ezio D'ALEMA, Stefania DANESI, Romuald DANIEL, Anke DANNOWSKI, Iva DASOVIC, Anne DESCHAMPS, Jean-Xavier DESSA, Cécile DOUBRE, Sven EGDORF, ETHZ-SED Electronics Lab, Tomislav FIKET, Kasper FISCHER, Florian FUCHS, Sigward FUNKE, Domenico GIARDINI, Aladino GOVONI, Zoltán GRÁCZER, Gidera GRÖSCHL, Stefan HEIMERS, Ben HEIT, Davorka HERAK, Marijan HERAK, Johann HUBER, Dejan JARIC, Petr JEDLICKA, Yan JIA, Hélène JUND, Edi KISSLING, Stefan KLINGEN, Bernhard KLOTZ, Petr KOLÍNSKÝ, Heidrun KOPP, Michael KORN, Josef KOTEK, Lothar KÜHNE, Krešo

KUK, Dietrich LANGE, Jürgen LOOS, Sara LOVATI, Deny MALENGROS, Lucia MARGHERITI, Christophe MARON, Xavier MARTIN, Marco MASSA, Francesco MAZZARINI, Thomas MEIER, Laurent MÉTRAL, Irene MOLINARI, Milena MORETTI, Anna NARDI, Jurij PAHOR, Anne PAUL, Catherine PÉQUEGNAT, Daniel PETERSEN, Damiano PESARESI, Davide PICCININI, Claudia PIROMALLO, Thomas PLENEFISCH, Jaroslava PLOMEROVÁ, Silvia PONDRELLI, Snježan PREVOLNIK, Roman RACINE, Marc RÉGNIER, Miriam REISS, Joachim RITTER, Georg RÜMPKER, Simone SALIMBENI, Marco SANTULIN, Werner SCHERER, Sven SCHIPPKUS, Detlef SCHULTE-KORTNACK, Vesna ŠIPKA, Stefano SOLARINO, Daniele SPALLAROSSA, Kathrin SPIEKER, Josip STIPCEVIC, Angelo STROLLO, Bálint SÜLE, Gyöngyvér SZANYI, Eszter SZUCS, Christine THOMAS, Martin THORWART, Frederik TILMANN, Stefan UEDING, Massimiliano VALLOCCHIA, Ludek VECSEY, René VOIGT, Joachim WASSERMANN, Zoltán WÉBER, Christian WEIDLE, Viktor WESZTERGOM, Gauthier WEYLAND, Stefan WIEMER, Felix WOLF, David WOLYNIEC, Thomas ZIEKE, Mladen ŽIVCIC, Helena ŽLEBCÍKOVÁ. **The AlpArray SWATH-D Field Team:** Luigia Cristiano (Freie Universität Berlin, Helmholtz Zentrum Potsdam Deutsches GeoForschungsZentrum - GFZ), Peter Pilz, Camilla Cattania, Francesco Maccaferri, Angelo Strollo, Günter Asch, Peter Wigger, James Mechie, Karl Otto, Patricia Ritter, Djamil Al-Halbouni, Alexandra Mauerberger, Ariane Siebert, Leonard Grabow, Susanne Hemmleb, Xiaohui Yuan, Thomas Zieke, Martin Haxter, Karl-Heinz Jaeckel, Christoph Sens-Schonfelder (GFZ), Michael Weber, Ludwig Kuhn, Florian Dorgerloh, Stefan Mauerberger, Jan Seidemann (Universität Potsdam), Frederik Tilmann, Rens Hofman (Freie Universität Berlin), Yan Jia, Nikolaus Horn, Helmut Hausmann, Stefan Weginger, Anton Vogelmann (Austria: Zentralanstalt für Meteorologie und Geodynamik - ZAMG), Claudio Carraro, Corrado Morelli (Südtirol/Bozen: Amt für Geologie und Baustoffprüfung), Günther Walcher, Martin Pernter, Markus Rauch (Civil Protection Bozen), Damiano Pesaresi, Giorgio Duri, Michele Bertoni, Paolo Fabris (Istituto Nazionale di Oceanografia e di Geofisica Sperimentale OGS - CRS Udine), Andrea Franceschini, Mauro Zambotto, Luca Froner, Marco Garbin (Ufficio Studi Sismici e Geotecnici -Trento) (also OGS).

*Author contributions.* AJN performed data preparation, analysis, and simultaneous inversion; she also prepared all figures and most of the text. CH supervised the analysis and contributed to the interpretation of the results. TR assisted with the inversion and modified the code. VV, ELB, and MRH contributed to the interpretation and the discussion of results. MW promoted the work and performed text corrections. All coauthors contributed to the text and the figures.

*Competing interests.* The authors declare that they have no conflict of interest.

*Acknowledgements.* We would like to express our appreciation to Ben Heit for the coordination of the SWATH-D network, all people involved in the instrument preparation, field work, and data archiving (Team list above; http://www.alparray.ethz.ch/en/research/compl ementary-experiments/swath-d/overview/), also to numerous landowners for their willingness to host the stations, and all communities, authorities, and institutes in the region for their great support of the project. Funding for the network and research work came from the German Science Foundation DFG through the Priority program SPP 4D-MB and the GFZ Potsdam. Instruments for the SWATH-D network were provided by the Geophysical Instrument Pool Potsdam GIPP of the GFZ Potsdam. We also would like to acknowledge the AlpArray Seismic Network Team (Team list above; http://www.alparray.ethz.ch/en/seismic_network/backbone/backbone/). We thank the national

seismological agencies of Italy (INGV, OGS), Austria (ZAMG), and Switzerland (SED) for their comprehensive earthquake catalogs. We also appreciate Christian Sippl for his guidance in implementing the automatic workflow. Discussions with many colleagues within the Priority program are greatly acknowledged. For plotting the figures, the Generic Mapping Tools (GMT; Wessel and Smith, 1991; Wessel et al., 2019, https://www.generic-mapping-tools.org/) are used. The authors would like to thank reviewer Edi Kissling and two other anonymous reviewers for their thorough and constructive remarks which led to significant improvement of the manuscript.

735

**References**

[revised manuscript text omitted]